# Forest edges increase pollinator network robustness to extinction with declining area

Peng Ren [1], Raphael K. Didham [2,3], Mark V. Murphy [2], Di Zeng [4], Xingfeng Si [4] & Ping Ding [1] ✉

Edge effects often exacerbate the negative effects of habitat loss on biodiversity. In forested ecosystems, however, many pollinators actually prefer open sunny conditions created by edge disturbances. We tested the hypothesis that forest edges have a positive buffering effect on plant-pollinator interaction networks in the face of declining forest area. In a fragmented land-bridge island system, we recorded ~20,000 plant-pollinator interactions on 41 islands over 3 yr. We show that plant richness and floral resources decline with decreasing forest area at both interior and edge sites, but edges maintain 10-fold higher pollinator abundance and richness regardless of area loss. Edge networks contain highly specialized species, with higher nestedness and lower modularity than interior networks, maintaining high robustness to extinction following area loss while forest interior networks collapse. Anthropogenic forest edges benefit community diversity and network robustness to extinction in the absence of natural gap-phase dynamics in small degraded forest remnants.

Habitat loss has profound consequences for pollinator diversity[1], and the resulting habitat loss has become one of the leading causes of species endangerment[2], extinction of ecological interactions[3] and consequent loss of essential pollination services[2,4]. Moreover, ecosystem decay within small habitat remnants can double biodiversity loss due to habitat loss[5]. Increasing fragmentation and subdivision of habitats following habitat loss exacerbates ecosystem decay[6]. In particular, multiple components of fragmentation, such as increasing edge effects and decreasing connectivity of remnant habitats, dramatically alter the spatial distribution of pollinator resources[1,2,6,7] with far-reaching impacts on species interaction networks[4,8–11]. The conventional narrative is that the combined effects of habitat fragmentation on plant-pollinator communities tend to exacerbate the negative effects of habitat loss alone[1,9,12,13]. However, this generalization stands in striking contrast to empirical field observations that open light-filled forest edges support more flowers and more pollinators than the dark interior within closed-canopy forests[6,14–17].

The mismatch in observations is almost certainly because most studies on the effects of habitat fragmentation on plant-pollinator communities have focused on open-habitat systems such as grasslands or croplands where edge effects are probably negative or neutral[8–10,12,13], rather than on closed-canopy forest systems in which forest edge effects might be positive. Historically, natural gap-phase dynamics in continuous old-growth forests would have provided space and resources for light-loving pollinator species, but these habitats are now rare in contemporary degraded and regenerating forests[18–21]. Even so, the edges of secondary forests may provide an anthropogenic analogue of natural gap-phase conditions, and thus have a positive rather than negative edge effect on pollinator communities in forest systems. Surprisingly, comparatively little is known about the potential for positive edge effects on plant-pollinator communities in forest systems beyond a limited focus on abundance or species richness[15,22,23], rather than the potential network-wide consequences of positive edge effects mitigating the negative effects of forest loss on network structure.

[1]MOE Key Laboratory of Biosystems Homeostasis and Protection, College of Life Sciences, Zhejiang University, Hangzhou, Zhejiang, China. [2]School of Biological Sciences, The University of Western Australia, Crawley, Western Australia, Australia. [3]CSIRO Health and Biosecurity, Centre for Environment and Life Sciences, Floreat, Western Australia, Australia. [4]Zhejiang Zhoushan Archipelago Observation and Research Station, Institute of Eco-Chongming, Zhejiang Tiantong Forest Ecosystem National Observation and Research Station, School of Ecological and Environmental Sciences, East China Normal University, Shanghai, China. ✉e-mail: dingping@zju.edu.cn

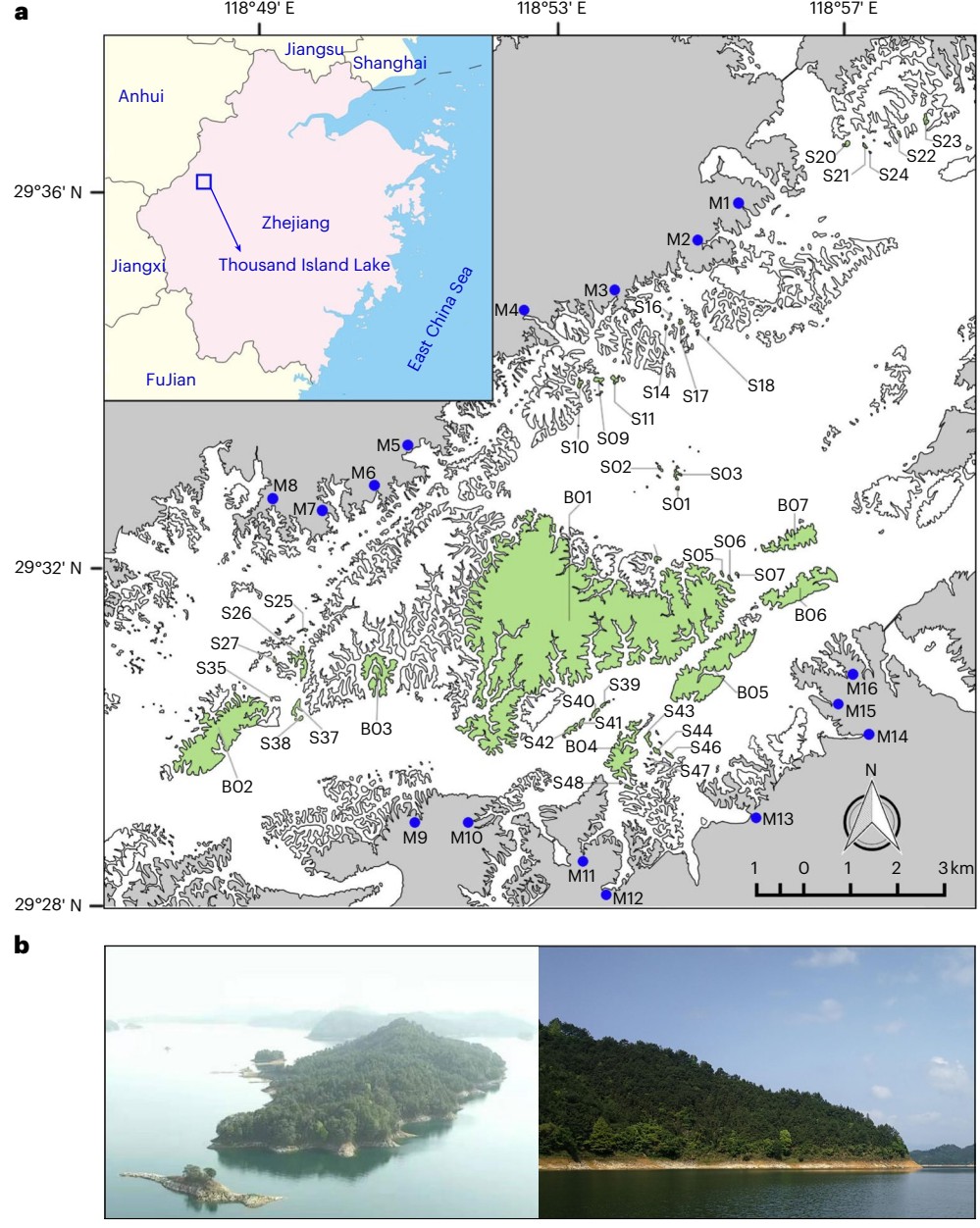

**Fig. 1 | Study site locations. a**, The 41 study islands (green shading; B01–B07, S01–S48) and the 16 mainland sampling sites (grey shading and blue dots; M1–M16) at the Thousand Island Lake, Zhejiang Province, eastern China (partially reproduced from Ren et al. [63]). **b**, The topography at the study site is mountainous, and flooding of the valley produced complex island shapes. Photo credits: P.R.

At the whole network scale, habitat loss has well-recognized negative effects on plant–pollinator interactions and network architecture[8–10,24–26], based predominantly on studies in non-forest systems. Habitat loss causes non-random loss of interactions[8,27–29], which disrupts plant–pollinator interactions, leading to higher network modularity and lower nestedness[9,26], which will intensify competition among species at the same trophic level[30,31] and destabilize networks in the face of disturbance[24,32]. Meanwhile, network connectance typically increases as habitat area declines[9], suggesting that fewer potential resource linkages remain in the network[33,34]. Although plant–pollinator networks can be relatively robust to changing spatial configuration of habitat fragments, by responding to species loss through high potential for adaptive switching of interaction partners[35–37], habitat fragmentation is widely considered to have a negative exacerbating influence on the impacts of habitat loss alone[1,9,12,13]. Once again, these generalizations appear to be largely driven by studies in non-forest systems, whereas if open light-filled forest edges have a contrasting positive effect on plant–pollinator networks within forests, then we would expect to see opposite effects with edges buffering rather than exacerbating changes in network robustness in the face of forest area loss.

Here we use structural equation models (SEM) to test the hypothesis that forest edges buffer the negative effects of forest area loss on plant–pollinator community structure and network architecture. The SEM partitions the direct and cascading indirect effects of forest fragmentation on floral resource availability, plant and pollinator community structure and the architecture of species interaction networks, using ~20,000 flower visitation records from 68 flowering plant species on 41 islands and 16 mainland sites in the Thousand Island Lake (TIL; Fig. 1) region of eastern China. The TIL system offers a unique opportunity to overcome potential confounding influences of heterogeneous initial

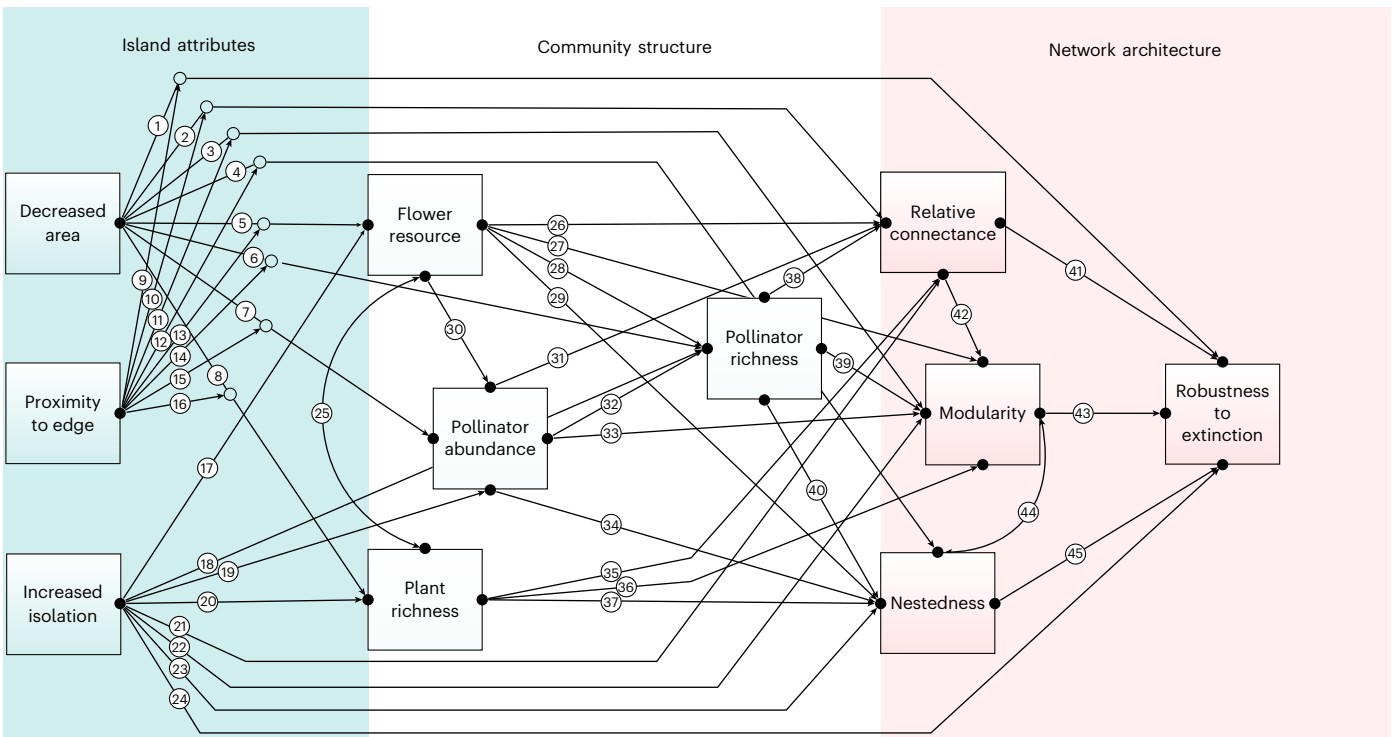

**Fig. 2 | Hypothesized path structure for the piecewise SEM model showing the direct and indirect pathways through which habitat fragmentation influences plant-pollinator community structure and network architecture.** Squares represent measured variables and arrows represent the direction of hypothesized causal relationships among variables. Double-headed arrows represent correlations. Interaction effects between predictors are shown by one causal path intersecting another causal path (with an open circle shown at the junction). Circled numbers along each path are assigned path numbers, and the final SEM paths are a subset of these. For further details, see Supplementary Methods 4.

starting conditions that might mask the relative effects of land-use change[38]. In 1959, all forests surrounding a new hydroelectric dam in eastern China were clear-felled at the same time, effectively resetting plant-pollinator community re-assembly to a common set of initial starting conditions[39]. The continuous mainland sites thus provide an ideal 'unfragmented' reference state against which to examine changes in plant-pollinator community structure and network architecture on islands of varying area, distance of isolation and degree of edge influence. We show that forest edges have 10-fold higher pollinator abundance and species richness than the forest interior, and maintain higher nestedness, lower modularity and greater robustness to cascading secondary extinction in the face of forest area loss, while interior forest networks collapse in small fragments. These results suggest that in fragmented forest systems, the creation of forest edges does not tend to exacerbate the negative effects of forest area loss on pollinators, but instead increases pollinator network robustness to extinction.

## Results

Over 3 yr we conducted 20 surveys of each pair of edge versus interior transects using time-standardized direct observations of flowers and hand-netting of insect pollinators, documenting a total of 19,486 individual pollinator interactions with plants in ~960 h of flower observations spanning 3,226 observed associations between 68 species of flowering plants and 313 species of pollinators (Supplementary Tables 1–3). Species similarity between islands and mainland was high, with 54 plant species and 269 pollinators in the mainland, 67 plant species and 310 pollinators on the 41 islands, and only 1 plant species and 3 pollinators that were not found on islands. Sample coverage was uniformly high for plants across islands (mean 0.89 ± 0.02 s.d.), but was lower for pollinators (0.84 ± 0.03) and for plant-pollinator interactions (0.68 ± 0.03), and declined noticeably with decreasing forest area

(Extended Data Fig. 1). Floral resources, pollinator abundance, and plant and pollinator richness at forest edges were up to 10-fold higher than in the interior (Supplementary Table 1), even after time-standardization of flower-pollinator interactions per hour to account for differential survey times per transect (Extended Data Fig. 2). Species composition of both plants and pollinators differed significantly between edge and interior (PERMANOVA: plants, d.f. = 1, F = 17.12, P < 0.001; pollinators, d.f. = 1, F = 13.64, P < 0.001) (Extended Data Fig. 3). Importantly, however, edge communities were not predominantly made up of generalist species and species in interior networks were not more specialized (Supplementary Results 1 and Extended Data Fig. 4).

In the final most-parsimonious piecewise SEM model (Figs. 2 and 3; Akaike information criterion, AIC = 141.83, Fisher's C = 51.83, P = 0.63), 24 of the 45 hypothesized paths were retained in the model (Supplementary Tables 4 and 5). There were striking effects of decreased area and increasing proximity to edge on plant and pollinator community structure (Fig. 3). The abundance and richness of both plants and pollinators declined significantly on smaller islands (negative effect sizes for standardized path coefficients in Figs. 3 and 4a–d). For floral resources and plant richness, null model comparisons (Extended Data Figs. 5–8) indicated that community metrics on most islands were significantly lower than expected based on passive sampling effects from the mainland reference pool (Fig. 4a,b). This indicates that declining forest area strongly constrains floral resource availability and host plant richness for pollinating insects. By contrast, the strong declines observed in pollinator abundance on smaller islands were similar to values that would be expected under a passive sampling model on the majority of islands (Fig. 4c). This suggests that lower sample coverage for pollinators on small islands (Extended Data Fig. 1) is an equally parsimonious explanation for the observed abundance trends, rather than necessarily fragmentation-driven impacts. Importantly, in the case of decreased

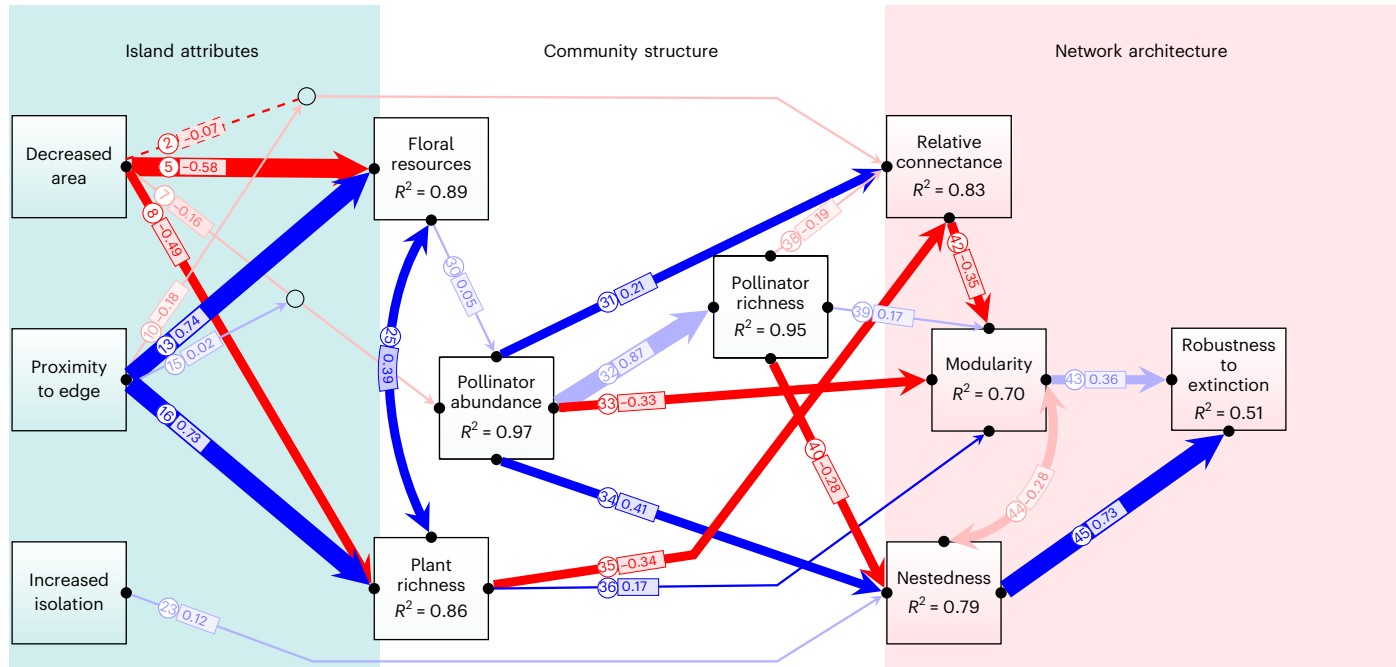

**Fig. 3 | Generalized multilevel SEM showing the direct and indirect pathways through which forest fragmentation influences plant-pollinator community structure and network architecture.** Overall, 24 paths remain in the final SEM model. Squares represent measured variables and arrows represent the direction of hypothesized causal relationships among variables. Interaction effects between predictors are shown by one causal path intersecting another causal path (with an open circle shown at the junction). Values along numbered paths are standardized partial regression coefficients. The width of each arrow is scaled to the absolute value of the standardized path coefficient. Blue vs red arrows represent significant ($P < 0.05$) positive vs negative paths, respectively. Dashed lines represent non-significant paths that were retained in the final best-fit SEM model. The lighter shades of blue or red arrows represent the path coefficients that do not deviate significantly from null SEM expectations ($P > 0.05$; see Fig. 5). The adjusted $R^2$ values within the squares show the explained proportion of variance in the response variables.

area effects on pollinator richness, the observed trends were driven predominantly by the indirect effect of decreased area on pollinator abundance (Figs. 4c,d and 5a), rather than any direct relationship between floral resources and pollinator richness or between plant richness and pollinator richness, suggesting that a passive sampling effect drives pollinator richness declines with decreasing forest area in this case.

Proximity to edge had even larger effect sizes on plant and pollinator community attributes than decreased area, and these effects were uniformly positive rather than negative (Fig. 3). Forest edges had significantly greater floral resources, plant species richness, pollinator abundance and pollinator richness than interiors (Fig. 4a–d). Unexpectedly, the edge to interior difference in community attributes was consistent across all islands (Fig. 4a–d), with area by edge interactions being weak or non-significant (Figs. 3 and 5a). Moreover, community responses to decreased area and proximity to edge were not influenced by distance of isolation from the mainland (Fig. 5a).

Network architecture varied significantly between edge and interior, and between small versus large islands (Fig. 4e–h). However, these effects were not due to direct influences of island attributes on network architecture, but instead occurred via the indirect influences of altered plant and pollinator community structure (Fig. 5b). There was only a weak positive effect of island isolation on nestedness, and a weak interaction effect of proximity to edge exacerbating the negative effect of decreased area on relative connectance (Figs. 3 and 5b). In both cases, observed effect sizes fell within the 95% CI of null model SEM simulations (Fig. 6), suggesting that these effects were not greater than would be expected from passive sampling effects from the mainland reference pool.

Indirect effects on network architecture varied between interior and edge communities. For forest interior communities, a decrease in forest area had a negative indirect effect on relative connectance

(that is, higher relative connectance within the smaller plant-pollinator assemblages on small islands compared with large islands; Fig. 4e), and significantly lower nestedness and higher modularity on small compared with large islands (Fig. 4f,g). For forest edge communities, by contrast, relative connectance was substantially lower and nestedness substantially higher than observed for forest interior communities, and these network parameters were comparatively less affected by declining area (Fig. 4e,f). These changes in network parameters had a cascading effect on network robustness to extinction. Higher nestedness at edges, in particular, had a strong positive effect (path 45 in Fig. 3) on robustness to extinction for edge communities compared with interior communities (Fig. 4h), while forest interior communities had a lower robustness to extinction on small compared with large islands (Fig. 4h). On large islands, there was 'only' a 10–20% higher robustness to extinction at edges compared with interiors, whereas on small islands there was a 50–100% higher robustness to extinction at edges compared with interiors (Fig. 4h).

## Discussion

Both forest area loss and changing spatial configuration of the remaining habitat can disrupt ecological networks[2,6]. In a naturally regenerating secondary forest system in eastern China, we showed that fragmentation processes (edge and forest area) do not always have compounding negative effects. Unanticipated cascading positive effects can act to mitigate, not exacerbate, biodiversity loss and network collapse. In our study system, forest edges had positive effects, buffering rather than exacerbating the decrease in network robustness to species extinction caused by forest area loss. Our SEM revealed the complex relationship among island attributes, community structure and network architecture in a fragmented forest system. Whereas the direct and indirect effects of declining habitat area on plant-pollinator community

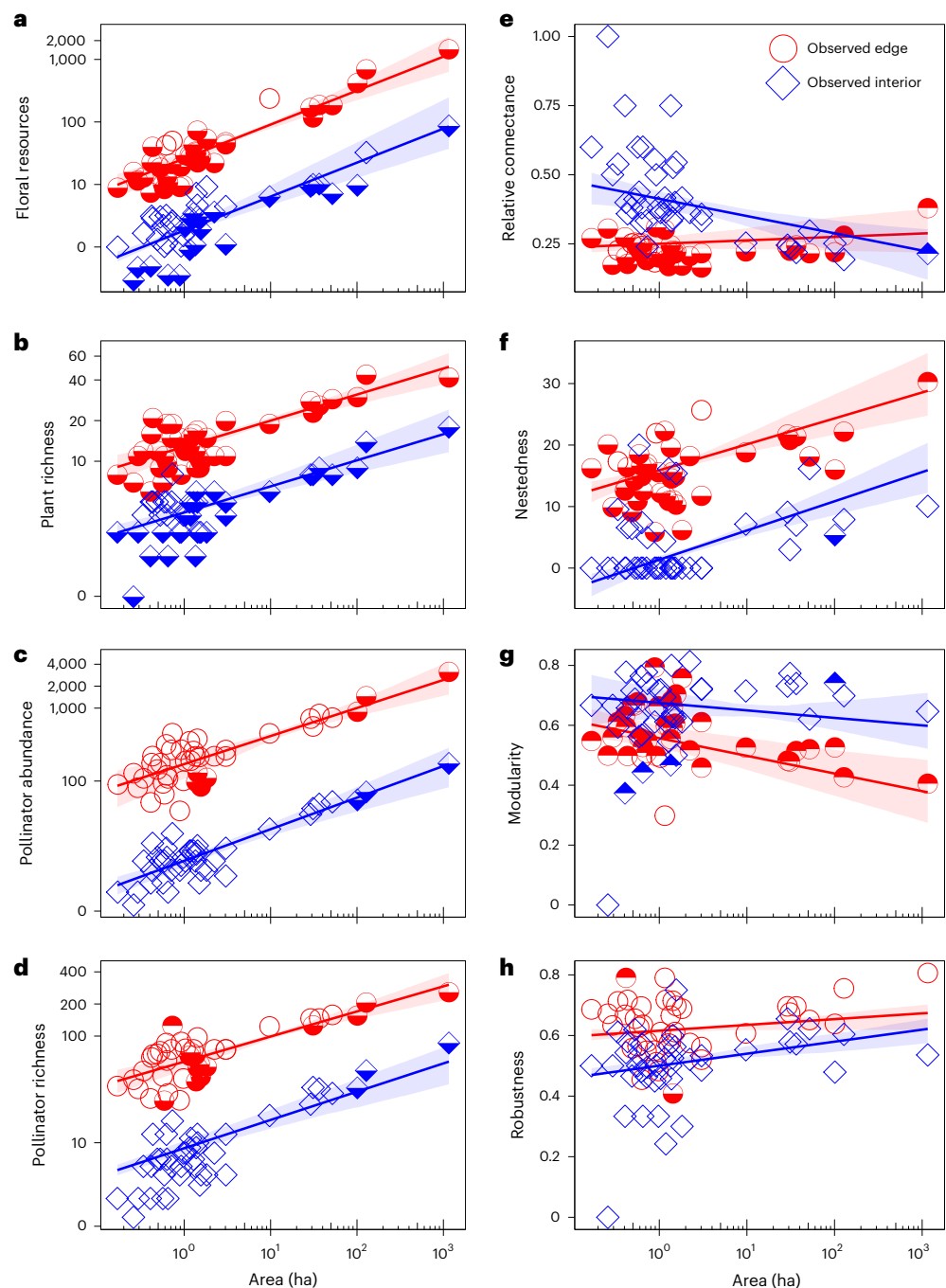

**Fig. 4 | Modelled effects of forest area and proximity to edge on community structure and network architecture. a–h**, Plots show the partial effects of area and proximity to edge after controlling for other predictor variables (by setting to mean values). Symbols represent the observed values on the 41 islands and shading within symbols indicates deviation of observed values from null model expectations (based on 1,000 random draws from the mainland reference network; see Methods for details). Open symbols: observed values within the 95% confidence interval (CI) of null model predictions; shading in the upper half of symbols: observed values greater than the 95% CI of null model predictions; shading in the lower half of symbols: observed values lower than the 95% CI of null model predictions.

structure and network architecture were predominantly negative, these were counter-balanced by predominantly positive effects of proximity to habitat edge. Edges maintained 10-fold higher pollinator abundance and richness than forest interiors, and these effects were not driven by increasing frequency of generalist plant-pollinator interactions at edges. Edge networks contained many highly specialized species, even when compared with interior networks. More importantly, edges increased network robustness to extinction rather than exacerbating

the negative effect of forest area loss on network architecture, thereby reducing the risk of plant-pollinator system collapse.

Positive effects of forest fragmentation on plant-pollinator community structure and network architecture may seem surprising at first glance, given the weight of evidence showing negative effects[1,9,12,13]. However, previous network studies have been conducted almost entirely in non-forest systems in which the relative contrast in light conditions and floral resources between edge and interior is weak (or

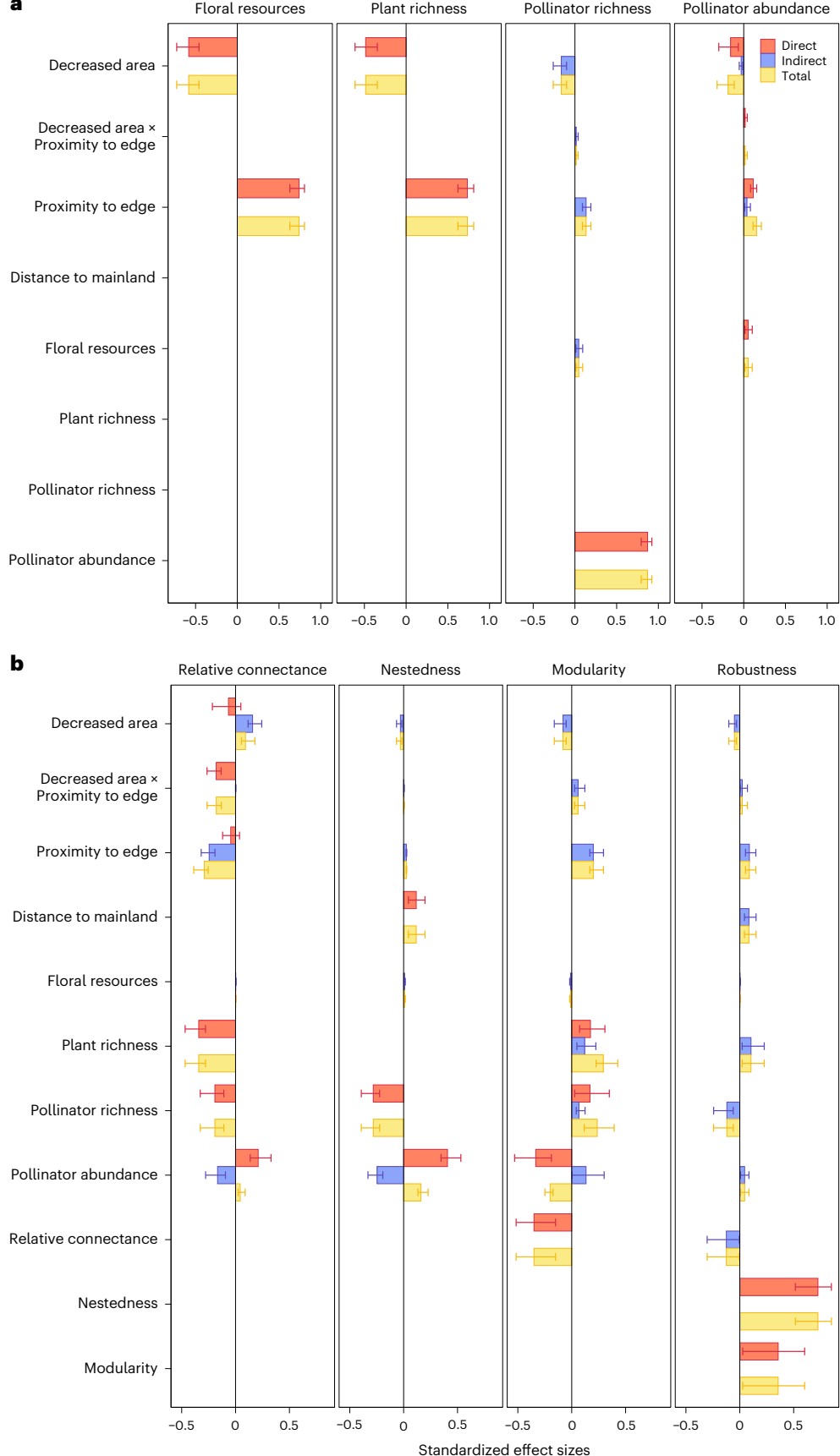

**Fig. 5 | Summary of direct, indirect and total effects of island attributes. a**, Effects on plant-pollinator community structure. **b**, Effects on network architecture. Bars and error bars represent mean ± 95% CI; $n = 1{,}000$ bootstrapped estimates for each response.

| Number | Path | Observed path coefficient | Null model CI | Observed value vs Null model |
|---|---|---|---|---|
| ② | DA→RC | −0.066 *** | [ 0.070, 0.407] | |
| ⑤ | DA→FR | −0.582 *** | [−0.405, −0.361] | |
| ⑦ | DA→AB | −0.157 | [−0.502, −0.120] | |
| ⑧ | DA→PL | −0.485 *** | [−0.130, −0.099] | |
| ⑩ | DA×PE→RC | −0.180 | [−0.297, −0.052] | |
| ⑬ | PE→FR | 0.741 *** | [ 0.881, 0.910] | |
| ⑮ | DA×PE→AB | 0.018 | [−0.025, 0.021] | |
| ⑯ | PE→PL | 0.734 *** | [ 0.842, 0.942] | |
| ㉓ | DM→NE | 0.117 | [−0.008, 0.196] | |
| ㉕ | FR↔PL | 0.388 ** | [ 0.465, 0.805] | |
| ㉚ | FR→AB | 0.053 | [−0.055, 0.064] | |
| ㉛ | AB→RC | 0.210 * | [−0.134, 0.185] | |
| ㉜ | AB→PO | 0.871 | [ 0.803, 0.882] | |
| ㉝ | AB→MO | −0.334 * | [−0.315, −0.071] | |
| ㉞ | AB→NE | 0.407 ** | [ 0.036, 0.329] | |
| ㉟ | PL→RC | −0.342 ** | [−0.317, 0.064] | |
| ㊱ | PL→MO | 0.174 *** | [−0.379, −0.076] | |
| ㊳ | PO→RC | −0.192 | [−0.203, 0.159] | |
| ㊴ | PO→MO | 0.170 | [ 0.012, 0.294] | |
| ㊵ | PO→NE | −0.282 ** | [−0.203, 0.088] | |
| ㊷ | RC→MO | −0.350 *** | [−0.747, −0.457] | |
| ㊸ | MO→RO | 0.356 | [−0.099, 0.416] | |
| ㊹ | MO↔NE | −0.275 | [−0.430, 0.006] | |
| ㊺ | NE→RO | 0.725 * | [ 0.300, 0.685] | |

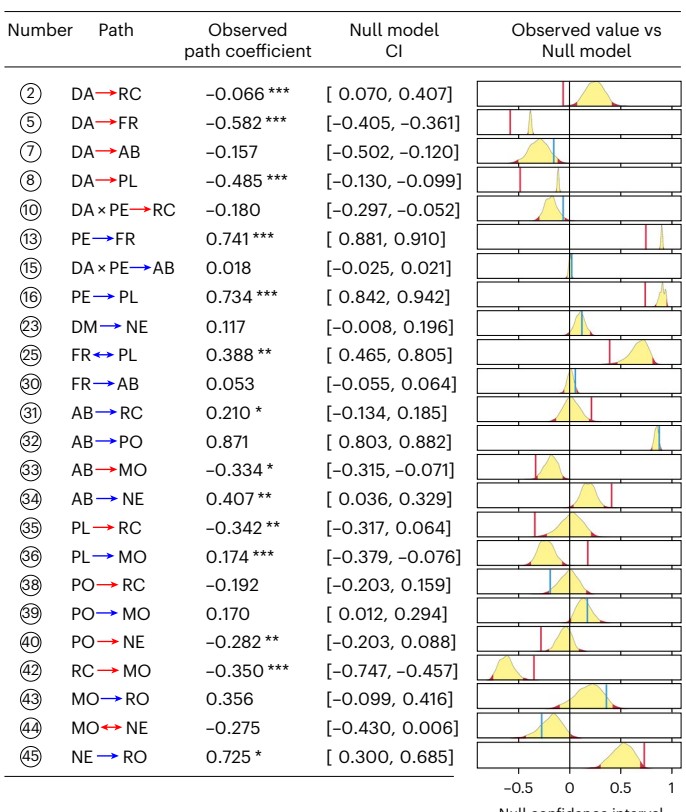

−0.5    0    0.5    1

Null confidence interval

**Fig. 6 | Observed standardized path coefficients for the SEM compared with null model estimates based on 1,000 random draws from the mainland reference pool.** Numbered circles correspond to the paths in Fig. 2. DA, decreased area; PE, proximity to edge; DM, distance to mainland; FR, floral resources; PL, plant richness; AB, pollinator abundance; PO, pollinator richness; RC, relative connectance; NE, nestedness; MO, modularity; RO, robustness to extinction. Blue and red arrows represent positive and negative path coefficients, respectively. Yellow shading in the probability distributions represents the 95% CI of path coefficients in the null SEM, and the red shaded areas are the zones outside the 95% CI. The red vertical lines indicate the position of observed path coefficient if it is significantly different from null expectation, while blue vertical lines indicate no significant deviation from null expectation. *P < 0.05, **P < 0.01, ***P < 0.001.

favours interior habitats), whereas our findings for the forested system at TIL are more in line with the observation that pollinators prefer open sunny conditions rather than the dark interior within closed-canopy forests. For example, on the islands we studied, forest canopy density was high (~90% under *Pinus massoniana*)[40], few shrub species could persist (for example, *Loropetalum chinense*, *Symplocos sumuntia* and *S. paniculata*), herbs were rare and only a few shade-tolerant species were seen flowering occasionally (for example, *Liriope spicata* and *Lysimachia congestiflora*). In other forest systems, floral resources in the sunny open environment of the upper canopy might be expected to have a positive effect similar to that described here for the forest edge[41,42], but the TIL forests are dominated by gymnosperms (predominantly *P. massoniana*) with relatively few flowering lianas (*Rosa multiflora*, *Wisteria sinensis*, *Millettia dielsiana*). However, floral resource availability in the forest canopy warrants greater attention, notwithstanding the logistical challenges in ensuring standardized sampling[43,44].

Floral resource availability in the interior was clearly insufficient to attract insect pollinators[45–48], and as they tend to have high mobility, they are less affected by distance to edge or even distance of isolation from the mainland[49–51], hence pollinator abundance tended to mirror the distribution of floral resources along edges. We were careful to

discriminate how these shifts in resource and abundance distributions might influence ecological trends in richness and network architecture, after first accounting for potentially spurious confounding influences of lower sample coverage[52] in small compared with large fragments[53] and interior sites versus edges. Few studies have taken this type of null model approach to discriminating true fragmentation effects from potential sampling artefacts. In our system, we found that observed floral resources and plant richness on islands were significantly lower than would be expected from simple sampling effects alone (compared with a random draw from the mainland reference communities; Extended Data Figs. 5–8), whereas the strong declines in observed pollinator abundance and richness from edge to interior and from large to small islands were not significantly different from null expectation based on equivalent sampling effort at mainland reference interiors and edges. This suggests that floral resources and plant richness are strongly negatively affected by fragmentation, but associated declines in pollinator abundance and richness are a passive (indirect) consequence of floral resource shifts, rather than pollinators being directly negatively affected by other mechanisms associated with declining patch area or edge effects, in this case.

The positive effect of habitat edges is also reflected in the way that habitat area effects on network architecture were strongly buffered by proximity to edge. Plant-pollinator networks in the interior of smaller islands had low network size, higher relative connectance among interaction partners[26,54–56], lower nestedness and higher network modularity. In fragmented habitats, it is typically thought that changes in network architecture are driven by a loss of specialist pollinators that visit only a limited number of plants and are more likely to become extinct than generalists[12,57]. Under this scenario, the high extinction risk of specialists potentially breaks the degree of interaction asymmetry in smaller networks, which is a key attribute of nestedness[58], and finally reduces nestedness and increases the modularity of island interior networks. In our study, however, edges were not dominated solely by generalist species, but also had many highly specialized species. Instead, the greater floral resource availability and larger size of plant-pollinator networks at forest edges appear to buffer the impact of forest area loss on relative connectance, and significantly increase nestedness, decrease modularity and increase network robustness to plant extinction. Naturally, an important caveat on inferring changes in network robustness analytically (from observed patterns of network architecture) is that it reflects a static view of network fragility, which does not allow for plasticity of behavioural responses, such as interaction rewiring in the face of resource extinction. Although this warrants further study, particularly from trait-based and phylogenetic perspectives on edge alteration of coevolved pollination syndromes, we would expect that any potential effects of interaction rewiring on network robustness should be greatest for generalist compared with specialist pollinators. But again, we did not find that edges predominantly harboured more generalist 'opportunistic' species in our study system. As in previous studies[32,34,59], increasing network nestedness in response to decreasing habitat area appears to be the key factor in mitigating secondary extinctions from species loss. It is increasingly appreciated that anthropogenic stressors can potentially offset one another, reducing secondary extinction risk[57]. Although our effect size for edge influence on network robustness is relatively small, the direction of effect we find in our forest system is actually the opposite of what is found in non-forest systems. Anthropogenic edge effects are typically considered to exacerbate the effects of forest area loss on pollinators, but here we show that forest edges not only dampen the response but also completely reverse the sign of the effect. In fragmented secondary forest, our research provides evidence that habitat edges have the potential to play a positive role.

It is worth pointing out in the SEM that all strongly supporting paths underpinning our main conclusions differed significantly in (absolute) effect size than would be expected from a null-draw SEM

constructed from the mainland reference community, while other components paths in cascading chains of indirect effects were not different than would be expected by chance. Even some paths with strong standardized effect sizes (up to 0.87 for the association between pollinator abundance and pollinator richness) were equivalent in direction and magnitude of effect to that predicted in the null-draw SEM, indicating that these community and network properties are inherently strongly covarying, even in unfragmented forest networks. Integrating a null SEM approach provides a simple and effective means of discriminating the causal paths most influenced by extrinsic variables of interest (such as forest area and proximity to edge, in our case) from spurious sample biases or incidental associations among variables.

We conclude that the open sunny conditions and high floral resources at anthropogenic forest edges in TIL enhance the abundance, richness, network architecture and persistence of plant-pollinator networks in regenerating forest remnants. Regenerating forest systems have come to dominate increasingly large portions of the globe[60,61], but in the early phases of secondary succession they lack the natural gap-phase dynamics of old-growth forest systems[18,19,21] on which many insect pollinators depend. From a conservation perspective then, there is an argument for recognizing the value of anthropogenic edges for plant-pollinator communities in small regenerating forest fragments. It is important to recognize we are not advocating that forest fragmentation is 'good' for biodiversity in a general sense[62], but that in the absence of effective restoration strategies to promote heterogeneity of forest gap structure and diversity of floral resources for pollinators, anthropogenic edges can have an unexpected beneficial role to play in network resilience[17], at least in the early decades of forest restoration.

## Methods

### Study area

The study was carried out in TIL, Zhejiang Province, eastern China (29°22′–29°50′ N, 118°34′–119°15′ E; Fig. 1). This large artificial reservoir was created by the construction of the Xin'anjiang Dam for hydroelectricity in 1959, resulting in the flooding of an area of approximately 573 km² at the high-water mark (108 m above sea level) and the creation of 1,078 islands (0.25–1,320 ha) out of former hilltops[38,63]. The main habitat type is unmanaged secondary forest (typical coverage ~90%) and the dominant plant species is *P. massoniana*[40]. The natural vegetation type on the islands is a mix of subtropical deciduous and coniferous forest, with many broad-leaved tree and shrub species, such as *Cyclobalanopsis glauca*, *Castanopsis sclerophylla*, *Smilax davidiana*, *Grewia biloba* and *Loropetalum chinense*. In the island interior, the understorey shrub and herb layers are relatively sparse and dominated by generalist shrubs *Loropetalum chinense*, *S. paniculata* and *S. sumuntia*, whereas at the island edge, the understory shrub and herb layers are denser and more diverse, simultaneously containing many specialist species as well as widespread generalists. The climate is typical of the subtropical monsoon zone and is highly seasonal. Median annual precipitation in this area is 1,430 mm, mainly concentrated in the rainy season between April and June. The average annual temperature is 17.0 °C, ranging from −7.6 °C to 41.8 °C[39,40].

### Sampling design

We conducted surveys of plant-pollinator interactions at paired edge versus interior transects on 41 islands and 16 mainland sites over 3 yr (Fig. 1). The islands were selected to encompass as much variation in forest area and distance of isolation from the mainland as possible. Mainland sites were selected to achieve maximum spatial coverage around the lake margin and similarity of vegetation types to those on the sampled islands. We established paired transect lines (that is, 100 × 4 m), with one along the edge (hereafter called the 'edge transect') and one extending perpendicular from the edge into the forest interior at each site (hereafter called the 'interior transect')[63]. The interior transects penetrated all the way into the centre of small islands (<10 ha)

and most of the way into the centre of even the largest islands (≥10 ha) because of the mountainous topography of the region and complex island shapes (Fig. 1). The average difference in elevation between edge transects and interior transects on each island varied from just 6.80 to 48.60 m across sites (Supplementary Table 1). In this study, we were interested in partitioning the effects of both island area and distance from forest edge, so we could not sample interior transects only at the discrete 'core' of each island because this would have created a confounding bias as interior distance from edge would intrinsically increase from small to large islands. We would also point out that any potential limitations of sampling 'edge to interior' transects (rather than strictly 'core' habitat) will only tend to make our findings more conservative because partial sampling of near-to-edge distances within our interior transects somewhat dampened differences that would be expected at greater distances into the interior.

Across the 41 islands, the number of pairs of transects varied from 1 to 16 and was roughly proportional to (ln-transformed) island size[53] (Supplementary Table 1). On islands with more than two paired transects, each paired transect was separated by ≥0.5 km. When we sampled, we used at least 4 paired edge versus interior transects on large islands (7 islands ≥10 ha), which allowed us to cover all four aspects (east, west, north, south), whereas there were 1, 2 or 3 paired transects randomly located on small islands (34 islands <10 ha; Supplementary Table 1), which could not completely cover edges on the four aspects. However, on the smallest islands, even a single edge transect wrapped around a reasonable portion of the island, allowing coverage of both the shady and sunny sides of each island to the greatest degree possible.

Along each transect, we were able to observe all flowers on herbaceous plants under 3.5 m from the ground, but only low individual flowering branches of shrubs and trees that were taller than 3.5 m. We walked each edge transect at a mean pace of 7 m min⁻¹ and each interior transect at a mean pace of 10 m min⁻¹ (considering the larger number of flowering plants at edges, we used a 15 min survey interval for each edge transect, but a 10 min survey interval for each interior transect). For the same level of survey effort, the likelihood of detecting all the flowers along the transect was higher in the interior (not lower), and the available observation time per flower was actually greater in the interior (because there were relatively few flowers). The observed results show that floral resources were approximately 18.52 times higher at the edge than in the interior, and pollinator abundance approximately 18.43 times higher at the edge. The 1.5 times difference in survey interval cannot account for the very large difference in observed detections. A detailed rationale of the difference in survey effort between edge and interior transects is provided in Supplementary Results 2, and any potential bias in survey effort is fully accounted for in statistical analyses. Observations were carried out only in calm and sunny weather from 8:30 to 12:00 and from 13:00 to 17:00. Flowering phenology differed among species, so we observed transects at multiple times throughout the season over 3 yr. We sampled all sites one time every two weeks on average, with 6 surveys conducted at each site from 20th April to 20th July in 2017, 7 surveys at each site from 23rd March to 14th July in 2018 and 7 surveys at each site from 13th March to 20th July in 2019 (that is, 20 surveys in total at each site)[63]. We randomized the sequence in which transects were sampled during the 2-week sampling periods to overcome potential differences in phenology within or between sites. These survey periods represent peak flowering and the peak of insect flower visitation, ensuring that our survey periods covered almost the entire flowering phenology of all dominant plant species in the TIL region.

We recorded an insect as a putative pollinator only if it was actually touching the anthers and/or stigmas of the flowers. For pollinators and plant species that could not be identified in the field, voucher specimens were hand-collected and identified in the laboratory. These identifications were later verified by specialists (see Acknowledgements). We also recorded the number of pollinators of each insect species visiting each flowering plant species during the sampling period. At the

 

same time, we estimated the flower resource availability of each plant along each transect at each sampling period and summed total flower resource availability together as an estimate of floral resources on the islands (see Supplementary Methods 1). The same sampling strategy was used at the 16 mainland sites, with sampling effort matching the number of surveys on the largest island (16 pairs of edge and interior transects). Lists of species and their recorded abundances are presented in Supplementary Tables 1–3. We also calculated the sampling completeness of plants, pollinators and their interactions on each island (see Supplementary Methods 2), as well as species composition on the mainland and 41 islands (see Supplementary Methods 3).

### Network metrics

Using the combined data from multiple plant-pollinator surveys per transect, we assembled quantitative interaction networks (with the 3 yr data combined) for each island using the edge and interior transects separately. That is, on each island, we first pooled the 20 surveys of each transect over 3 yr, then assembled two separate interaction networks for the edge and interior networks of each island (or mainland) location. Pooling across the 3 yr was necessary to obtain relatively complete network data on each island (Supplementary Table 1 and Extended Data Fig. 1), covering the full phenological period of interaction of plants and pollinators while reducing noise due to stochastic artefacts of adverse local conditions on particular days. The availability of relatively complete network data is an important premise for testing the effects of forest fragmentation on plant-pollination interaction networks across islands. For the mainland network, quantitative interaction networks were constructed in the same way as each island network for the 16 edge transects separately from the 16 interior transects, which represented the mainland ('unfragmented') reference network in null model analyses. For each of the observed island networks, we calculated four metrics of network architecture: relative connectance, nestedness, modularity and robustness.

Connectance, defined as the fraction of observed interactions relative to the total possible interactions within a network[54], is considered as a primary attribute of the network. However, the definition of connectance always overestimates the potentially realizable interactions (that is, ignoring 'forbidden links' in an evolutionary or ecological context), especially for large networks that contain a large number of plants and pollinators that are never likely to interact with one another, and which can cause connectance to be insensitive to changes in realized interactions[26,56]. Here we propose a different measure of 'relative connectance' to more reasonably constrain the number of potential interactions. Relative connectance is defined as $RC = I/S_{obs}$, the ratio of the number of observed interactions among species recorded in the network ($I$) to the total observed number of interactions of these species across all observation sites over the 3 yr study period ($S_{obs}$). The accuracy of relative connectance depends on the level of sampling completeness and the degree to which pooling of the sampled networks adequately represents the true regional network as a whole. Therefore, long-term observation data are necessary when using this parameter. When the $S_{obs}$ value in RC is based on long-term observations, it will more accurately reflect the actual degree of connectance within a network. The ecological interpretation of RC is the fraction of potentially realizable interactions across all observation sites, and values range between 0 (no interactions realized) and 1 (all interactions realized).

Nestedness and modularity are relevant to robustness of the network[64]. Nestedness is a pattern of specialist species interacting only with proper subsets of the species that interact with more generalist species[65]. We used the NODF index to represent nestedness, this index being one of the most popular methods based on overlap and decreasing fills[66]. Specifically, overlap denotes complete overlap of the presences (observed interactions) from lower to upper rows and from right to left columns, and decreasing fills denotes decreasing marginal totals between all pairs of columns and all pairs of rows[67]. Modularity is

the division of a network into compartments or 'modules' composed of species that more strongly interact with one another than with species from other modules[68]. We used the 'DIRTLPAwb+' algorithm to maximize a measure of modularity[69], that is, each node is randomly assigned a unique label, then at each iterating step, each node is forced to assume the label that is shared by the neighbours connected to each node within the graph. The algorithm then partitions the nodes of the network into separate subsets or modules, and finally, modularity is maximized whenever each of these modules is relatively isolated from the other modules in the network. We calculated NODF using the R package 'vegan' v.2.5–7[70], and modularity using the R package 'bipartite' v.2.16[71].

Robustness denotes the resilience of plant-pollinator networks to secondary extinctions of pollinators following the sequential loss of plants[59,72] under the assumption that plant-pollinator interactions are strongly driven by bottom-up effects[73,74]. We ascribed the likely extinction order of plants on the basis of their rarity (the area of flower resources) in transect surveys (Supplementary Table 2), as rare plants are susceptible to local extinction on islands. We chose this relatively simple but clear assumption as the basis for interpreting 'robustness' as a relative index of vulnerability to species loss, in lieu of any available empirical data on 'true' extinction sequence. We calculated robustness using the R package bipartite v.2.16[71].

### SEM

We used 'piecewise' SEM to partition the direct vs indirect effects of forest area loss and fragmentation (decreased area, proximity to edge and increased isolation) on plant-pollinator community structure (floral resources, plant richness, pollinator abundance and pollinator richness) and network architecture (relative connectance, nestedness, modularity and robustness). The hypothesized causal logic underpinning each path is presented in Supplementary Methods 4. The piecewise SEM consists of a series of separate linear models with local rather than global estimation of parameters and combines these into a single directed acyclic graph[75], which is particularly suited to hierarchical nested data structures and non-normal error distributions in models. Moreover, local estimation allows greater robustness in fitting smaller data sets[76], and we follow the recommendation of Grace et al.[77] in ensuring that we have more than five samples per variable estimated in the model. We tested the causal structure of the hypothesized model (Fig. 2) using 'piecewiseSEM' v.2.1.0, which extends SEM to non-normal distribution models[76]. Specifically, models testing the direct and indirect effects of fragmentation on plant richness, pollinator richness and abundance used Poisson generalized linear mixed-effects models (GLMMs) in 'lme4' v.1.1–23[78], while fragmentation effects on floral resources and the four network attributes were tested with linear mixed-effects models (LMMs). We used ln-transformation of forest area and distance of isolation to linearize relationships. Models contained a random effect for island identity to account for non-independence of paired edge versus interior transects sampled within each island. Overall model fit was tested using Shipley's d-separation test via a Fisher's $C$ statistic and $\chi^2$-based $P$ value[75,79]. We selected a 'final' SEM by sequentially removing model predictors (direct paths) with the lowest AIC value until all remaining paths were significant and the 'global' SEM $P$ value was non-significant (that is, no remaining 'missing' paths). Direct, indirect and total effects for the SEM were calculated using the 'semEff' package v.0.6.0[80], with effect sizes adjusted for multicollinearity among predictors[81]. The 95% confidence interval for effects was calculated using 1,000 bootstrapped estimates for each response. Model-predicted total effects are presented using partial regression coefficients calculated using the 'predEff' function in the 'semEff' package.

### Null model for community structure and network architecture

Plant-pollinator community structure and network architecture on islands might differ from the mainland reference sites both as a result of

fragmentation processes and of biases in sampling effort. The purpose of the null model is to determine whether variation in plant-pollinator community structure and network architecture on islands is significantly greater (or less) than expected from a simple 'passive sampling effect' from the continuous mainland reference pool (https://doi.org/10.6084/m9.figshare.20477889). In selecting the reference pool, we follow the principle of Gotelli and Graves (1996)[82] that "A null model is a pattern-generating model that is based on randomization of ecological data..[…].. designed to produce a pattern that would be expected in the absence of a particular ecological mechanism". In our case, we are interested in the null pattern of community diversity and network structure that would be expected in the absence of fragmentation and reduction in island area. Given that we do not have 'pre-fragmentation' data to directly test re-assembly trajectories through time on each island, the combined set of sampling plots from the adjacent 'unfragmented' mainland is the most appropriate reference pool available for the null draws. To do this, we compiled plant-pollinator interaction data from all sampling transects on the mainland edge and interior, respectively, then used these as our expected 'reference' pools. From edge and interior reference pools (separately), we used two methods to simulate 'null communities' and 'null networks': (1) Null model I: a random draw (with replacement) of the same number of transects from the mainland as observed on each of the 41 sampled islands (that is, constraining the number of sampling transects used to acquire a null estimate of floral resources, plant richness, pollinator richness and pollinator abundance); (2) Null model II: a random draw (with replacement) of the same number of pairwise interactions, while ensuring the same numbers of plant and pollinator species were selected as those observed on each of the 41 sampled islands (that is, constraining both network abundance and network size) (https://doi.org/10.6084/m9.figshare.20477889). The concepts, step-by-step procedures and R code for the null-draw methods are presented in Supplementary Methods 5 and Code availability.

We calculated the standardized effect size, that is, SES = ($\alpha_{obs}$ − $\alpha_{null}$) / s.d.($\alpha_{null}$) as a measure of the magnitude and direction of the difference between the observed and the null values for each island[83]. A positive or negative value of SES indicates that the observed value is above or below the mean of the null distribution, respectively. We used approximate statistical significance at the 5% level for a two-tailed test when estimating significance. If the observed values differed significantly from the null model values, this indicates that the observed values showed non-random assembly trajectories as a result of forest area loss and fragmentation, over and above any stochastic effects due to confounding bias in sampling effort between islands. In contrast, if the observed values were not significantly different from the null model, variation in diversity or network structure was considered indistinguishable from the stochastic biases that might be expected from passive sampling effects.

**Null model SEM**

After using the observed data to construct the SEM (observed SEM) and acquiring the null model simulation results for community structure and network architecture, we further wanted to evaluate whether the observed causal relationships (paths) deviated significantly from what might be expected under a passive sampling effect from the continuous mainland reference sites. Therefore, we ran the same SEM path structure using 1,000 randomly drawn null model values for community structure and network architecture (as described above) to construct 1,000 SEM models and evaluate the significance of each observed path against the mean expected path coefficients (±95% CI) from the 1,000 constructed null model SEMs. All statistical analyses were conducted using R v.4.0.1[84].

**Reporting summary**

Further information on research design is available in the Nature Portfolio Reporting Summary linked to this article.

**Data availability**

The dataset used in our analyses can be found in the Figshare repository (https://doi.org/10.6084/m9.figshare.20477889).

**Code availability**

The code for null models is available from the Figshare repository with the following download link: https://doi.org/10.6084/m9.figshare.20477889.

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

## Acknowledgements
We thank the Xin'an River Ecological Development Group Corporation and Forestry Bureau of Chun'an County for permits necessary to conduct research at the Thousand Island Lake; Q. Jiang, M. Zhang and other lab group members for field assistance; M. C. Wilson for map construction; Z. Xu at Zhejiang A&F University for the identification of insect species; S. Yao for the identification of plant species; and X. Xu, Y. Han and X. Zhang for help with data analysis. This study was funded by the National Natural Science Foundation of China (Grant/Award number 32030066 to P.D., 31872210 and 32071545 to X.S., and 31930073 to Mingjian Yu), the Natural Science Foundation of Zhejiang Province (#LD21C030002 to P.D.), the China Scholarship Council (Grant number 201906320342 to P.R.), the Shanghai Rising-Star Program (#19QA1403300 to X.S.) and the Program for Professor of Special Appointment (Eastern Scholar) (#TP2020016 to X.S.).

## Author contributions
P.R. and P.D. conceived the idea and designed the study, with support from R.K.D.; P.R. and R.K.D. developed and implemented null model methods; D.Z. and M.V.M. helped write statistical model analysis code; P.R., X.S. and P.D. conducted the analyses; P.R. and R.K.D. led the writing, with substantial contributions from P.D., X.S., M.V.M. and D.Z. All authors provided input on manuscript revisions.

## Competing interests
The authors declare no competing interests.

## Additional information
**Extended data** is available for this paper at https://doi.org/10.1038/s41559-022-01973-y.

**Correspondence and requests for materials** should be addressed to Ping Ding.

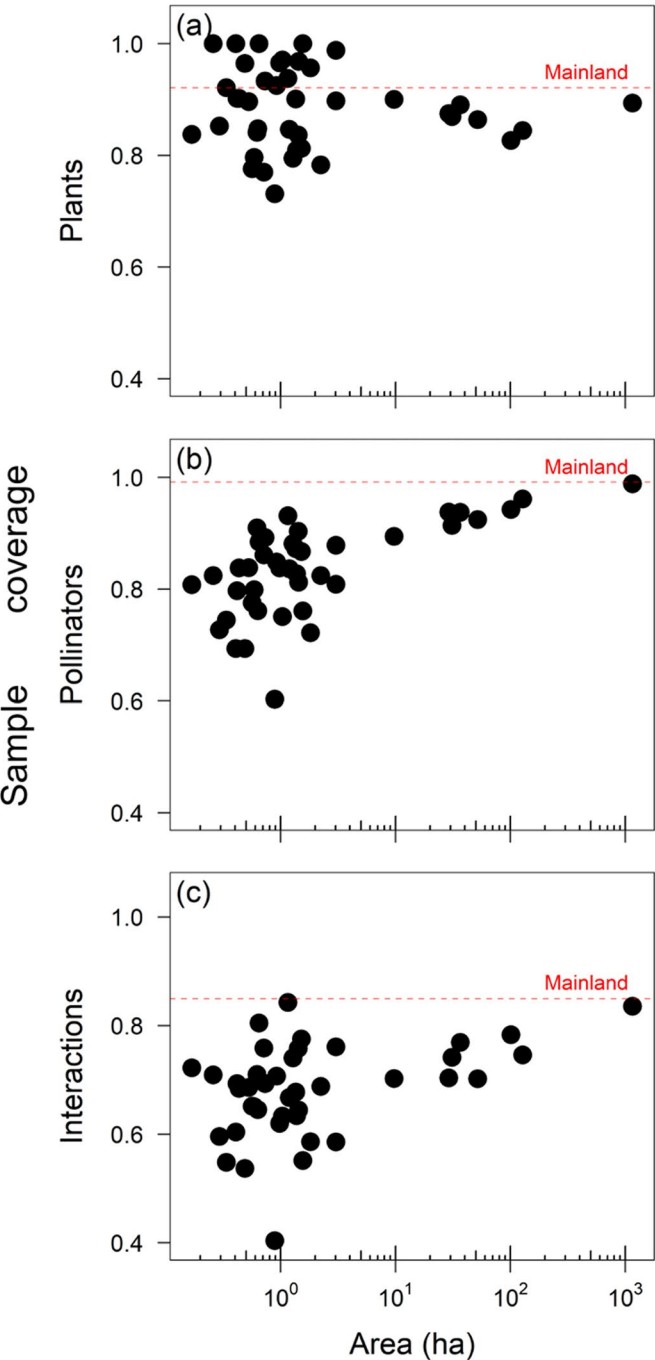

**Extended Data Fig. 1 | Estimated sample coverage values for plants, pollinators and their interactions after 20 surveys per site over 3 years.** Black symbols are for the 41 islands and the dashed red line represents the 16 mainland reference sites (combined).

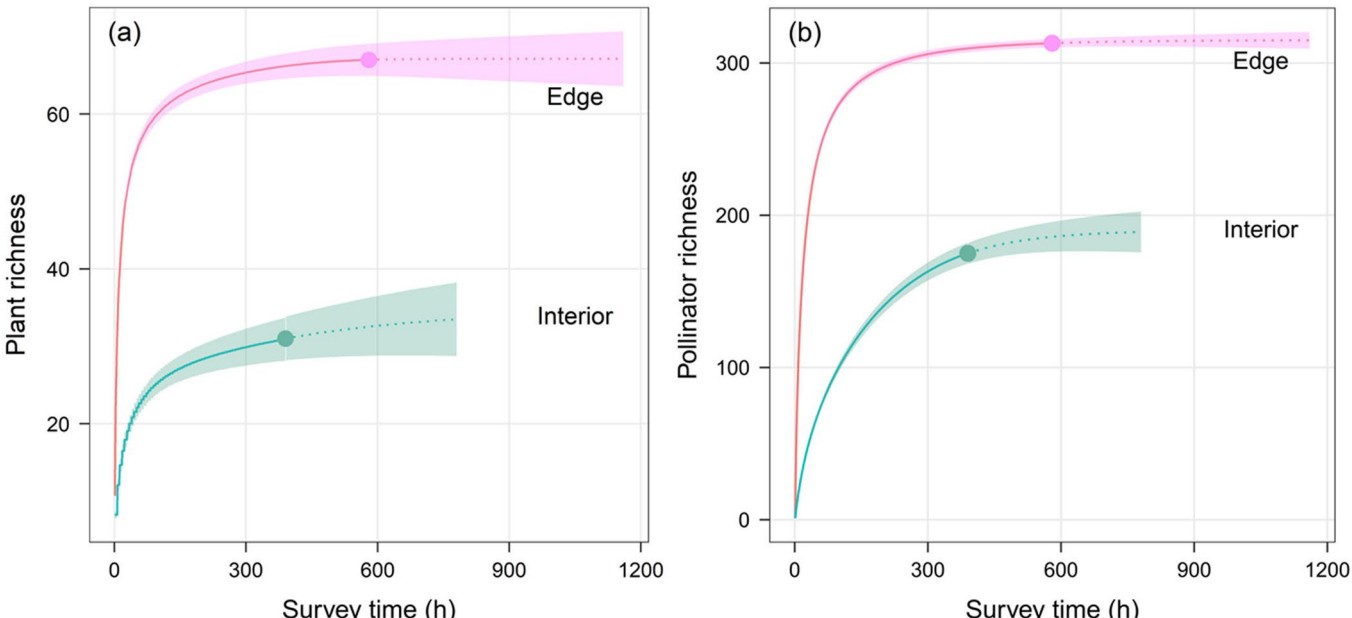

**Extended Data Fig. 2 | Sensitivity test of differing transect survey times at interior versus edge sites.** Survey-time standardized interpolation (solid line segments) and extrapolation (dotted line segments, n = 50 resamples) of estimated species richness (shaded regions represent the 95% confidence interval, that is, mean ± CI) are completely non-overlapping between edges and interiors of the 41 islands, for both: (a) plant species and (b) pollinator species. The dotted lines denote the extrapolated sample accumulation curves when the sampling intensity is twice the actual sampling intensity. The filled circles represent the total observed plant/pollinator richness.

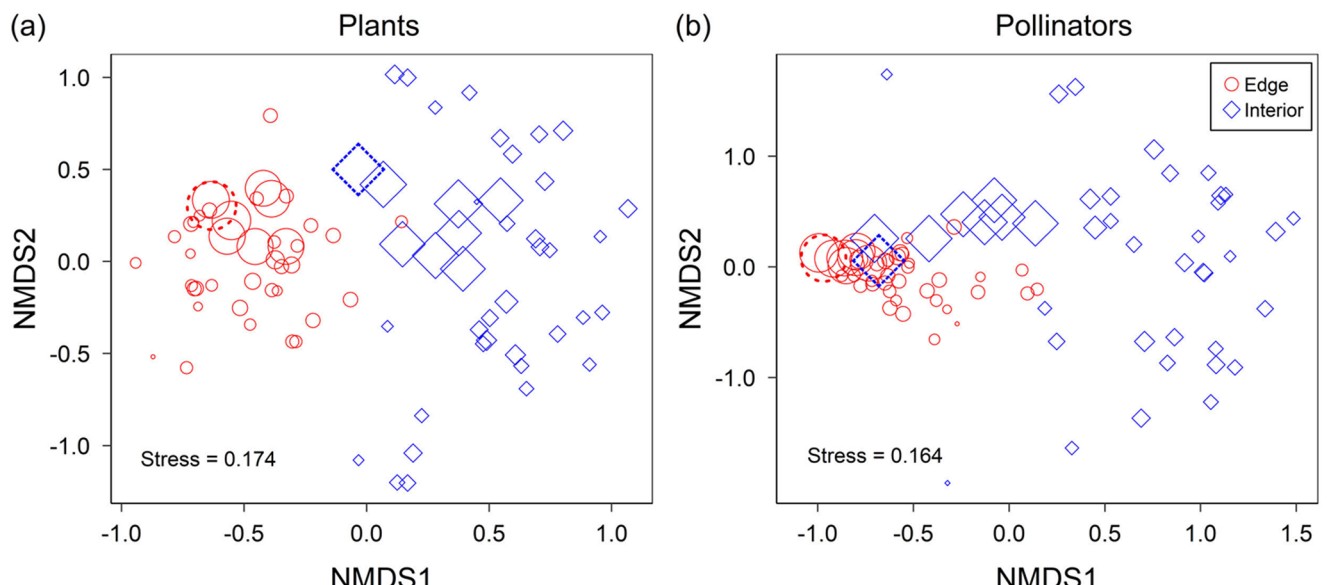

**Extended Data Fig. 3 | Non-metric multidimensional scaling (NMDS) ordination diagrams based on three dimensions for (a) plants and (b) pollinators.** Red circles with solid outlines represent island edge sites and the blue diamonds with solid outlines represent island interior sites, respectively. Red circles and blue diamonds with dashed outlines represent the mainland edge and mainland interior sites, respectively. Symbol size is proportional to the island area.

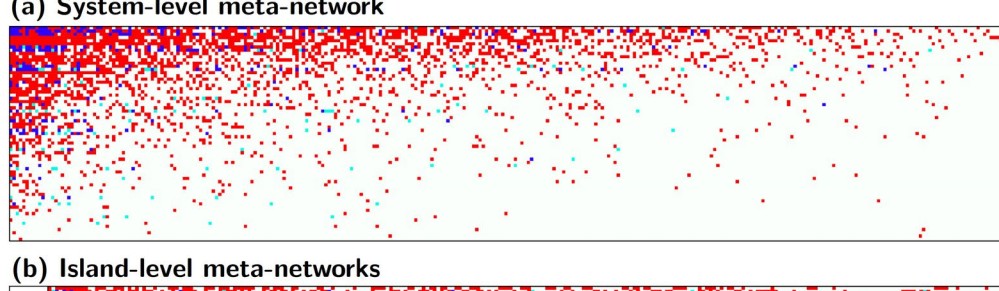

**Extended Data Fig. 4 | Network nestedness for (a) the system-level meta-network combining all edge and interior networks of all 41 islands, and (b) 41 island-level meta-networks combining the edge network and the interior network of each island.** Rows represent flowering plant species, columns represent insect pollinator species. Species towards the upper left of each meta-network are increasingly 'generalist' in their species interactions (see inset diagram to the right of island meta-network B02). Red boxes represent interactions occurring only at the island edge, while dark blue boxes represent interactions occurring only in the island interior, and light blue boxes represent interactions occurring at both the edge and interior. The size of the checkerboard is proportional to the size of the network.

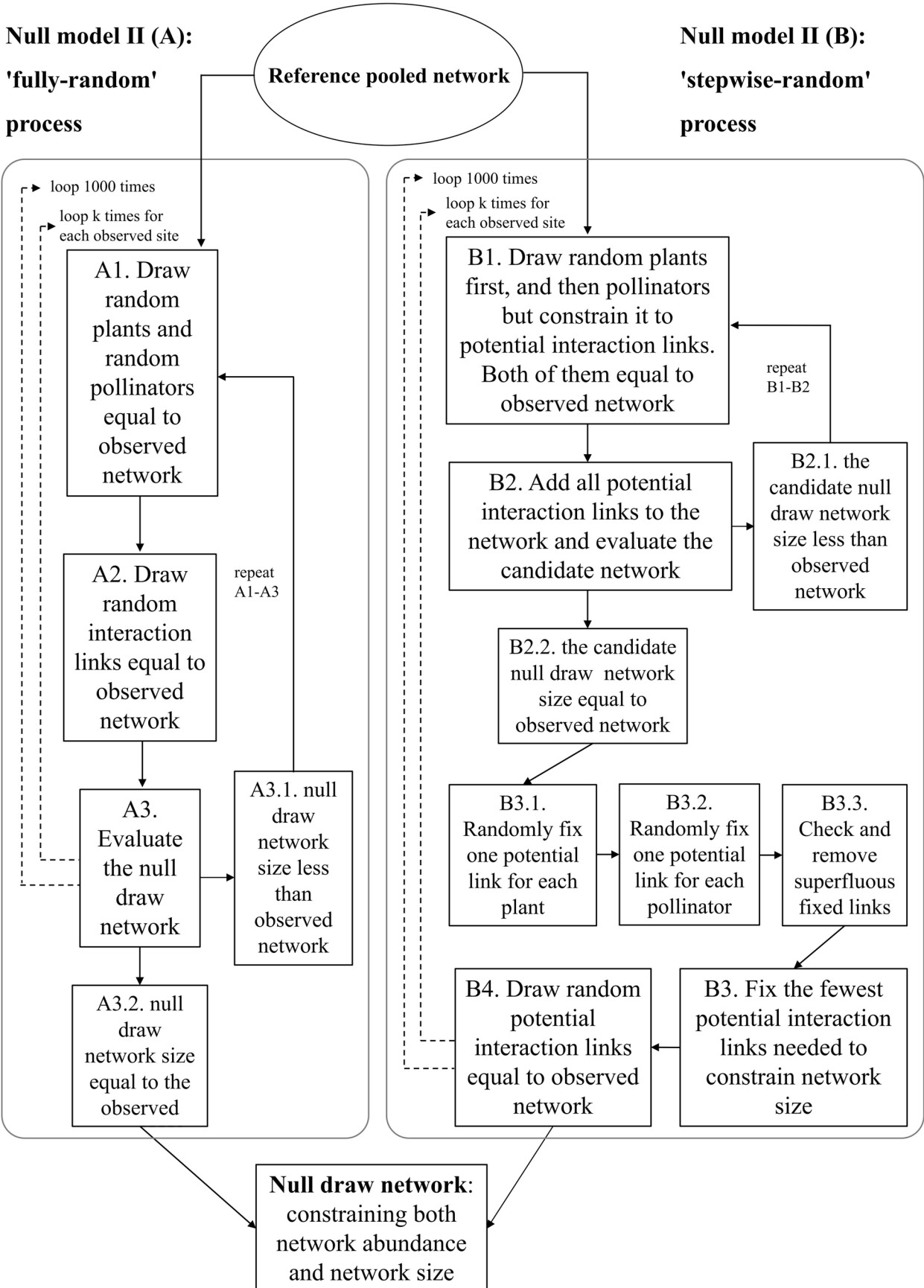

**Extended Data Fig. 5 | Schematic representation of the main steps in the null draw process for null model II to generate 1000 random networks from the mainland reference pool.** The goal of this approach is to constrain both network abundance and network size, (**a**) using a simple (but slow) fully-random draw process, and (**b**) using a more complex (but substantially faster) stepwise-random process. See Extended Data Figs. 7, 8 for further details.

| The observed network in the interior of island S20 | | | | | | |
|---|---|---|---|---|---|---|
| **S20** | PO144 | PO84 | PO214 | PO161 | PO5 | PO73 | PO3 |
| PL64 | 1 | 1 | 0 | 0 | 0 | 0 | 0 |
| PL3 | 0 | 0 | 2 | 0 | 0 | 0 | 0 |
| PL42 | 1 | 0 | 0 | 1 | 0 | 0 | 0 |
| PL18 | 0 | 0 | 0 | 0 | 1 | 0 | 0 |
| PL65 | 2 | 0 | 0 | 0 | 0 | 1 | 1 |

**Extended Data Fig. 6 | The observed plant-pollinator network in the interior of island S20.** Island S20 had one of the simplest network structures, containing only 5 plant species (PL), 7 pollinator species (PO), and a network abundance of 11.

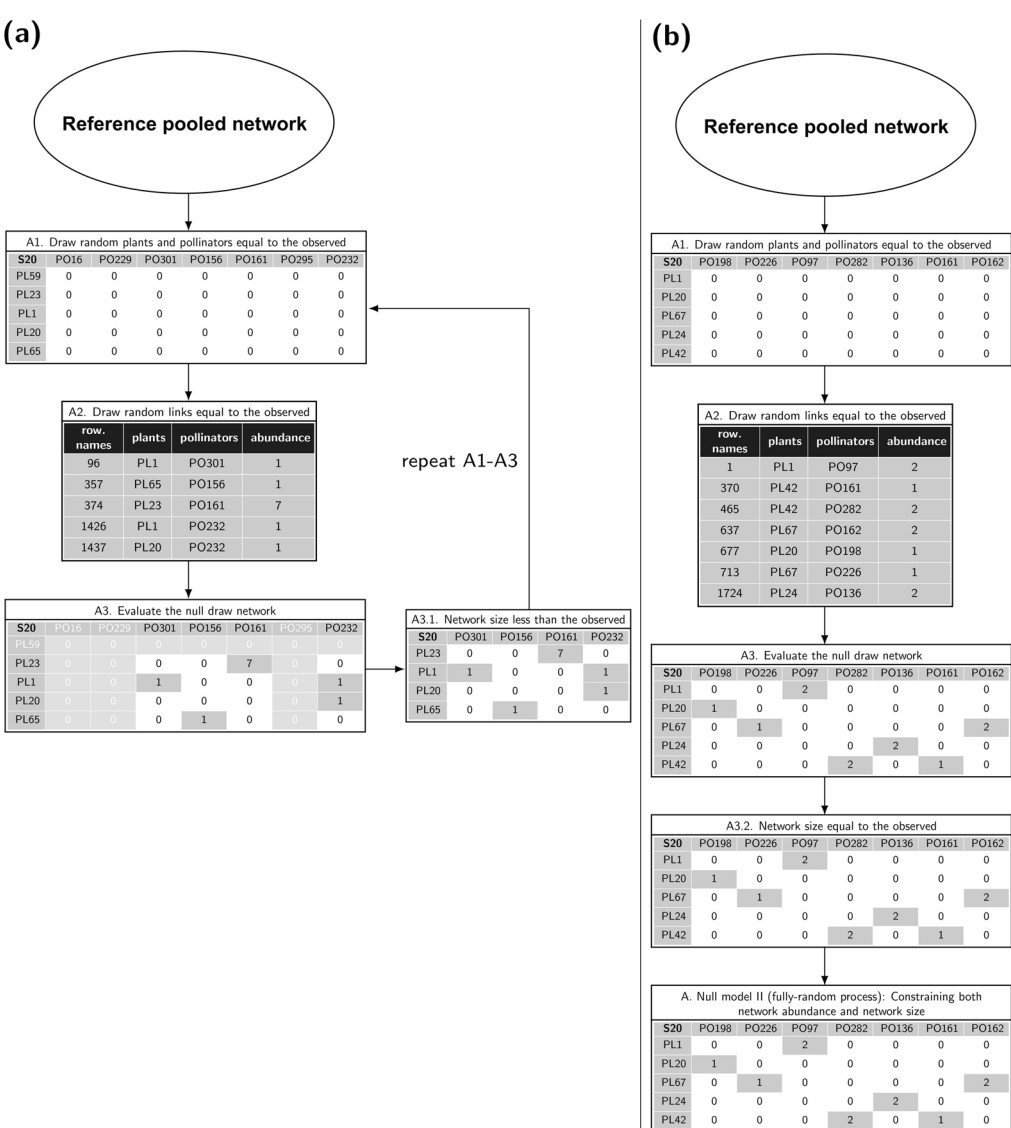

**Extended Data Fig. 7 | An illustration of the process followed in null model II using the fully-random process (Extended Data Fig. 5a) to produce one random network constraining network abundance and network size to be the same as that observed at the interior site on island S20 (see Extended Data Fig. 6).** (**a**) One of the repeated processes to randomly draw out one expected network, in this case with network size less than the observed S20. (**b**)

One successful process to randomly draw out an acceptable network with the same abundance and size as the observed S20 (low efficiency: one successful loop takes an average of 29.7 seconds, and as network size increased, the time needed dramatically increased). The 'reference pooled network' is the mainland interior network.

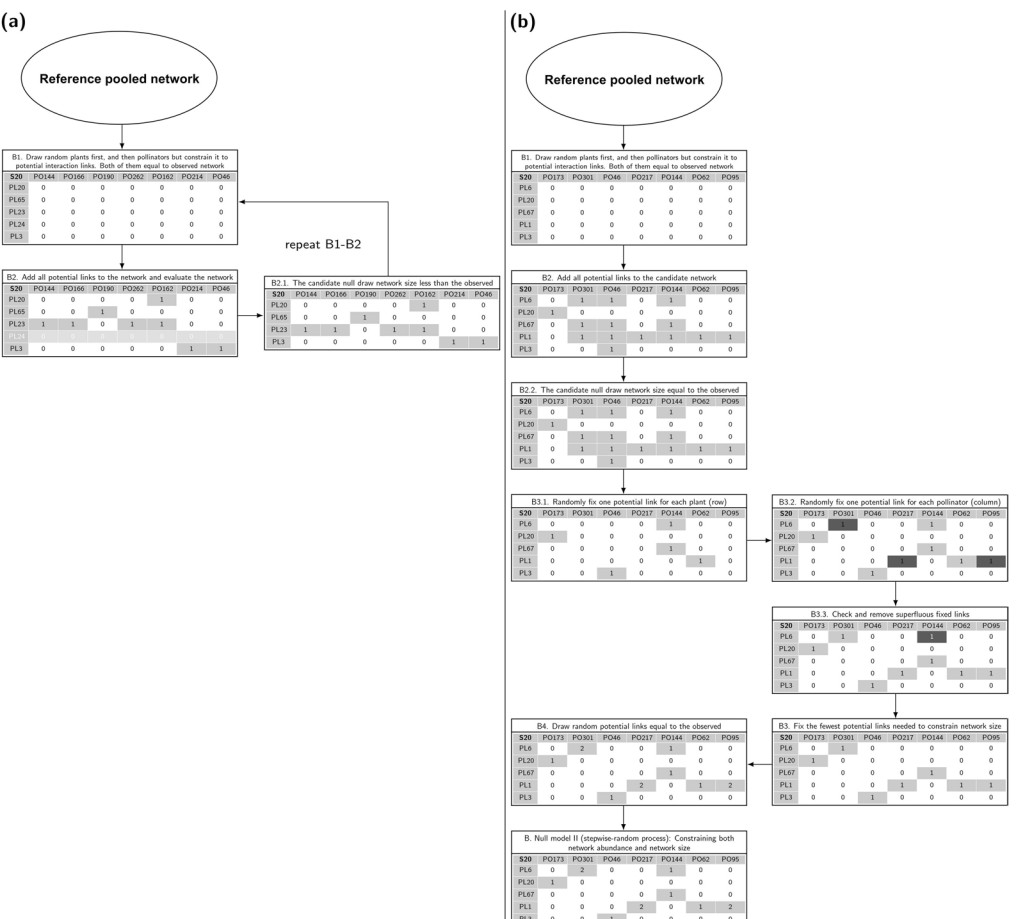

**Extended Data Fig. 8 | An illustration of the process followed in null model II using the stepwise-random process (Extended Data Fig. 5b) to produce one random network constraining network abundance and network size to be the same as that observed at the interior site on island S20 (see Extended Data Fig. 6).** (**a**) One of the repeated processes to randomly draw out one candidate network, in this case with network size less than the observed S20. (**b**) One successful process to randomly draw out an acceptable network of the same abundance and size as the observed S20 (high efficiency: one successful loop takes an average of 0.12 seconds, and remains efficient as network size increases). The 'reference pooled network' is the mainland interior network.

# Reporting Summary

## Statistics

For all statistical analyses, confirm that the following items are present in the figure legend, table legend, main text, or Methods section.

| n/a | Confirmed | |
|---|---|---|
| ☐ | ☒ | The exact sample size (*n*) for each experimental group/condition, given as a discrete number and unit of measurement |
| ☐ | ☒ | A statement on whether measurements were taken from distinct samples or whether the same sample was measured repeatedly |
| ☐ | ☒ | The statistical test(s) used AND whether they are one- or two-sided<br>*Only common tests should be described solely by name; describe more complex techniques in the Methods section.* |
| ☐ | ☒ | A description of all covariates tested |
| ☐ | ☒ | A description of any assumptions or corrections, such as tests of normality and adjustment for multiple comparisons |
| ☐ | ☒ | A full description of the statistical parameters including central tendency (e.g. means) or other basic estimates (e.g. regression coefficient) AND variation (e.g. standard deviation) or associated estimates of uncertainty (e.g. confidence intervals) |
| ☐ | ☒ | For null hypothesis testing, the test statistic (e.g. $F$, $t$, $r$) with confidence intervals, effect sizes, degrees of freedom and $P$ value noted<br>*Give P values as exact values whenever suitable.* |
| ☒ | ☐ | For Bayesian analysis, information on the choice of priors and Markov chain Monte Carlo settings |
| ☐ | ☒ | For hierarchical and complex designs, identification of the appropriate level for tests and full reporting of outcomes |
| ☐ | ☒ | Estimates of effect sizes (e.g. Cohen's *d*, Pearson's *r*), indicating how they were calculated |

*Our web collection on statistics for biologists contains articles on many of the points above.*

## Software and code

Policy information about availability of computer code

| Data collection | No software was used for data collection. |
|---|---|
| Data analysis | All analyses were performed with the open source R (version 4.0.1). We used the bipartite (version 2.16) and vegan (version 2.5-7) packages for plant-pollinator network architecture analysis and the piecewiseSEM package (version 2.1.0) for piecewise structural equation models. Direct, indirect, and total effects from piecewiseSEM were calculated using the semEff package (version 0.6.0). Null models for plant-pollinator community structure and network architecture were generated using our own code written in R. Plots were generated with R (version 4.0.1) and ggplot2 (version 3.3.5) or in LaTeX (version 2.9.2). |

For manuscripts utilizing custom algorithms or software that are central to the research but not yet described in published literature, software must be made available to editors and reviewers. We strongly encourage code deposition in a community repository (e.g. GitHub). See the Nature Portfolio guidelines for submitting code & software for further information.

## Data

Policy information about availability of data

All manuscripts must include a data availability statement. This statement should provide the following information, where applicable:

- Accession codes, unique identifiers, or web links for publicly available datasets
- A description of any restrictions on data availability
- For clinical datasets or third party data, please ensure that the statement adheres to our policy

Scripts for null model generation are available in figshare at https://doi.org/10.6084/m9.figshare.20477889
All raw data and processed code will be made available upon reasonable request.

## Human research participants

Policy information about studies involving human research participants and Sex and Gender in Research.

| | |
|---|---|
| Reporting on sex and gender | This study did not involve human participants. |
| Population characteristics | This study did not involve human participants. |
| Recruitment | This study did not involve human participants. |
| Ethics oversight | This study did not involve human participants. |

Note that full information on the approval of the study protocol must also be provided in the manuscript.

# Field-specific reporting

Please select the one below that is the best fit for your research. If you are not sure, read the appropriate sections before making your selection.

☐ Life sciences ☐ Behavioural & social sciences ☒ Ecological, evolutionary & environmental sciences

For a reference copy of the document with all sections, see nature.com/documents/nr-reporting-summary-flat.pdf

# Ecological, evolutionary & environmental sciences study design

All studies must disclose on these points even when the disclosure is negative.

| | |
|---|---|
| Study description | We conducted surveys of plant-pollinator interactions at paired edge versus interior sites on 41 islands and 16 mainland sites over 3 years. Mainland sites were selected to achieve maximum spatial coverage around the lake margin, and similarity of vegetation types to those on the sampled islands. We established paired transect lines (100 × 4 m), with one along the edge and one extending perpendicular from the edge into forest interior at each site. On islands, the number of pairs of transects varied from 1 to 16, and was roughly proportional to (ln-transformed) island size. On islands with more than two pairs of transects, each pair was separated by ≥ 0.5 km. |
| Research sample | Our research samples consisted of plant and pollinators species, as well as their interactions. |
| Sampling strategy | Along each transect, we observed individual flowering branches of shrubs and trees, or the whole plant in the case of herbaceous plants under 3.5 m from the ground. We walked each edge transect at a mean pace of 7m / min and each interior transect at mean pace of 10m / min (due to the larger number of flowering plants at edges we used a 15-minute survey interval for edge transects, but a 10-minute survey interval for interior transects). |
| Data collection | Observations were carried out only in calm and sunny weather, from 8:30 AM to 12:00 noon and from 1:00 PM to 5:00 PM. We considered an insect to be a putative pollinator only if it was touching the anthers and/or stigmas of the flowers. |
| Timing and spatial scale | We observed transects at multiple times throughout the season on 41 islands and 16 mainland sites over 3 years. We sampled once every two weeks, on average, with six surveys conducted at each site from 20th April to 20th July in 2017, seven surveys at each site from 23rd March to 14th July in 2018 and seven surveys at each site from 13th March to 20th July in 2019 (i.e., 20 surveys in total at each site). |
| Data exclusions | No data were excluded from the analysis. |
| Reproducibility | All data necessary to repeat the analyses will be made publicly available. |
| Randomization | Transect selection was stratified across dominant spatial and temporal gradients of variation, in order to standardise sample collection. |

Null model analysis used a fully-random and stratified-random procedures to draw null samples from the mainland reference pool of plant-pollinator interactions.

Blinding | Not applicable to this study.

Did the study involve field work? ☒ Yes ☐ No

## Field work, collection and transport

Field conditions | The main habitat type is unmanaged secondary forest (typical coverage ~90%) and the dominant plant species is Pinus massoniana. The climate is typical of the subtropical monsoon zone and is highly seasonal. Median annual precipitation in this area is 1430 mm, mainly concentrated in the rainy season between April and June. The average annual temperature is 17.0°C, ranging from -7.6 °C to 41.8 °C.

Location | The study was carried out in the Thousand Island Lake, Zhejiang Province, eastern China (29°22"–29°50" N, 118°34"–119°15" E). This large artificial reservoir was created in 1959 by the construction of the Xin'anjiang Dam for hydroelectricity.

Access & import/export | The research was carried out under the Xin'an River Ecological Development Group Corporation and Forestry Bureau of Chun'an County permit.

Disturbance | No disturbance was caused in the study sites.

# Reporting for specific materials, systems and methods

We require information from authors about some types of materials, experimental systems and methods used in many studies. Here, indicate whether each material, system or method listed is relevant to your study. If you are not sure if a list item applies to your research, read the appropriate section before selecting a response.

## Materials & experimental systems

| n/a | Involved in the study |
|-----|----------------------|
| ☒ ☐ | Antibodies |
| ☒ ☐ | Eukaryotic cell lines |
| ☒ ☐ | Palaeontology and archaeology |
| ☒ ☐ | Animals and other organisms |
| ☒ ☐ | Clinical data |
| ☒ ☐ | Dual use research of concern |

## Methods

| n/a | Involved in the study |
|-----|----------------------|
| ☒ ☐ | ChIP-seq |
| ☒ ☐ | Flow cytometry |
| ☒ ☐ | MRI-based neuroimaging |

