## [Peer Review File · Nature Ecology & Evolution]

Peer Review Information

Journal: Nature Ecology & Evolution

Manuscript Title: Forest edges increase pollinator network robustness to extinction with declining area

Corresponding author name(s): Ping Ding

Editorial Notes:

Reviewer Comments & Decisions:

Decision Letter, initial version:

25th March 2022

Dear Professor Ding,

Your Article, "Edges increase pollinator network robustness to extinction with declining forest area" has now been seen by 3 reviewers. You will see from their comments copied below that while they find your work of considerable potential interest, they have raised quite substantial concerns that must be addressed. In light of these comments, we cannot accept the manuscript for publication, but would be very interested in considering a revised version that addresses these serious concerns.

We hope you will find the reviewers' comments useful as you decide how to proceed. If you wish to submit a substantially revised manuscript, please bear in mind that we will be reluctant to approach the reviewers again in the absence of major revisions.

In particular, although Referees #1 and #3 was positive about the work, Referee #2 raises a major concern over the validity of the sampling regime, and whether it is truly comparing edges with interiors. They also had concerns about the rate of sampling between edges and interiors, and whether these were matched adequately. Both Referee #2 and Referee #3 would also like to see some consideration or discussion about the relative impacts on generalist vs specialist species.

If you choose to revise your manuscript taking into account all reviewer and editor comments, please highlight all changes in the manuscript text file [OPTIONAL: in Microsoft Word format]. In particular, for resubmission to Nature Ecology and Evolution you will need to provide a robust justification of the validity of the sampling, and if possible some evaluation of its sensitivity in capturing edges vs interiors.

* Include a "Response to reviewers" document detailing, point-by-point, how you addressed each referee comment. If no action was taken to address a point, you must provide a compelling argument.

2This response will be sent back to the referees along with the revised manuscript.

* If you have not done so already we suggest that you begin to revise your manuscript so that it conforms to our Article format instructions at <http://www.nature.com/natecolevol/info/final-submission>. Refer also to any guidelines provided in this letter.

[REDACTED]

If you wish to submit a suitably revised manuscript we would hope to receive it within 6 months. If you cannot send it within this time, please let us know. We will be happy to consider your revision so long as nothing similar has been accepted for publication at Nature Ecology & Evolution or published elsewhere.

Nature Ecology & Evolution is committed to improving transparency in authorship. As part of our efforts in this direction, we are now requesting that all authors identified as 'corresponding author' on published papers create and link their Open Researcher and Contributor Identifier (ORCID) with their account on the Manuscript Tracking System (MTS), prior to acceptance. This applies to primary research papers only. ORCID helps the scientific community achieve unambiguous attribution of all scholarly contributions. You can create and link your ORCID from the home page of the MTS by clicking on 'Modify my Springer Nature account'. For more information please visit please visit www.springernature.com/orcid.

Thank you for the opportunity to review your work.

[REDACTED]

Reviewer expertise:

Reviewer #1: Pollinators, networks

Reviewer #2: Edge effects

2Reviewer #3: Pollination, landscape ecology, networks

Reviewers' comments:

Reviewer #1 (Remarks to the Author):

The manuscript by Pen Reng et al "Forest edges increase pollinator network robustness to extinction with declining forest area" is an important piece of work and very well worked out.

I suggest to publish this work after minor revisions as high priority.

Cover of dense and diverse forests are declining as do pollinator diversity. Compared to other insect taxa, the majority of pollinators prefer open habitats and not dense, shady forests, which makes it interesting to study the role of forest ecosystems for pollinator diversity and community complexity using bipartite ecological networks. This study is one of the very few available studies showing how forest edges shape pollinator networks pointing to the importance of these ecotones for pollinating insects. The study is well designed with a high number of replicates (although it is not justified why data are pooled) and well analysed. In general, results are clearly presented and discussed. I only have one major point to be considered by the authors. I am sceptical about the generalisation of the results for the habitat fragmentation processes in general. I do not think you can transfer the findings to other ecotones or if you think you can do that, so please discuss this carefully. I suggest revising the paper and clearly focus on forest fragmentation and not habitat fragmentation in general. Currently, you mix arguments and papers of habitat fragmentation/edges with the specific habitat of forests. I would clearly separate this. For example if you fragment an extensive meadow with lots of native flowering plants to intensive agricultural monocultures e.g. maize, will you come to the same conclusions? This is not well worked out. Beside this, I am attaching more detailed comments and suggestions directly in the manuscript. I very much like how carefully you analysed your data. So good luck for a fast publication.

Reviewer #2 (Remarks to the Author):

Dear Editor,

Regarding the manuscript "Edges increase pollinator network robustness to extinction with declining forest area":

This is an interesting study, based on extensive field sampling, showing how island size and edges affect pollinator network. It has the surprising conclusion that edges buffer the pollinator networks against extinction, by showing, among other things, that edges have greater plant and pollinator richness and network nestedness. The manuscript is well written and the methods are explained in sufficient detail.

3However, I have several concerns related to the methods and the interpretation of the results, and I think that these issues must be addressed prior to publication:

- If I understood correctly, the "interior" transects were not located in the interior, but instead were placed perpendicular to the edge, thus sampling from the edge to a maximum distance of 100 m. I don't think that this is a valid method of sampling forest interior.
- Interior transects were sampled at a faster pace than edge transects. I think that this may have resulted in lower detection of flowering plants and of pollinators in the interior transects.
- I have concerns regarding the number of paths in the structural equations used and whether the sampling size is sufficient for such a complex model.
- In addition, stepwise model selection based on p-values has been criticized and may lead to inconsistent results and has the problem of multiple testing.
- I am not sure whether the way the reference pool was made, using only mainland transect, was appropriate, considering the relatively low sampling size in the mainland.
- The conclusions speak of edges buffering the islands from the negative impacts of habitat loss, but I don't think that such conclusions can be made without looking at species identity and whether, for example, the edges are occupied mostly by generalist species. Increased species richness at edges is a relatively common pattern, but it is often due to generalist, invasive and/or alien species. In addition, there is still the issue of differences in sampling pace between edge and interior transects.
- In figure 3, area and edge effects on network robustness are quite small. Thus, although there is a cascading effect apparent from the SEM, the effects on network robustness do not appear to be strong enough to justify the study's conclusions.

Additional comments below:

Abstract

There was no mention on whether the species occurring at forest edges are the same as those in forest interior. This is essential information: it is possible that pollinator networks at edges are more robust because they are dominated by generalist species, with specialists being lost. So, even though these networks are robust, they may not represent the networks in interior forest.

"We test" should be "We tested"

In some parts of the text, it seems that "habitat fragmentation" actually refers to "habitat loss and fragmentation", which are two related but different processes - for example, decreased area is habitat loss but not fragmentation.

Regarding the methods:

The differences between pace and survey interval at edge and interior transects might have resulted in decreased detection probability of flowering plants in the interior transects.

Site description:

I suggest including information on whether the herb and shrub layers at edges are composed mostly of generalist species or not. I also missed information regarding slope and the difference in altitude between edge and interior - as the islands are former hilltops, there may be important differences in altitude.

Sampling design:

The method described is "paired edge versus interior sites", including 41 islands and 16 mainland sites. However, the transects were 100 m long, one along the edge and another perpendicular to the edge - so, as I understand, the perpendicular transect would span from the edge to 100 m into the forest. I see two problems here: first, this is not edge versus interior sites, which would require a transect at the edge and another at a certain distance from the edge; second, edge effects may extend more than 100 m into the forest, so the "interior" transects may also be subject to edge effects.

The pace and survey interval differed between edge and "interior" transects, with the "interior" transects being sampled at different rates. The reason given for this is that there were more flowering plants at edges, demanding longer survey intervals. Although this makes some sense, this may also have resulted in lower detectability of flowering plants and of pollinators in the "interior" transects.

Is it possible that phenology differed between edge and interior, for example with peak flowering in the interior happening after peak flowering at the edge? If so, could this have influenced the results?

"We considered an insect to be a putative pollinator only if it was touching the anthers and/or stigmas of the flowers." - this is appropriate, but I'm wondering about detectability. Was the observation time sufficient to ensure that pollinators actually touched the anthers, especially considering the difference in pace between edge and interior?

"we estimated the flower area of each plant along each transect at each sampling period" - How was this done? It is not clear to me what you mean by "flower area" and it is not clear how estimates were made along 100 m-long transects.

Finally, if larger islands had more transects sampled, wouldn't these differences in sampling effort be confounded with area effects? This should be discussed and justified.

Network metrics

The relative connectance measure is defined as the number of observed interactions divided by the number of interactions of these species in the whole region. It is not clear, however, how it would be calculated if a species is absent from a transect. For example, pollinator Po1 interacted with plant PI1 in the network, but in the whole region it was observed to interact with plants PI1, PI2 and PI3.

However, plant PI3 was not found in the transect. In the total number of observed interactions for Po1

52 or 3, in this case?

The NODF index should be explained. The modularity score and the algorithm (DIRTLPAwb+) should also be explained.

Structural Equation Modelling:

The hypotheses and the underlying logic make sense to me.

Are structural equations really appropriate for "non-normal error distributions in models"? As far as I know, one of their assumptions is multivariate normality (although I think there are alternatives).

I think that figure S2 could be in the manuscript and not in the supplementary material.

In addition, I have some concerns regarding the model complexity compared to sample size. A total of 41 islands were samples, in addition to 16 mainland transects. If I understood correctly, the edge transects at a given island were combined to form an "edge" network, as were the "interior" transects. The same was done for the 16 mainland edge transects and the 16 mainland interior transects. Thus, there would be a total of 82 networks, but only 42 independent sampling units. Yet there are 45 paths in the model in Figure S2.

Is the stepwise approach to calculate a "final" SEM appropriate? There is much criticism to stepwise model selection in which non-significant effects are sequentially removed (e.g. <https://besjournals.onlinelibrary.wiley.com/doi/full/10.1111/j.1365-2656.2006.01141.x?s=09>). Considering the large number of paths and the problems with multiple testing in sequential model selection, I think that this approach may have resulted in biased results. I would suggest using AIC to compare between a predefined set of ecological models or, alternatively, focusing on simpler, univariate analysis. Alternatively, it should be justified why the approach is appropriate and would not result in statistical artifacts.

Null models:

As I understand, the mainland transects were used as an "interactions pool" (which I think can be considered analogous to a species pool). However, sampling effort in the mainland was relatively small compared to the sampling effort in the islands. Wouldn't it be more appropriate to use all the data (from the mainland and the islands) as the pool of possible interactions? In addition to the sampling effort issue, it is possible that novel interactions occur at the islands which are not present in the mainland, due to novel ecological conditions.

"If the observed values differed significantly from the null model values, this indicates that the observed values showed non-random assembly trajectories as a result of habitat fragmentation, over and above any stochastic effects due to confounding bias in sampling effort between islands." - I think that this would make sense if the mainland transects were truly representative, but I am not convinced that this is the case. Thus, I think that deviations from the null models could have arisen due to sampling error in the mainland transects, with many interactions not being detected there.

I think this manuscript may be of interest: <https://www.annualreviews.org/doi/abs/10.1146/annurev->

6ecolsys-120213-091759

In Null Model I, were the transects drawn with replacement (as in bootstrap)?

Regarding Null Model II, in its stepwise version: "we constrained the reference pollinator pool to only the potential (i.e. non-forbidden) links" - did this exclude pollinators that had forbidden links with some, but not all, plants in the random network? If so, would this not somehow bias the results? Apart from this, I don't see any problems with this null model - the algorithm seems valid and well developed, for which I congratulate the authors.

Null model SEM:

This analysis seems valid to me.

Additional suggestion:

When dealing with effects of area loss and fragmentation, it is important to also look at changes in species composition. Thus, I suggest including at least a simple comparison (e.g. PERMANOVA or canonical correspondence analysis) to see how island area and edges affect species composition.

One more additional suggestion:

As the discussion is focused mostly on area and edge effects, are SEM truly necessary? Couldn't the study questions be answered with simple GLMMs testing for edge and area effects individually for the response variables?

Results:

Overall the results make sense. Although there is little mention of effect sizes in the text, they can be seen in figures 2 and 3. In figures 2 and 3, I suggest changing the colors to colorblind-friendly ones. Also, I liked the shading in figure 3.

However, there is one issue. In Figure 3, there are clear differences between edge and interior for plant and pollinator richness and abundance (or floral resources), as well as for nestedness, but the differences between edge and interior for network robustness are quite small. So I am not convinced that edges actually increase network robustness, notwithstanding the indirect effects.

Discussion:

"we show that habitat edges can buffer the likelihood of pollinator network to collapse with declining forest area" - I don't think that the results permit to make this statement. Although network robustness was greater at edges, it is possible that the species at the edges differ from those in the interior. Thus, interior pollinator networks could collapse and only the edge networks would remain. If there are spillover effects from the edge into interior, I think it may even be possible that the increased abundance of generalist pollinators at the edge may outcompete specialist pollinators in the interior, resulting in species and interaction losses.

- Pavel Dodonov

Reviewer #3 (Remarks to the Author):

The manuscript entitled “Edges increase pollinator network robustness to extinction with declining forest area” by Ren and collaborators presents a study of pollination networks in a fragmented land-bridge island system. The study analyzes, through Structural Equation Models and null models, the direct and indirect effects of fragmentation variables on network robustness through community and network structural patterns. The authors found a positive effect of forest edges on biodiversity that somehow buffers the negative effect of decreasing fragment areas and preventing further extinctions. This result is interesting because it presents a holistic view of the habitat fragmentation process that contradicts our prejudices and provides a counterintuitive mechanism that allows communities to persist in modified habitats.

Despite its complexity, the methods applied are appropriate to test the hypothesis and the conclusions are coherent with the results. The null model approach to disentangle sampling effects vs fragmentation effects looks effective although a bit forced. Nevertheless, I think the approach is quite conservative and the conclusions made by the authors are appropriate. The codes applied are provided by the authors which make this study repeatable.

Despite the manuscript being appealing, I have some concerns listed below:

1-I wonder if we are evaluating pollinator communities’ responses to fragmentation in the same way we evaluate other, forest-specialized organisms. Although evidence is always necessary, I think pollination-oriented studies should focus more on the vulnerability of interactions rather than focusing on species diversity and abundance. In the same logic and from a conservation perspective, the dilemma between interior vs edges should contemplate the identity of interactions occurring at both habitats. It could be possible that some interior interactions are more rare or specialized than those occurring at edges and its conservation value be higher. Accordingly, higher nestedness in edges vs higher modularity in interiors could reflect the presence of generalist, opportunistic species disrupting a syndrome-based pollination network and, consequently the co-evolutive dynamics. Although this sentence is merely speculative, the change in network patterns between interior and edges deserves more study, maybe involving trait-based analysis which is beyond the scope of this study. I would like to read what authors think about these topics and if possible, mention them in the discussion.

Also, the conclusions about edge-interior differences should consider what could be happening in the canopy of forest interiors to have a complete picture of what is going on in these islands. I understand that it would be logistically difficult but at least it should be mentioned in the discussion.

2-the robustness analysis often reflects a static view of network fragility by not allowing different behavioral responses of consumers in face of resource extinctions such as rewiring. How do the authors think the presence of opportunistic and generalist species in edges could affect network robustness?

3-I found the “relative connectance” idea interesting, nevertheless, its accuracy may depend on how complete is the sampling of the regional network and how you define the limits of the region. Some clarifications on this regard would be needed.

4-Is the orientation of edges important in your area? In some cases the orientation of edges can have a huge impact on microclimatic conditions, affecting community structure (Bernaschini et al. 2020,

8

2021).

5-SI, line 73-74. I think it is not necessarily so. In a highly compartmentalized network with strong boundaries between modules adding new species would not necessarily increase nestedness.

-line 213-214: replace "tends to suggest" with "suggest"

-lines 219-224: I don't see so clear causality between high connectance and low nestedness. I think those variables are not necessarily related.

-lines 230-231: "in the face of" phrase is repeated.

-Line 231: replace "link" with factor or similar word.

References:

Bernaschini, M., M. Rossetti, G. Valladares, and A. Salvo. 2021. Microclimatic edge effects in a fragmented forest: disentangling the drivers of ecological processes in plant-leafminer-parasitoid food webs. *Ecological Entomology*.

Bernaschini, M., G. Valladares, and A. Salvo. 2020. Edge effects on insect-plant food webs: assessing the influence of geographical orientation and microclimatic conditions. *Ecological Entomology*.

Reviewer #1 Review Attachment #1:aklein

Notiz

I suggest to write "forest edges" as you can not conclude from your study that this is the case for different kind of habitat edges

1

For submission to: *Nature Ecology & Evolution*

Content type: Articles

**Edges increase pollinator network robustness to extinction with declining forest area**

Peng Ren¹, Raphael K. Didham^{2,3}, Mark V. Murphy², Di Zeng⁴, Xingfeng Si⁴, Ping Ding^{1*}

¹ MOE Key Laboratory of Biosystems Homeostasis & Protection, College of Life Sciences,

Zhejiang University, Hangzhou, Zhejiang, 310058 P.R. China

² School of Biological Sciences, The University of Western Australia, Crawley, WA, 6009

Australia

³ CSIRO Health & Biosecurity, Centre for Environment and Life Sciences, Floreat, WA, 6014

Australia

⁴ Zhejiang Tiantong Forest Ecosystem National Observation and Research Station, School of

Ecological and Environmental Sciences, East China Normal University, Shanghai 200241,

China

**Email address:**

Author-1: Peng Ren (renpeng@zju.edu.cn), ORCID iD: 0000-0001-6033-6188

Author-2: Raphael K. Didham (raphael.didham@uwa.edu.au), ORCID iD: 0000-0001-6685-

7005

Author-3: Mark V. Murphy (murphymv@gmail.com), ORCID iD: 0000-0001-9178-0871

Author-4: Di Zeng (zengdiseed@gmail.com), ORCID iD: 0000-0003-2780-3738

Author-5: Xingfeng Si (sixf@des.ecnu.edu.cn), ORCID iD: 0000-0003-4465-2759

*Author for Correspondence: Ping Ding (dingping@zju.edu.cn), ORCID iD: 0000-0001-

5825-0932, Telephone: +8613606640161

**Word count:** abstract 150 words; main text 2674 words; Methods 2198 words; 5 figures, 0

tables; 80 references

Abstract

Edge effects often exacerbate species decline due to habitat loss. In forested ecosystems,
however, many pollinators actually prefer open sunny conditions created by edge

disturbances. We test the hypothesis that forest edges buffer plant-pollinator interaction
this is a very general sentence, consider to revise
we recorded ca. 20,000 plant-pollinator interactions along edge and interior gradients of
islands over 3 years. Using Structural Equation Models, we
floral resources decline with decreasing island area at both interior and edge sites, but edges

maintain 10-fold higher pollinator abundance and richness regardless of area loss. Edge
networks have significantly higher nestedness and lower modularity than interior networks,
maintaining high robustness to extinction in the face of habitat area loss, while forest interior
networks collapse. Anthropogenic changes benefit community and network processes in the
absence of natural gap-phase dynamics in small degraded forest remnants.

**aklein** keywords: edge effects; habitat fragmentation; mutualist networks; network robustness;
**Notiz** pollinator decline; structural equation model
forest edge, right?

**Introduction**

Anthropogenic land-use change has profound consequences for pollinator diversity¹. Habitat
loss has become one of the leading causes of species endangerment², extinction of ecological
interactions³ and consequent loss of essential pollination services^{2,4}. Moreover, ecosystem
decay within small habitat remnants can double biodiversity loss due to habitat loss⁵. In
particular, multiple components of habitat fragmentation, such as increasing edge effects and
decreasing connectivity of remnant habitats, dramatically alter the spatial distribution of
pollinator resources^{1,2,6,7} with far-reaching impacts on species interaction networks^{4,8,9,10}.
57¹¹. The conventional narrative is that the combined effects of habitat fragmentation on plant-
58 pollinator communities tend to exacerbate the negative effects of habitat loss alone^{1,9,12,13}.
However, this generalization stands in striking contrast to empirical field observations that
open light-filled forest edges support more flowers and more pollinators than the dark interior
of closed-canopy forests^{6,14,15,16,17}. The mismatch in observations is almost certainly
because most studies on the effects of habitat fragmentation on plant-pollinator communities
have focused on open-habitat systems such as grasslands or croplands where edge effects are
likely negative or neutral^{8,9,10,12,13}, rather than on closed-canopy forest systems in which
edge effects might be positive. Historically, natural gap-phase dynamics in continuous old-
growth forests would have provided space and resources for light-loving pollinator species,
but in contemporary degraded and regenerating forests these habitats are now rare^{18,19,20,21}.
Fragment edges may provide an anthropogenic analogue of natural gap-phase conditions, and
thus have a positive, rather than negative, edge effect on pollinator communities in forest
systems. Surprisingly, comparatively little is known about the potential for positive edge

effects on plant-pollinator communities in forest systems, beyond a limited focus on
abundance or species richness^{16, 22, 23}, rather than the potential network-wide consequences if
positive edge effects mitigate the negative consequences of habitat loss on network structure
and stability
At the whole network scale, habitat loss has well-recognised
pollinator interactions and network architecture^{8, 9, 10, 24, 25, 26}, based predominantly on studies
in non-forest systems. Habitat loss causes non-random loss of interactions^{8, 27, 28, 29}, which
disrupts plant-pollinator interactions, leading to higher network
nestedness^{9, 26}, which will intensify competition among species at the same trophic level^{30, 31}
and destabilize networks in the face of disturbance^{25, 32}. Meanwhile, network connectance
typically increases as habitat area declines⁹, suggesting fewer potential resource linkages
remain in the network^{33, 34}. Although plant-pollinator networks can be relatively robust to
changing spatial configuration of habitat fragments, by responding to species loss through
high potential for adaptive switching of interaction partners^{35, 36, 37}, habitat fragmentation is
widely considered to have a negative, exacerbating influence on the impacts of habitat loss

forest edges not habitat edges. Here, we show that open light-filled forest edges have a contrasting positive effect
on plant-pollinator networks within forests, then we would expect to see the opposite effects
with edges buffering rather than exacerbating network instability in the face of habitat loss.

Here, we use structural equation models (SEM) to test the hypothesis that habitat edges
91 buffer the negative effects of forest loss on plant-pollinator community structure and network
architecture. The SEM partitions the direct and cascading indirect effects of habitat

aklein
Notiz
what does network stability
means here?

aklein
Notiz
forest edges not habitat
edges

fragmentation on floral resource availability, plant and pollinator community structure, and
the architecture of species interaction networks, using ca. 20,000 flower visitation records
from 68 flowering plant species on 41 islands and 46 mainland sites in the Thousand Island
Lake (TIL) region of Eastern China. The TIL studies provide an opportunity to
overcome potential confounding influences of forest fragmentation and forest edges with forest
edges with forest edges. Conditions that would start to more generally
might mask the relative effects of land-use change on pollinator abundance. Understanding a new
hydroelectric dam in eastern China were clear-felled at the same time, effectively resetting
plant-pollinator community reassembly to a common set of initial starting conditions³⁹. The
continuous mainland sites thus provide an ideal ‘unfragmented’ reference state against which
to examine changes in plant-pollinator community structure and network architecture on
islands of varying area, distance of isolation and degree of edge influence. We show that
forest edges have 10-fold higher pollinator abundance and species richness than the forest

interior, and maintain higher species richness, lower modularity and greater robustness to
disturbance. These results suggest that in fragmented forest systems, the
effects of habitat loss on pollinators are not tend to exacerbate the negative effects of habitat loss on
pollinators, but instead buffers network robustness to extinction.

Results

Over 3 years (20 surveys per site) we documented a total of 19486 individual pollinator
interactions with plants in ca. 960 hours of flower observations, spanning 3226 observed
associations between 68 species of flowering plants and 313 species of pollinators (Tables

S2-S3). Species similarity between islands and mainland was high, with 54 plant species and
 269 pollinators in the mainland, 67 plant species and 310 pollinators on the 41 islands, and
 only 1 plant species and 3 pollinators that were not found on islands. Floral resources,
 pollinator abundance and richness on the edges of the islands were approximately 10-fold
 higher than in the interior (Table S1). Sample coverage was uniformly high for plants across

**aklein**
 **Notiz** lands (mean 0.89 ± 0.02 SD), but was lower for pollinators (0.84 ± 0.03) and for plant-
 better to use "forest area"? (0.68 ± 0.03), and declined noticeably with decreasing island area (Fig.
 S1).

In the final most-parsimonious piecewise SEM model (Fig. 2; Fisher's $C = 51.83$, $P =$
 0.63) 24 of the 45 hypothesized paths were retained in the model (Tables S4-S5). There were
 striking effects of decreased area and increasing proximity to edge on plant and pollinator
 community structure (Fig. 2). The abundance and richness of both plants and pollinators
 declined significantly on smaller islands (negative effect sizes for standardized path
 coefficients in Fig. 2, Fig. 3a-d). For floral resources and plant richness, null model
 comparisons indicated that community metrics on most islands were significantly lower than
 expected based on passive sampling effects from the mainland reference pool (Fig. 3a, b).
 This indicates that declining island area strongly constrains floral resource availability and
 host plant richness for pollinating insects. By contrast, the strong declines observed in
 pollinator abundance on smaller islands were similar to values that would be expected under
 a passive sampling model on the majority of islands (Fig. 3c). This suggests that lower
 sample coverage for pollinators on small islands (Fig. S1) is an equally parsimonious
 explanation for the observed abundance trends, rather than necessarily fragmentation-driven

impacts. Importantly, in the case of decreased area effects on pollinator richness, the observed
trends were driven predominantly by the indirect effect of island area on pollinator abundance
(Fig. 3c, d; Fig. 4a), rather than any direct relationship between floral resources and pollinator
richness or between plant richness and pollinator richness, suggesting a passive sampling
effect drives pollinator richness declines with decreasing island area in this case.

Proximity to edge had even larger effect sizes on plant and pollinator community
attributes than decreasing island area, and these effects were uniformly positive rather than
negative (Fig. 2). Island edges had significantly greater floral resources, plant species
richness, pollinator abundance and pollinator richness than island interiors (Fig. 3a-d).
Unexpectedly, the edge to interior difference in community attributes was consistent across
all islands (Fig. 3a-d), with area by edge interactions weak or non-significant (Fig. 2, Fig. 4a).
Moreover, community responses to island area and proximity to edge were not influenced by
distance of isolation from the mainland (Fig. 4a).

Network architecture varied significantly between edge and interior, and between small
versus large islands (Fig. 3e-h). However, these effects were not due to direct influences of
island attributes on network architecture, but instead occurred via the indirect influences of
altered plant and pollinator community structure (Fig. 4b). There was only a weak positive
effect of island isolation on nestedness, and a weak interaction effect of proximity to edge
exacerbating the negative effect of island area on relative connectance (Fig. 2, Fig. 4b). In
both cases, observed effect sizes fell within the 95% CI of null model SEM simulations (Fig.
5), suggesting these effects were no greater than would be expected from passive sampling
effects from the mainland reference pool.

Indirect effects on network architecture varied between interior and edge communities.
 For forest interior communities, a decrease in island area had a negative indirect effect on
 relative connectance (i.e., higher relative connectance within the smaller plant-pollinator
 assemblages on small islands, compared with large islands; Fig. 3e), and significantly lower
 nestedness and higher modularity on small compared with large islands (Fig. 3f, g). These
 changes in network parameters had a cascading effect on network robustness to extinction,
 with forest interior communities having a lower robustness to extinction on small islands
 compared with large islands (Fig. 3h). For forest edge communities, by contrast, relative
 connectance was substantially lower and nestedness substantially higher than observed for
 interior communities, and these network parameters were comparatively less affected
 by declining forest area (Fig. 3e, f). Higher nestedness at edges, in particular, had a strong positive
 effect (path 45 in Fig. 2) on robustness to extinction for edge communities compared with
 interior communities (Fig. 3h).

aklein
 Notiz
 add "fragementation"

Discussion

Both habitat area loss and changing spatial configuration of the remaining habitat can disrupt
 ecological networks^{2,6}. Here, we show that the combined effects of these  sses do not
 always have compounding negative effects^{14,16}. Unanticipated cascading indirect effects can
 act to mitigate, not exacerbate, biodiversity loss and network collapse. In a naturally-
 regenerating forest system in eastern China, we show that habitat edges can buffer the
 likelihood of pollinator network to collapse with declining forest area. Our SEM revealed the
 complex relationship among island attributes, community structure and network architecture

in a fragmented forest system. Whereas the direct and indirect effects of declining habitat
area on plant-pollinator community structure and network architecture were predominantly
negative, these were counter-balanced by predominantly positive effects of proximity to
habitat edge. Edges maintained 10-fold higher pollinator abundance and richness than the
forest interiors. More importantly, edges increased network robustness to extinction, rather
than exacerbating the negative effect of forest area loss on network architecture, thus
reducing the risk of plant-pollinator system collapse.

Positive effects of habitat fragmentation on plant-pollinator community structure and
network architecture may seem surprising at first glance, given the weight of evidence
showing negative effects^{1, 9, 12, 13}. However, previous network studies have been conducted
almost entirely in non-forest systems in which the relative contrast in light conditions and
floral resources between edge and interior is weak (or favours interior habitats), whereas our
findings for the forested system at TIL are more in line with the observation that pollinators
prefer open sunny conditions rather than the dark interior of closed-canopy forests. For
example, on the islands we studied, canopy density was high in the forest interior (typically >
90% canopy cover)⁴⁰, few shrub species could persist (e.g., *Loropetalum chinense*,
*Symplocos sumuntia* and *Symplocos paniculata*), herbs were rare, and only a few shade-
tolerant species were seen occasionally flowering (e.g., *Liriope spicata* and *Lysimachia*
*congestiflora*). Floral resource availability in the interior was clearly insufficient to attract
insect pollinators^{41, 42, 43, 44}, and as they tend to have high mobility they are less affected by
distance to edge or even distance of isolation from the mainland^{45, 46, 47, 48, 49}, so pollinator
abundance tended to mirror the distribution of floral resources along edges (Fig.3, Table S1).

We were careful to discriminate how these shifts in resource and abundance distributions
might influence ecological trends in richness and network architecture, after first accounting
for potentially spurious confounding influences of lower sample coverage⁵⁰ in small
compared with large fragments⁵¹ and interior sites versus edges. Few studies have taken this
type of null model approach to discriminating true fragmentation effects from potential
sampling artefacts. In our system, we found that observed floral resources and plant richness
on islands were significantly lower than would be expected from simple sampling effects
alone (compared with a random draw from the mainland reference communities), whereas the
strong declines in observed pollinator abundance and richness from interior to edge and from
large to small islands were not significantly different from null expectation based on
equivalent sampling effort at mainland reference interiors and edges, respectively. This tends
to suggest that plant richness and floral abundance are strongly negatively affected by
fragmentation, but associated declines in pollinator abundance and richness are a passive
(indirect) consequence of floral resource shifts, rather than pollinators being directly
negatively affected by other mechanisms associated with declining patch area or edge effects,
in this case.

The positive effect of habitat edges is also reflected in the way that habitat area effects on
network architecture were strongly buffered by proximity to edge. Plant-pollinator networks
in the interior of smaller islands had very low network size and consequently higher relative
connectance among interaction partners^{26, 52, 53, 54}, leading to dramatically lower nestedness
and higher network modularity. In fragmented habitats, specialist pollinators visit only a
limited number of plants and are more likely to become extinct than generalists^{12, 55}. The

high extinction risk of specialists potentially breaks the degree of interaction asymmetry in
smaller networks, which is a key attribute of nestedness⁵⁶, and finally reduces nestedness and
increases the modularity of island interior networks. The larger size of plant-pollinator
networks at habitat edges buffers the impact of habitat loss on relative connectance, and
significantly increases nestedness, decreases modularity and increases network robustness in
the face of plant extinction. As found in previous studies^{32, 34, 57}, increasing network
nestedness in the face of decreasing habitat area appears to be the key link to mitigating
secondary extinctions from species loss. It is increasingly appreciated that anthropogenic
stressors can potentially offset one another, reducing secondary extinction risk⁵⁵. In
fragmented secondary forest, our research provides evidence that habitat edges have the
potential to play a positive role.

[revised manuscript text omitted]

We considered an insect to be a putative pollinator only if it was touching the anthers
and/or stigmas of the flowers. For pollinators and plant species that could not be identified in
the field, voucher specimens were hand-collected and identified in the laboratory. The
identifications were later verified by specialists (see Acknowledgments). We recorded the
number of pollinators of each insect species visiting each flowering plant species during the
sampling period. At the same time, we estimated the flower area of each plant along each
transect at each sampling period and summed total flower area together as an estimate of
floral resources on the islands during the intervals that were sampled. The same sampling
strategy was used at the 16 mainland sites, with sampling effort designed to match the

number of surveys on the largest island (i.e., 16 pairs of edge and interior transects). Lists of
 species and their recorded abundances are presented in Tables S1-S3. We also calculate the
 sampling completeness of plants, pollinators and their interactions in each island (see
 Supplementary Methods 1, Fig S1).

Network Metrics

Using the data from the plant-pollinator surveys, we assembled quantitative interaction
 networks (with the 3 years of data combined  for each of the 41 islands using the edge and
 interior transects separately. For the mainland network, quantitative interaction networks
 were constructed in the same way for the 16 edge transects separately from the 16 interior
 transects, which represented the mainland ('unimodal' reference network in null model
 analyses. For each of the observed island networks we calculated four metrics of network
 architecture: relative connectance, nestedness, modularity, and robustness.

[revised manuscript text omitted]

among predictors ⁷⁷. The 95% confidence interval for effects was calculated using 1000
bootstrapped estimates for each response. Model-predicted total effects were presented using
partial regression coefficients calculated using the *predEff* function in the *semEff* package.

**Null model for community structure and network architecture**

Plant-pollinator community structure and network architecture on islands might differ from
the mainland reference sites both as a result of fragmentation processes and as a result of
biases in sampling effort. The purpose of the null model is to determine whether variation in
plant-pollinator community structure and network architecture on islands is significantly
greater (or less) than expected from a simple 'passive sampling effect' from the continuous
mainland reference pool. To do this, we compiled plant-pollinator interaction data from all
sampling transects on the mainland edge and interior, respectively, then used this as our
expected 'reference' pools. From the edge and interior reference pools, we use two methods to
simulate 'null communities' and 'null networks': (1) Null model I: a random draw of the
same number of transects from the mainland as observed on each of the 41 sampled islands
(i.e., constraining the number of sampling transects used to acquire a null estimate of floral
resources, plant richness, pollinator richness and pollinator abundance); (2) Null model II: a
random draw of the same number of pairwise interactions, while ensuring the same numbers
of plant and pollinator species were selected as those observed on each of the 41 sampled
islands (i.e., constraining both network abundance and network size). With respect to null
model II, we refer to the principle of network null models originally proposed by Vázquez et
al. ⁷⁸, but with the difference that we considered forbidden and non-forbidden interactions.
Forbidden interactions were considered to be potential interactions that were never observed

to occur in the three years of sampling. Then we conducted null
 interactions, we only considered the non-forbidden links. At the
 probabilities are determined by relative abundances of plants and
 sampling process was repeated 1000 times, generating 1000 simulated null networks for each
 island. We repeated this process separately for the edge and interior networks of each island.

aklein
 Notiz
 This was said before.
 Consider to delete.

The concepts, step-by-step procedures and R code for the null draw methods are presented in
 Supplementary Methods 3 (Fig. S3-S6) and Code availability.

We calculated the standardized effect size, that is $SES = (\alpha_{obs} - \alpha_{null}) / SD(\alpha_{null})$, as a
 measure of the magnitude and direction of the difference between the observed and the null
 values for each island⁷⁹. A positive or negative value of SES indicates that the observed
 value is above or below the mean of the null distribution, respectively. We used the
 approximate statistical significance at the 5% level for a two-tailed test when estimating the
 significance. If the observed values differed significantly from the null model values, this
 indicates that the observed values showed non-random assembly trajectories as a result of
 habitat fragmentation, over and above any stochastic effects due to confounding bias in
 sampling effort between islands. In contrast, if the observed values were not significantly
 different from the null model, variation in diversity or network structure was considered
 indistinguishable from the stochastic biases that might be expected from passive sampling
 effects.

Null model SEM

After using the observed data to construct the SEM (observed SEM) and acquiring the null
 model simulation results for community structure and network architecture, we further

wanted to evaluate whether the observed causal relationships (paths) deviated significantly
from what might be expected under a passive sampling effect from the continuous mainland
reference sites. Therefore, we ran the same SEM path structure, using 1000 randomly-drawn
null model values for community structure and network architecture (as described above) to
construct 1000 SEM models and evaluate the significance of each observed path against the
mean expected path coefficients (\pm 95% Confidence Intervals) from the 1000 null model
SEMs. All statistical analyses were conducted using R version 4.0.1⁸⁰.

Acknowledgements

We thank the Xin'an River Ecological Development Group Corporation and Forestry Bureau
of Chun'an County for permits necessary to conduct research at the Thousand Island Lake.
We are very grateful to Quanguai Jiang, Miaoyuan Zhang, and other lab group members for
field assistance and Maxwell C. Wilson for the map construction. We acknowledge Zhihong
Xu at Zhejiang A&F University for the identification of insect species. We acknowledge
Shenhao Yao for the identification of plant species. We thank Xinyu Xu, Yuxiao Han and Xue
Zhang for their help with data analysis. This study was funded by the National Natural
Science Foundation of China (Grant/Award number 32030066 to P.D., 31872210 and
32071545 to X.F.S, 31930073 to Mingjian Yu), the Natural Science Foundation of Zhejiang
Province (#LD21C030002 to P.D.), the China Scholarship Council (Grant number
201906320342 to P.R.), the Shanghai Rising-Star Program (#19QA1403300 to X.F.S) and the
Program for Professor of Special Appointment (Eastern Scholar) (#TP2020016 to X.F.S).

Author contributions

P.R. and P.D. came up with the idea and designed the study, with support from R.K.D. P.R.
and R.K.D developed and implemented null model methods. D.Z and M.V.M helped write
statistical model analysis code. P.R., X.F.S and P.D. conducted the analyses. P.R. and R.K.D
led the writing, with substantial contributions from P.D., X.F.S, M.V.M. and D.Z. All authors
provided input on manuscript revisions.

Competing interests

The authors declare no competing interests.

Correspondence and requests for materials should be addressed to Ping Ding.

Code availability

The code for null models is available from the *figshare* repository with the following
download link: <https://figshare.com/s/f533c88821f028d5c35c>

References

- 1. Millard L, *et al.* Global effects of land-use intensity on local biodiversity. *Nature*
*communications* **12**, 1-11 (2021).
2. Haddad NM, *et al.* Habitat fragmentation and its lasting impact on biodiversity. *Science*
*advances* **1**, e1500052 (2015).

31

 3. Valiente-Banuet A, *et al.* Beyond species loss: the extinction of ecological interactions in a changing
 world. *Functional Ecology* **29**, 299-307 (2015).
 4. Rybicki J, Abrego N, Ovaskainen O. Habitat fragmentation and species diversity in competitive
 communities. *Ecology Letters* **23**, 506-517 (2020).
 5. Chase JM, Blowes SA, Knight TM, Gerstner K, May F. Ecosystem decay exacerbates biodiversity loss
 with habitat loss. *Nature* **584**, 238-243 (2020).
 6. Ewers RM, Didham RK. Confounding factors in the detection of species loss in fragmented habitats
 fragmentation. *Biological Reviews* **81**, 117-142 (2006).
 7. Didham RK. Ecological consequences of habitat fragmentation. In: *Ecological Consequences of Habitat
 (ELS) (ed. R. Jansson)*. John Wiley & Sons, UK (2010).
 8. Aizen MA, Sabatino M, Tylianakis JM. Specialization and rarity predict nonrandom loss of interactions
 from mutualist networks. *Science* **335**, 1486-1489 (2012).
 9. Spiesman BJ, Inouye BD. Habitat loss alters the architecture of plant-pollinator interaction networks.
 *Ecology* **94**, 2688-2696 (2013).
 10. Aizen MA, Gleiser G, Sabatino M, Gilarranz LJ, Bascompte J, Verdu M. The phylogenetic structure of
 plant-pollinator networks increases with habitat size and isolation. *Ecology Letters* **19**, 29-36 (2016).
 11. Emer C, *et al.* Seed-dispersal interactions in fragmented landscapes-a metanetwork approach. *Ecology
 Letters* **21**, 484-493 (2018).
 12. Fortuna MA, Bascompte J. Habitat loss and the structure of plant-animal mutualistic networks. *Ecology
 Letters* **9**, 278-283 (2006).
 13. Grass I, Jauker B, Steffan-Dewenter I, Tschamtko T, Jauker F. Past and potential future effects of
 habitat fragmentation on structure and stability of plant-pollinator and host-parasitoid networks. *Nature
 Ecology & Evolution* **2**, 1408-1417 (2018).
 14. Wolf SM, Barrie BTC. Assessing the integrated effects of landscape
 fragmentation on plants and plant communities: the challenge of multiprocess-multiresponse dynamics.
 *Journal of Ecology* **102**, 882-895 (2014).
 15. Hunter ML, Hunter Jr ML. *Maintaining biodiversity in forest ecosystems*. Cambridge university press,
 pp. 210-233 (1999).
 16. Hadley AS, Betts MG. The effects of landscape fragmentation on pollination dynamics: absence of
 evidence not evidence of absence. *Biological Reviews* **87**, 526-544 (2012).

aklein
 Notiz
 "Biological Reviews" not
 "Biological reviews"

aklein
 Notiz
 chance to "University Press"

17. Morante JJ, Thompson JR, Tang X, Reinmann AB, Hutyra LR. Elevated growth and biomass along
 temperate forest edges. *Nature Communications* **12**, 1-8 (2021).

18. ... dynamics and tree regeneration. *Journal of Forest Research* **5**, 223-229
 change to "Evolution"

19. Martinez-Ramos M, Alvarez-Buylla E, Sarukhan J. Tree demography and gap dynamics in a tropical
 rain forest. *Ecology* **70**, 555-558 (1989).

20. ... Princeton, New Jersey: Princeton University Press, pp. 188-226. (2011).

21. Schnitzer SA, Carson WP. Treefall gaps and the maintenance of species diversity in a tropical forest.
 *Ecology* **82**, 913-919 (2001).

22. Gayler C, *et al.* Flowering fields, organic farming and edge habitats promote diversity of plants and
 arthropods on arable land. *Journal of Applied Ecology* **58**, 1155-1166 (2021).

23. Bailey S, Requier F, Nusillard B, Roberts SP, Potts SG, Bouget C. Distance from forest edge affects bee
 pollinators in oilseed rape fields. *Ecology and evolution* **5**, 370-380 (2014).

24. Hagen M, *et al.* Biodiversity, Species Interactions and Ecological Networks in a Fragmented World.
 *Advances in Ecological Research, Vol 46: Global Change in Multispecies Systems, Pt 1* **46**, 89-210
 (2012).

25. Thebault E, Fontaine C. Stability of ecological communities and the architecture of mutualistic and
 trophic networks. *Science* **329**, 853-856 (2010).

26. Traveset A, Castro-Urgal R, Rotllan-Puig X, Lazaro A. Effects of habitat loss on the plant-flower
 visitor network structure of a dune community. *Oikos* **127**, 45-55 (2018).

27. Rezende EL, Lavabre JE, Guimaraes PR, Jordano P, Bascompte J. Non-random coextinctions in
 phylogenetically structured mutualistic networks. *Nature* **448**, 925-928 (2007).

28. Staddon P, Lindo Z, Crittenden PD, Gilbert F, Gonzalez A. Connectivity, non-random extinction and
 ecosystem function in experimental metacommunities. *Ecology Letters* **13**, 543-552 (2010).

29. Wardle DA, Bardgett RD, Callaway RM, Van der Putten WH. Terrestrial Ecosystem Responses to
 Species Gains and Loss. *Science* **332**, 1273-1277 (2011).

30. Sargent RD, Ackerly DD. Plant-pollinator interactions and the assembly of plant communities. *Trends*
 *in Ecology & Evolution* **23**, 123-130 (2008).

aklein
 Notiz

aklein
 Notiz

Do not use capital letters in the title

- 31. Bastolla U, Fortuna MA, Pascual-García A, Ferrera A, Luque B, Bascompte J. The architecture of
mutualistic networks minimizes competition and increases biodiversity. *Nature* **458**, 1018-1020 (2009).
- 32. Rohr RP, Saavedra S, Bascompte J. On the structural stability of mutualistic systems. *Science* **345**,
1253497 (2014).
- 33. Dunne JA, Williams RJ, Martinez ND. Network structure and biodiversity loss in food webs:
robustness increases with connectance. *Ecology Letters* **5**, 558-567 (2002).
- 34. Pawar S. Why are plant-pollinator networks nested? *Science* **345**, 383-383 (2014).
- 35. Kaiser-Bunbury CN, Muff S, Memmott J, Muller CB, Caflich A. The robustness of pollination
networks to the loss of species and interactions: a quantitative approach incorporating pollinator
behaviour. *Ecology Letters* **13**, 442-452 (2010).
- 36. Evans DM, Pocock MJO, Memmott J. The robustness of a network of ecological networks to habitat
loss. *Ecology Letters* **16**, 844-852 (2013).
- 37. Ponisio LC, Gaiarsa MP, Kremen C. Opportunistic attachment assembles plant-pollinator networks.
*Ecology Letters* **20**, 1261-1272 (2017).
- 38. Wilson MC, *et al.* Habitat fragmentation and biodiversity conservation: key findings and future
challenges. *Landscape Ecology* **31**, 219-227 (2016).
- 39. Zhong L, Didham RK, Liu J, Jin Y, Yu M. Community re-assembly and divergence of woody plant
traits in an island-mainland system after more than 50 years of regeneration. *Diversity and*
*Distributions* **27**, 1435-1448 (2021).
- 40. Liu J, *et al.* The asymmetric relationships of the distribution of conspecific saplings and adults in forest
fragments. *Journal of Plant Ecology* **13**, 398-404 (2020).
- 41. Lennartsson T. Extinction thresholds and disrupted plant-pollinator interactions in fragmented plant
populations. *Ecology* **83**, 3060-3072 (2002).
- 42. Aguilar R, Ashworth L, Galetto L, Aizen MA. Plant reproductive susceptibility to habitat
fragmentation: review and synthesis through a meta-analysis. *Ecology Letters* **9**, 968-980 (2006).
- 43. Kremen C, *et al.* Pollination and other ecosystem services produced by mobile organisms: a conceptual
framework for the effects of land-use change. *Ecology Letters* **10**, 299-314 (2007).
- 44. Goulson D, Nicholls E, Botias C, Rotheray EL. Bee declines driven by combined stress from parasites,
pesticides, and lack of flowers. *Science* **347**, 1255957 (2015).

- 45. Gathmann A, Tschamtko T. Foraging ranges of solitary bees. *Journal of Animal Ecology* **71**, 757-764
(2002).
- 46. Tomé-Noguera A, *et al.* Determinants of spatial distribution in a bee community: nesting resources,
flower resources, and body size. *PLoS One* **9**, e97255 (2014).
- 47. Greenleaf SS, Williams NM, Winfree R, Kremen C. **klein** foraging ranges and their relationship to body
size. *Oecologia* **153**, 589-596 (2007). **Notiz**
- 48. Winfree R, Bartomeus I, Cariveau DP. Native pollinators in agroecosystems. *Annual Review of*
*Ecology, Evolution, and Systematics* **42**, 1-22 (2011).
- 49. Jauker F, Diekötter T, Schwarzbach F, Walters V. Pollinator dispersal in an agricultural matrix:
opposing responses of wild bees and hoverflies to landscape structure and distance from main habitat.
*Landscape Ecology* **24**, 547-555 (2009).
- 50. Roswell M, Dushoff J, Winfree R. A conceptual guide to measuring species diversity. *Oikos* **130**, 321-
338 (2021).
- 51. Schoereder JH, *et al.* Should we use proportional sampling for species-area studies? *Journal of*
*Biogeography* **31**, 1219-1226 (2004).
- 52. Devoto M, Medan D, Montaldo NH. Patterns of interaction between plants and pollinators along an
environmental gradient. *Oikos* **109**, 461-472 (2005).
- 53. Jordano P. Patterns of Mutualistic Interactions in Pollination and Seed Dispersal-Connectance,
Dependence Asymmetries, and Coevolution. *Am Nat* **129**, 657-677 (1987).
- 54. Petanidou T, Kallimanis AS, Tzanopoulos J, Sgardelis SP, Pantis JD. Long-term observation of a
pollination network: fluctuation in species and interactions, relative invariance of network structure and
implications for estimates of specialization. *Ecology Letters* **11**, 564-575 (2008).
- 55. Brodie JF, *et al.* Secondary extinctions of biodiversity. *Trends in ecology & evolution* **29**, 664-672
(2014).
- 56. Vazquez DP, Aizen MA. Asymmetric specialization: A pervasive feature of plant-pollinator
interactions. *Ecology* **85**, 1251-1257 (2004).
- 57. Memmott J, Waser NM, Price MV. Tolerance of pollination networks to species extinctions.
*Proceedings of the Royal Society B-Biological Sciences* **271**, 2605-2611 (2004).
- 58. Malhi Y, Gardner TA, Goldsmith GR, Silman MR, Zelazowski P. Tropical forests in the Anthropocene.
*Annual Review of Environment and Resources* **39**, 125-159 (2014).

aklein Lewis SL, Edwards DP, Galbraith D. Increasing human dominance of tropical forests. *Science* **349**,
 Notiz 827-832 (2015).

Conservation with a capital Is habitat fragmentation good for biodiversity? *Biological conservation* **226**, 9-15
 letter (2018).

61. Fortuna MA, *et al.* Nestness versus modularity in ecological networks: two sides of the same coin?
 *Journal of Animal Ecology* **79**, 811-817 (2010).

62. Bascompte J, Jordano P, Melian CJ, Olesen JM. The nested assembly of plant-animal mutualistic
 networks. *Proceedings of the National Academy of Sciences of the United States of America* **100**, 9383-
 9387 (2003).

63. Almeida-Neto M, Guimaraes P, Guimaraes Jr PR, Loyola RD, Ulrich W. A consistent metric for
 nestedness analysis in ecological systems: reconciling concept and measurement. *Oikos* **117**, 1227-
 1239 (2008).

aklein Ulrich W, Almeida-Neto M, Gotelli NJ. A consumer's guide to nestedness analysis. *Oikos* **118**, 3-17
 Notiz (2009).

use captial letters for the
 journal and in general
 carefully format your
 references according to the
 journal style

67. Oksanen J, *et al.* vegan: Community ecology package. R package version 2.5-7. [https://CRAN.R-](https://CRAN.R-project.org/package=vegan)
 [project.org/package=vegan](https://CRAN.R-project.org/package=vegan) (2020).

68. Dormann CF, Fruend, J., Gruber, B., Dormann, M.C.F., LazyData,  ByteCompile, T. Package
 'bipartite'. R package version 2.16. <https://CRAN.R-project.org/package=bipartite>. (2021).

69. Poccock MJO, Evans DM, Memmott J. The Robustness and Restoration of a Network of Ecological
 Networks. *Science* **335**, 973-977 (2012).

70. Scherber C, *et al.* Bottom-up effects of plant  multitrophic interactions in a biodiversity
 experiment. *Nature* **468**, 553-556 (2010).

71. Schleuning M, *et al.* Ecological networks are  this is a different format. I
 climate change. *Nature communications* **7**, 1-9  stop commenting on the
 references as there are too
 many inconsistencies.

72. Shipley B. Confirmatory path analysis in a general

- 73. Lefcheck JS. PIECEWISESEM: Piecewise structural equation modelling in R for ecology, evolution,
and systematics. *Methods in Ecology and Evolution* **7**, 573-579 (2016).
- 74. Bates D, Mächler M, Bolker B, Walker S. Fitting linear mixed-effects models using lme4. *Journal of*
*Statistical Software* **67**, 1-48 (2014).
- 75. Shipley B. The AIC model selection method applied to path analytic models compared using a d-
separation test. *Ecology* **94**, 560-564 (2013).
- 76. Murphy M. semEff: Automatic calculation of effects for piecewise structural equation models. *R*
*package version 040*, (2020).
- 77. Dudgeon P. A comparative investigation of confidence intervals for independent variables in linear
regression. *Multivariate behavioral research* **51**, 139-153 (2016).
- 78. Vazquez DP, Melian CJ, Williams NM, Bluthgen N, Krasnov BR, Poulin R. Species abundance and
asymmetric interaction strength in ecological networks. *Oikos* **116**, 1120-1127 (2007).
- 79. Jung V, Violle C, Mondy C, Hoffmann L, Muller S. Intraspecific variability and trait-based community
assembly. *Journal of Ecology* **98**, 1134-1140 (2010).
- 80. R Core Team. R: A Language and Environment for Statistical Computing. In: *R Foundation for*
*Statistical Computing* (2020).

 **Figure 1.** Study site locations. The 41 study islands (green shading; B01-B07, S01-S48) and
 the 16 mainland sampling sites (grey shading and red dots; M1-M16) on the Thousand Island
 Lake, Zhejiang Province, eastern China.

aklein
 Notiz
 do not use a combination of
 green and red

 **Figure 2.** Generalised multilevel structural equation model (SEM) showing the direct and
 indirect pathways through which habitat fragmentation influences plant-pollinator community
 structure and network architecture. Overall, 24 paths remain in the final SEM model. Squares
 represent measured variables and arrows represent the direction of hypothesized causal
 relationships among variables. Interaction effects between predictors are shown by one causal
 path intersecting another causal path (with an open circle shown at the junction). Values
 along numbered paths are standardized partial regression coefficients. The width of each
 path is proportional to the absolute value of the standardized path coefficient. Blue vs. red arrows
 represent significant ($P < 0.05$) positive vs. negative paths, respectively. Dotted lines
 represent non-significant paths that were retained in the final best-fit SEM model. The
 lighter shade of blue or red arrows represent the path coefficients that do not deviate
 significantly from null SEM expectations ($P > 0.05$; see Fig. 5). The adjusted R^2 values
 within the squares show the explained proportion of variance in the response variables.

**Figure 3.** Modelled effects of island area and proximity to edge on community structure and

network architecture. Plots show the partial effects of area and proximity to edge after

controlling for other predictor variables (by setting to mean values). Symbols represent the

observed values on the 41 islands and shading within symbols indicates deviation of observed

values from null model expectations (based on 1000 random draws from the mainland

reference network; see Methods for details). Open symbols: observed values within the 95%

confidence interval (CI) of null model predictions; shading in the upper half of symbols:
observed values greater than the 95% CI of null model predictions; shading in the lower half
of symbols: observed values lower than the 95% CI of null model predictions.
739740

**Figure 4.** Summary of direct, indirect, and total effects of island attributes on (a) plant-

pollinator community structure and (b) network architecture. Errors bars represent the 95%

confidence interval (CI) of effects.

**Figure 5.** Observed standardized path coefficients for the SEM, compared with null model
 estimates based on 1000 random draws from the mainland reference pool. Numbered circles
 corresponded to the paths in Figure 2. Notation: DA: decreased area; PE, proximity to edge;
 DM: distance to mainland; FR: floral resources; PL: plant richness; AB, pollinator abundance;
 PO: pollinator richness; RC: relative connectance; NE: nestedness; MO: modularity; RO:
 robustness to extinction. Blue and red arrows represent positive and negative path

coefficients, respectively. Yellow shading in the probability distributions represents the 95%
confidence interval (CI) of path coefficients in the null SEM, and the red shaded areas are the
zones outside the 95% CI. The red vertical lines indicate the position of observed path
coefficient if it is significantly different from null expectation, while blue vertical lines
indicate no significant deviation from null expectation. *, $P < 0.05$; **, $P < 0.01$; ***, $P <$
0.001.

Author Rebuttal to Initial comments

Reviewer #1 (Remarks to the Author):

The manuscript by Pen Reng et al "Forest edges increase pollinator network robustness to extinction with declining forest area" is an important piece of work and very well worked out.

I suggest to publish this work after minor revisions as high priority.

R1.01 Thank you for the positive comments.

Cover of dense and diverse forests are declining as do pollinator diversity. Compared to other insect taxa, the majority of pollinators prefer open habitats and not dense, shady forests, which makes it interesting to study the role of forest ecosystems for pollinator diversity and community complexity using bipartite ecological networks. This study is one of the very few available studies showing how forest edges shape pollinator networks pointing to the importance of these ecotones for pollinating insects. The study is well designed with a high number of replicates (although it is not justified why data are pooled) and well analysed. In general, results are clearly presented and discussed.

R1.02 Thank you for these points.

In terms of justifying the pooling of data, we now describe in more detail how and why the 3 years data were combined:

"Using the combined data from multiple plant-pollinator surveys per transect, we assembled quantitative interaction networks (with the 3 years of data combined) for each island using the edge and interior transects separately. That is, on each island, we first pooled the 20 surveys of each transect over 3 years, then assembled two separate interaction networks for the edge and interior networks of each island (or mainland) location. Pooling across the 3 years was necessary to obtain relatively complete network data on each island (Table S1; Fig S1), covering the full phenological period of interaction of plants and pollinators, while reducing noise due to stochastic artefacts of adverse local conditions on particular days. The availability of relatively complete network data is an important premise for testing the effects of forest fragmentation on plant-pollination interaction networks across islands."--Lines 376-385 in the revised paper.

I only have one major point to be considered by the authors. I am sceptical about the generalisation of the results for the habitat fragmentation processes in general. I do not think you

45can transfer the findings to other ecotones or if you think you can do that, so please discuss this carefully. I suggest revising the paper and clearly focus on forest fragmentation and not habitat fragmentation in general. Currently, you mix arguments and papers of habitat fragmentation/edges with the specific habitat of forests. I would clearly separate this. For example if you fragment an extensive meadow with lots of native flowering plants to intensive agricultural monocultures e.g. maize, will you come to the same conclusions? This is not well worked out.

R1.03 This is a fair point, and quite correct that we are specifically interested in forest edge ecotones. Our key conclusion is that the effects are different to what we typically see for pollinators at non-forest edges. We have carefully checked through the manuscript and narrowed the terminology to forest fragmentation rather than habitat fragmentation. Please see Lines 46, 94, 202, 290, 385, 435, 512, 839 in the revised paper.

In addition, we also corrected other related terminology in the revised paper, such as “forest area loss” and “forest edges”. In summary, we now highlight “forest” fragmentation, specifically, throughout the revised paper, except at the start of the Introduction where it is necessary to explain the relationship to the wider subject of habitat fragmentation (Lines 58, 64, 85), more broadly.

Beside this, I am attaching more detailed comments and suggestions directly in the manuscript. I very much like how carefully you analysed your data. So good luck for a fast publication.

R1.04 Thank you for the detailed comments on the manuscript. We have listed responses to each detailed comment below in terms of the original line number, and revised the manuscript accordingly in the revised version.

Detailed comments and suggestions in the manuscript:

Line 4: I suggest to write "forest edges" as you can not conclude from your study that this is the case for different kind of habitat edges.

R1.05 Corrected, as suggested. Please see Line 4 in the revised paper.

As we explain in R1.03, we have highlighted “forest” throughout the revised paper, including “forest edges”.

Line 33: this is a very general sentence, consider to revise or delete

R1.06. We feel that this sentence is quite important to raise the contrast between the typical negative effects of edges that are frequently reported, versus the potential positive effects for

pollinators at sunny open forest edges. Therefore we would prefer not to delete it, and instead we have revised the sentence to improve the wording. Please see Line 33 in the revised paper.

Line 35: be more specific. What does "buffer interactions networks" mean? buffer in terms of what?

R1.07 In the revised paper we now clarify what we mean by a positive buffering effect against the negative consequences of forest area loss, i.e., *"We tested the hypothesis that forest edges have a positive buffering effect on plant-pollinator interaction networks in the face of declining forest area."* Please see Lines 35-36 in the revised paper.

Line 43: I can guess what community and network processes are but you should explain such terms better.

R1.08 In the revised paper, we now clarify these terms (although note that we are constrained by word limits): *"Anthropogenic forest edges benefit community diversity and network robustness to extinction in the absence of natural gap-phase dynamics in small degraded forest remnants."* Please see Lines 42-44 in the revised paper.

Line 43: forest edge, right? not sure you will find the same pattern for other edge habitats and your study only refers to forest edges.

R1.09 Yes, it is forest edge. In the revised paper, we have modified "edges" to "forest edges". Please see Line 43 in the revised paper.

As we explain in R1.03 and R1.05, we have highlighted "forest" throughout the revised paper, including "forest edges".

Line 74: what does network stability means here?

R1.10 In the revised paper, we have deleted the term "stability". Please see Lines 74-75 in the revised paper.

Line 90: forest edges not habitat edges

R1.11 Corrected, as suggested. Please see Line 92 in the revised paper.

As presented in R1.03, R1.05 and R1.09, we have highlighted "forest" throughout the revised paper, including "forest edges".

Line 109: for me you mix too much habitat fragmentation and edges with forest edges. I would start to more generally explain the fragmentation process and role of habitat edge and then come more specifically to forest edges. As forests are a special habitat for bee communities I would clearly work this out.

R1.12 As suggested, we have improved the introduction according to the logical flow from habitat loss, to fragmentation process (Lines 54-58), then the role of habitat edge in general (Line 65), then forest edge more specifically (Line 66). We highlight that forests are a special habitat for plant-pollinator communities and different from other open-habitat systems such as grasslands or croplands (Lines 63-66, 69-72).

Line 112: I would add briefly the sampling method here to see there is no sampling bias as it is difficult to compare pollinators from more open habitats with forest interiors

R1.13 In the revised paper, we have added briefly the sampling method in the mentioned sentence, i.e., *“Over 3 years we conducted 20 surveys of each pair of edge versus interior transects using time-standardised direct observations of flowers and hand-netting of insect pollinators, documenting a total of...”*. Please see Lines 114-116 in the revised paper.

Line 121: better to use "forest area"?

R1.14 Corrected, as suggested. Please see Line 123 in the revised paper.

As presented in R1.03, R1.05, R1.09 and R1.11, we have highlighted “forest” throughout the revised paper, including “forest area”.

Line 175: add "fragmentation"

R1.15 Added, as suggested. Please see Line 187 in the revised paper.

Line 240: delete "very"

R1.16 Deleted, as suggested. Please see Line 274 in the revised paper.

Line 318: this should get a justification.

R1.17 As presented in R1.02, in the revised paper, we described in more detail how and why the 3-year data were combined:

“That is, on each island, we first pooled the 20 surveys of each transect over 3 years, then assembled two separate interaction networks for the edge and interior networks of each island (or mainland) location. Pooling across the 3 years was necessary to obtain relatively complete network data on each island (Table S1; Fig S1), covering the full phenological period of interaction of plants and pollinators, while reducing noise due to stochastic artefacts of adverse local conditions on particular days. The availability of relatively complete network data is an important premise for testing the effects of forest fragmentation on plant-pollination interaction networks across islands.”--Lines 378-385 in the revised paper.

Line 400: This was said before. Consider to delete.

R1.18 Deleted, as suggested. Please see Line 491 in the revised paper, we have deleted the repeated definition of “Forbidden links”.

Detailed in the References:

Line 465: "Nature Communications" not "Nature communications"

R1.19 Corrected, as suggested.

Line 468: "Science Advances" not "Science advances"

R1.20 Corrected, as suggested.

Line 480: "Biological Reviews" not "Biological reviews"

R1.21 Corrected, as suggested.

Line 508: chance to "University Press"

R1.22 Corrected, as suggested.

Line 532: change to "Evolution"

R1.23 Corrected, as suggested.

Line 551: Do not use capital letters in the title

R1.24 Corrected, as suggested.

Line 625: do not use capital letters in the title and write out American Naturalist

R1.25 Corrected, as suggested.

Line 646: Conservation with a capital letter

R1.26 Corrected, as suggested.

Line 666: use capital letters for the journal and in general carefully format your references according to the journal style

R1.27 We have carefully checked all journal names in the references and capitalized the first letter of each word in journal names. Also, we have carefully formatted the references according to the journal style.

Line 673: this is a different format. I stop commenting on the references as there are too many inconsistencies.

R1.28 Thank you for the detailed comments on the references. Throughout the revised paper, we have carefully formatted all references according to the journal style.

Detailed in the Figure:

Line 713: do not use a combination of green and red

R1.29 We have modified the red colour to blue colour in Figure 1.

Line 723: space is not correct here

R1.30 The redundant space has been removed. We have also carefully checked the rest of the manuscript to correct similar problems. Thank you for the detailed suggestions.

Reviewer #2 (Remarks to the Author):

Dear Editor,

Regarding the manuscript "Edges increase pollinator network robustness to extinction with declining forest area":

This is an interesting study, based on extensive field sampling, showing how island size and edges affect pollinator network. It has the surprising conclusion that edges buffer the pollinator networks against extinction, by showing, among other things, that edges have greater plant and pollinator richness and network nestedness. The manuscript is well written and the methods are explained in sufficient detail.

R2.01 Thank you for the positive comments.

However, I have several concerns related to the methods and the interpretation of the results, and I think that these issues must be addressed prior to publication:

- If I understood correctly, the "interior" transects were not located in the interior, but instead were placed perpendicular to the edge, thus sampling from the edge to a maximum distance of 100 m. I don't think that this is a valid method of sampling forest interior.

R2.02 We apologize for the lack of clarity in explaining the rationale underpinning our layout of edge and interior sampling transects. We have now clarified the rationale more explicitly at Lines 320-329:

"We established paired transect lines (100 × 4 m), with one along the edge (hereafter called the 'edge transect') and one extending perpendicular from the edge into forest interior at each site (hereafter called the 'interior transect'). The interior transects penetrate all the way into the centre of small islands (<10ha) and most of the way into the centre of even the largest islands (≥10ha) because of the mountainous topography of the region and complex island shapes (Fig. 1). In this study, we were interested in partitioning the effects of both island area and distance from forest edge, so we could not sample interior transects only at the discrete 'core' of each island because this would have created a confounding bias as interior distance from edge would intrinsically increase from small to large islands."

The reviewer is correct that our approach does not facilitate sampling at very large distances from the edge, but we would point out that the topography of the region is mountainous and the island shapes are complex (see Fig. 1 in the manuscript, and Fig. R.1 below) such that ‘edge to interior’ transects do actually penetrate all the way into the center of small islands (<10ha) and most of the way into larger islands.

We would also point out that any potential limitations of sampling ‘edge to interior’ transects (rather than strictly ‘core’ habitat) will only make our findings more conservative, not less conservative. Sampling both near-to-edge distances and far-from-edge distances within our edge-to-interior transects will somewhat dampen differences that would be expected at greater distances into the interior. Yet, we still found dramatic edge influence on pollinator networks. We now raise this important point at Lines 329-333, where we state:

“We would also point out that any potential limitations of sampling ‘edge to interior’ transects (rather than strictly ‘core’ habitat) will only tend to make our findings more conservative, because partial sampling of near-to-edge distances within our interior transects will somewhat dampen differences that would be expected at greater distances into the interior.”

Fig. R.1. The topography at the study site is mountainous, and flooding of the valley produced complex island shapes (see also Fig. 1 in the main manuscript).

- Interior transects were sampled at a faster pace than edge transects. I think that this may have resulted in lower detection of flowering plants and of pollinators in the interior transects.

R2.03 Thank you for this point. Once again we failed to give an appropriate rationale for the approach in the initial version of the manuscript. In the revised paper, we have added a sentence to direct readers to a detailed justification, i.e., “A detailed rationale of the difference in survey

effort between edge and interior transects is provided in Appendix S2 (Supplementary Results 2), and any potential bias in survey effort is fully accounted for in statistical analyses.” Please see Lines 347-350 in the revised paper. Given space constraints in the manuscript we feel that it might be a suitable solution to add a new section to the supplementary materials to address this point (i.e., Appendix S2: Supplementary Results 2). However, we are happy to revisit this point if the Editor and reviewer prefer the detailed explanation within the manuscript itself.

Detailed explanation:

In dense forested systems, such as this, the abundance of flowering plants in the interior is much lower than at the edge, so the issue is actually the reverse of what the reviewer indicates. For the same level of survey effort, the likelihood of detecting all the flowers along the transect is higher in the interior (not lower), and the available observation time per flower is actually greater in the interior (because there are relatively few flowers).

Please see Table S1 in the Supplementary information, which shows that floral resources were approximately 18.52 times higher at the edge than in the interior, and pollinator abundance approximately 18.43 times higher at the edge. These data are from 116 pairs of edge vs interior transects surveyed 20 times each over 3 years, using the 15-minute survey interval for edge transects and 10-minute survey interval for interior transects. So, the 1.5 times difference in survey interval (i.e., 15 vs 10 minutes) cannot account for the massive difference in observed detections. Furthermore, we can show this quantitatively using sample accumulation curves (Fig. R.2), scaled by number of survey hours on the x-axis, emphasizing the dramatically lower encounter rate of flower-pollinator interactions per hour of cumulative survey time in the island interior. The potential bias that the reviewer refers to would occur if the accumulation curves for edge and interior transects completely overlapped, and total observed richness was lower at the interior site simply because of lower sampling effort. Fig. R.2 shows that this is not the case, as standardization by survey time still shows a dramatic edge versus interior difference. In the revised paper, we have added a sentence in the Results section to stress the dramatically lower encounter rate in the island interior even after time-standardisation, i.e., “*Floral resources, pollinator abundance, and plant and pollinator richness at forest edges were up to 10-fold higher than in the interior (Table S1), even after time-standardisation of flower-pollinator interactions per hour to account for differential survey times per transect (Fig. S2).*” Please see Lines 124-126 in the revised paper.

Logistically, we felt that it was more important to survey all 116 pairs of transects in a rapid timeframe (i.e., within 2 weeks) and then repeat these surveys at many intervals through time (i.e., 20 times intervals) in order to cover phenological variation in flowering, rather than conducting longer or more intensive local surveys. Therefore, the shorter 10 minutes survey time for interior transects, with lower flower density, facilitated better phenological coverage of our sampling throughout the season.

It is important to stress that we fully realized, and accounted for, the difference in survey effort in all relevant analyses, by using null models (Line 463, i.e., “Null model for community structure and network architecture” in the revised paper) that standardise for non-random assembly trajectories within edge or interior transects as a result of forest area loss and fragmentation, over and above any stochastic effects due to confounding bias in sampling effort between islands (Lines 510-513 in the revised paper).

Fig. R.2. Sensitivity test of differing transect survey times at interior versus edge sites. Sample-time standardised interpolation (solid line segments) and extrapolation (dotted line segments) of estimated species richness (with shaded regions for 95% confidence intervals) are completely non-overlapping between edges and interiors of the 41 islands, for both: (a) plant species and (b) pollinator species. The dotted lines denote the extrapolated sample accumulation curves when the sampling intensity is twice the actual sampling intensity. The filled circles represent the total observed plant/pollinator richness.

- I have concerns regarding the number of paths in the structural equations used and whether the sampling size is sufficient for such a complex model.

R2.04 Yes, we were careful to look at this same point specifically, and we are confident that the sampling size is sufficient for the piecewise SEM model. In the revised paper, we added the sentence (Lines 442-445) “*Moreover, local estimation allows greater robustness in fitting smaller data sets*”⁷⁵, and we follow the recommendation of Grace *et al.* (2015)⁷⁶ in ensuring that we have more than five samples per variable estimated in the model.”

Detailed explanation:

In piecewise SEM, the path diagram is translated to a set of linear (structured) equations, which are then evaluated individually. The switch from global estimation, where equations are solved simultaneously, to local estimation, where each equation is solved separately, allows for the fitting of a wide range of distributions and sampling designs (Shipley 2000, 2009). It also, in theory, permits the fitting of smaller data sets, since there only need be enough degrees of freedom to fit any given component model (Lefcheck 2016).

So, as the reviewer rightly points out, the hypothesized model should be devised beforehand and be used to inform data collection, to ensure sufficient replication from the start. As a general rule, Grace *et al.* (2015) proposed that the ratio (d) of the total number of samples to the number of variables should not fall below $d = 5$. In our study, our models involve 82 networks and 11 parameters, where there is a maximum of 6 predictors per component model, i.e., $RC \sim Area + edge + Area:edge + PI + Po + Po_ab + (1 | site)$. So, our models follow the recommended rule by Grace *et al.* (2015), i.e., $d=82/6=13.67 > 5$.

In addition, in our model fitting, we use the *semEff* package (Murphy 2021) to calculate the confidence intervals for each path using bootstrapping techniques (Table S5), which provides an accurate coverage across a range of bootstrap sampling distributions, i.e., “*Direct, indirect, and total effects for the SEM were calculated using the semEff package, version 0.6.0*”⁷⁹, with effect sizes adjusted for multicollinearity among predictors⁸⁰. The 95% confidence interval for effects was calculated using 1000 bootstrapped estimates for each response.” (Lines 457-460 in the revised paper).

-Grace JB, Scheiner SM, Schoolmaster Jr DR. Structural equation modeling: building and evaluating causal models. *Ecological Statistics: From Principles to Applications*, (eds G.A. Fox, S. Negrete-Yankelevich & V.J. Sosa), pp. 168-199 (2015).

- Lefcheck JS. PIECEWISESEM: Piecewise structural equation modelling in R for ecology, evolution, and systematics. *Methods in Ecology and Evolution* 7, 573-579 (2016).
- Murphy M. semEff: Automatic calculation of effects for piecewise structural equation models. R package version 0.6.0. (2021).
- Shipley, B. A new inferential test for path models based on directed acyclic graphs. *Structural Equation Modeling*, 7(2), 206-218 (2000).
- Shipley B. Confirmatory path analysis in a generalized multilevel context. *Ecology* 90, 363-368 (2009).

- In addition, stepwise model selection based on p-values has been criticized and may lead to inconsistent results and has the problem of multiple testing.

R2.05 We now follow the reviewer's suggestion of using AIC-based comparisons of candidate models rather than stepwise selection based on p-values (see, for example, Line 132 in the revised paper). This does not result in any changes in the models selected.

So, in the revised paper, we have improved related descriptions by replacing “with non-significant effects ($p > 0.05$)” with “*We selected a 'final' SEM by sequentially removing model predictors (direct paths) with the lowest AIC value until all remaining paths were significant and the 'global' SEM p-value was non-significant (i.e., no remaining 'missing' paths).*” Lines 454-457 in the revised paper.

Please note that the basis of the piecewiseSEM approach is founded on a carefully constructed set of ‘independence claims’ within the causal model structure, and is explicitly not an all-subsets ‘dredge’ of every possible path among predictors (see Fig. 2 and Appendix S1: Supplementary Methods 4).

Note also, that (unlike many other piecewiseSEM studies) we also validate the reliability of the models with the lowest AIC values using null model testing, and bootstrapped confidence intervals for all paths coefficients (Table S5); i.e., “*Direct, indirect, and total effects for the SEM were calculated using the semEff package, version 0.6.0⁷⁹, with effect sizes adjusted for multicollinearity among predictors⁸⁰. The 95% confidence interval for effects was calculated using 1000 bootstrapped estimates for each response.*” (Lines 457-460 in the revised paper).

- I am not sure whether the way the reference pool was made, using only mainland transect, was appropriate, considering the relatively low sampling size in the mainland.

R2.06 There are two slightly different concepts tied up in this question. One is whether the level of sampling is representative of the whole community, and the other is the conceptual basis for choosing a particular reference pool for null model comparisons.

Sampling size:

Our mainland sampling size might appear lower overall (in a relative sense) when comparing 16 mainland pairs of transects against 100 pairs in total of all island transects. However, the mainland is highly representative in terms of (1) containing a larger assemblage and network size than any of the islands (mainland: 54 plant species and 269 pollinator species; largest island B01: 47 plant species and 264 pollinators; Table S1); and more importantly (2) the exceptionally high level of sampling completeness of the mainland reference pool at our level of sampling effort. The sampling completeness of plants and pollinators on the mainland were 0.92 and 0.99, respectively (sample coverage values in Appendix Fig. S1; displayed below). This representativeness illustrates that the sampling size is adequate to make strong inference about assemblage composition.

Figure S1. Estimated sample coverage values for plants, pollinators and their interactions after 20 surveys per site over 3 years. Black symbols are for the 41 islands and the dashed red line represents the 16 mainland reference sites (combined).

It is also worth pointing out that there were logistical constraints on the ability to sample the mainland sites at the same overall intensity as the island sites (to our knowledge, no study has ever collected reference samples for plant-pollinator networks with this degree of intensity; 100 pairs of transects). We were also mindful of restricting the degree of impact on the environment from potentially over-sampling at mainland reference sites.

Conceptual basis for reference pool selection:

The second issue is about the appropriate reference pool to use for null model selection. Null model selection is acknowledged to be contentious in Ecology, but if there is one point of agreement it is that selection should be targeted and specific to the ecological question that is being addressed. Here, we are not only interested in controlling for sample effects (number of individuals), but we are also interested in determining non-random network assembly trajectories. In this regard, we follow the principle of Gotelli and Graves (1996) that “A null model is a pattern-generating model that is based on randomization of ecological data or random sampling from a known or specified distribution...[...].The randomization is designed to produce a pattern that would be expected in the absence of a particular ecological mechanism”. In our case, we are interested in the null pattern of community diversity and network structure that would be expected in the absence of fragmentation and reduction in island area. We do not have ‘pre-fragmentation’ data to directly test re-assembly trajectories through time on each island, therefore our best reference state for the null draws is the combined set of sampling plots from the adjacent ‘unfragmented’ mainland. The reference pool has more species and more individuals than any of the individual islands, so we are not drawing a larger sample from a smaller pool, as might have been implied.

By comparison to our approach, using the entire regional pool (both mainland and islands) for the random draw would violate the notion of generating null expected patterns in the absence of the ecological mechanism of interest, because the isolated/fragmented island transect samples would have been contained within the ‘regional pool’ as well.

In hindsight, we acknowledge that we did not explain this clearly enough in the methods, and only made a loose statement about the purpose of the null model and sampling effects. We have now corrected this oversight by adding the explanation above, at Lines 469-477 in the methods section, “*In selecting the reference pool, we follow the principle of..... the adjacent ‘unfragmented’ mainland is the most appropriate reference pool available for the null draws.*”

Gotelli, N. J., & Graves, G. R. (1996). *Null models in ecology*. Smithsonian Institution Press, Washington DC, USA.

- The conclusions speak of edges buffering the islands from the negative impacts of habitat loss, but I don't think that such conclusions can be made without looking at species identity and whether, for example, the edges are occupied mostly by generalist species. Increased species richness at edges is a relatively common pattern, but it is often due to generalist, invasive and/or alien species. In addition, there is still the issue of differences in sampling pace between edge and interior transects.

R2.07 Many thanks for your comments. With regard to sampling pace along the transect, please see R2.03 above.

With respect to species identities and the frequent observation of generalist invasive/alien species at edges, this is an excellent point and something that we have now addressed explicitly below and incorporated into our discussion.

After some thought, we feel that the most efficient way to represent the relative specialisation vs generalisation of plant and pollinator interactions at edges vs interiors is to create meta-networks ordered according to network nestedness. We show the complete system-level meta-network by combining edge networks and interior networks of all 41 islands (Fig. R.3a) as well as the individual island-level meta-networks by combining the edge network and the interior network of each island separately (Fig. R.3b), using a basic checkerboard diagram. In the checkerboard diagram, rows and columns denote plant species and pollinator species, respectively. The red, dark blue and light blue boxes represent the interactions occurring only at the island edge, only in the island interior, and at both edge and interior, respectively. Species in the checkerboard diagram are arranged from low specialization to high specialization (i.e., generalization decreases). From both a system-level and an island-level meta-network perspective, the results in Fig. R.3 show that most species in interior networks are not more specialized than species at edges, while the edge network simultaneously contains many specialists as well as generalists (not just one or the other). That is, island edges, in addition to having higher species richness, also contain highly specialized species.

In the revised paper, we put Fig. R.3 in the Appendix as Figure S4 (Appendix: Lines 569-578) and related results in Appendix S2: Supplementary Results 1 (Appendix: Lines 499-510). We amended the description of results in the abstract (Lines 40-42, "*Edge networks contain highly specialized species, with higher nestedness and lower modularity than interior networks,*

60maintaining high robustness to extinction following area loss...”), the Results section (Lines 129-131, “Importantly, though, edge communities were not predominantly made up of generalist species and species in interior network were not more specialized (Supplementary Results 1; Fig. S4).”), the Discussion section (Lines 195-199, “Edges maintained 10-fold higher pollinator abundance and richness than forest interiors, and these effects were not driven by increasing frequency of generalist plant-pollinator interactions at edges. Edge networks contained many highly specialized species, even compared to interior networks.”). As well, we added the Method section in Appendix S1: Supplementary Methods 3 (Appendix: Lines 81-91, “Furthermore, a basic checkerboard diagram is used to visually show how the relative specialisation vs. generalisation of plants and pollinators varies between edges vs. interiors. In the checkerboard diagram, rows and columns denote plant species and pollinator species, respectively. The red boxes represent the interactions occurring only at the island edge, dark blue boxes represent interactions occurring only in the island interior, and light blue boxes represent interactions occurring at both. Species in the checkerboard diagram are arranged from low specialization to high specialization (i.e., generalization decreases). We show the complete system-level meta-network by combining edge networks and interior networks of all 41 islands as well as the individual island-level meta-networks by combining the edge network and the interior network of each island separately. We draw the checkerboard diagram using R package bipartite, version 2.16¹⁸”).

(a) System-level meta-network

(b) Island-level meta-networks

Fig. R.3. Network nestedness for (a) the system-level meta-network combining all edge and interior networks of all 41 islands, and (b) 41 island-level meta-networks combining the edge network and the interior network of each island. Rows represent flowering plant species, columns represent insect pollinator species. Species towards the upper left of each meta-network are increasingly “generalist” in their species interactions (see inset diagram to the right of island meta-network B02). Red boxes represent interactions occurring only at the island edge, while dark blue boxes represent interactions occurring only in the island interior, and light blue boxes represent interactions occurring at both the edge and interior. The size of the checkerboard is proportional to the size of the network.

- In figure 3, area and edge effects on network robustness are quite small. Thus, although there is a cascading effect apparent from the SEM, the effects on network robustness do not appear to be strong enough to justify the study's conclusions.

R2.08 While the effect sizes for network robustness are ‘smaller’, in a relative sense, than the other effects in Fig. 3 (now Fig. 4 in the revised paper), they are nevertheless substantive in their own right, and ecologically important. Our conclusion is not just ‘whether’ there is an effect, but that the ‘direction’ of effect is actually the opposite of what is found in non-forest systems. In other words, anthropogenic edge effects are considered to exacerbate the effects of area loss on pollinators, but here we show that forest edges not only dampen the response but completely reverse the sign of the effect. At Lines 195-201 in the Discussion section, we write “*Edges maintained 10-fold higher pollinator abundance and richness than forest interiors, ... More importantly, edges increased network robustness to extinction, rather than exacerbating the negative effect of forest area loss on network architecture, thereby reducing the risk of plant-pollinator system collapse.*”

This is why we feel the conclusion for network robustness is so important.

We feel justified in calling this a ‘buffering’ effect of edges because of the statistically significant interaction effect between decreasing area and increasing proximity to edge. In Fig. 4, this is reflected in the diverging slopes of the fitted lines as area decreases. On large islands, there is ‘only’ a 10-20% higher robustness to extinction at edges compared with interiors, whereas on small islands there is a 50-100% higher robustness to extinction at edges compared with interiors (Fig. 4). True, this is not as great as the 10-fold greater values for richness and 2-fold higher values for measures of network architecture, but we feel these values are substantial nonetheless.

In light of these important comments, we have made additional effort to explain the importance of the findings more clearly.

At Lines 180-182 in the results section, i.e., *“On large islands, there was ‘only’ a 10-20% higher robustness to extinction at edges compared with interiors, whereas on small islands there was a 50-100% higher robustness to extinction at edges compared with interiors (Fig. 4h).”*

At Lines 263-268 in the discussion section, i.e., *“Although our effect size for edge influence on network robust is relatively small, the direction of effect we find in our forest system is actually the opposite of what is found in non-forest systems. Anthropogenic edge effects are typically considered to exacerbate the effects of forest area loss on pollinators, but here we show that forest edges not only dampen the response but completely reverse the sign of the effect.”*

Additional comments below:

Abstract

There was no mention on whether the species occurring at forest edges are the same as those in forest interior. This is essential information: it is possible that pollinator networks at edges are more robust because they are dominated by generalist species, with specialists being lost. So, even though these networks are robust, they may not represent the networks in interior forest.

R2.09 As detailed in R2.07 above, we have now investigated this point using a system-level meta-network (Fig. R.3a) as well as separate island-level meta-networks (Fig. R.3b), which show that edges are not more dominated by generalist species, but in fact have many highly specialised species (more so even than the interior sites). Please see manuscript revisions indicated in response R2.07.

"We test" should be "We tested"

R2.10 Corrected, as suggested.

In some parts of the text, it seems that "habitat fragmentation" actually refers to "habitat loss and fragmentation", which are two related but different processes - for example, decreased area is habitat loss but not fragmentation.

R2.11 In the revised manuscript, we have modified "habitat fragmentation" to "forest area loss and fragmentation" on Lines 435, 512.

Regarding the methods:

The differences between pace and survey interval at edge and interior transects might have resulted in decreased detection probability of flowering plants in the interior transects.

R2.12 In the revised paper, we have added a sentence for this question, i.e., “*A detailed rationale of the difference in survey effort between edge and interior transects is provided in Appendix S2 (Supplementary Results 2), and any potential bias in survey effort is fully accounted for in statistical analyses.*” (Lines 347-350 in the revised paper).

As explained in detail in R2.03 above, we do not believe that this would have been the case based on time-standardised accumulation curves for edge and interior transects (Fig. R.2). We show this quantitatively using sample accumulation curves, scaled by number of survey hours on the x axis, emphasizing the dramatically lower encounter rate of flower-pollinator interactions per hour of cumulative survey time in the island interior. Logistically, we felt that it was more important to survey all 116 pairs of transects in a rapid timeframe (i.e., within 2 weeks) and then repeat these surveys at many intervals through time (i.e., 20 times intervals) in order to cover phenological variation in flowering, rather than conducting longer or more intensive local surveys. Therefore, the shorter 10 minutes survey time for interior transects, with lower flower density, facilitated better phenological coverage of our sampling throughout the season.

Site description:

I suggest including information on whether the herb and shrub layers at edges are composed mostly of generalist species or not. I also missed information regarding slope and the different in altitude between edge and interior - as the islands are former hilltops, there may be important differences in altitude.

R2.13 *Species composition:* The herb and shrub layers at edges are not composed mostly of generalist species (newly added on Lines 309-311), i.e., “... *whereas at the island edge, the understory shrub and herb layers are more dense and diverse, simultaneously containing many specialist species as well as widespread generalists.*”. We include a full list of plant species in Table S2, showing that there are many light-loving species that provide flowering resources only at the edge but not the interior, whereas all plant species found in interior transects are also found in edge transects.

65As analyzed in response R2.07 above, Fig. R.3 shows that whether considered from the angle of either the system-level meta-network or island-level meta-networks, most species in the interior network are not more specialized, while the edge network simultaneously contains lots of specialists and generalist (not just one or the other). That is, island edges, in addition to having higher species richness, also have more specialized species.

Altitude and slope: The islands are indeed former hilltops, but the altitudinal range is not large enough to be a major factor structuring gradients in plant community composition (as would occur in other study systems where sites vary across hundreds of meters of elevation). The average difference in elevation between edge transect and interior transect on each island varied from just 6.80-48.60 m across sites. In the revised paper, we have added this information in a new column in Table S1 (**Appendix**). In our study system, it is unavoidable that the forest edge is at the lake edge and has lower elevation than the interior of the island, but the most parsimonious explanation for edge to interior differences in pollinator networks is floral resource availability at edges *vs* interiors, and not the small average difference in elevation.

Sampling design:

The method described is "paired edge versus interior sites", including 41 islands and 16 mainland sites. However, the transects were 100 m long, one along the edge and another *_perpendicular_* to the edge - so, as I understand, the perpendicular transect would span from the edge to 100 m into the forest. I see two problems here: first, this is not edge versus interior sites, which would require a transect at the edge and another at a certain distance from the edge; second, edge effects may extend more than 100 m into the forest, so the "interior" transects may also be subject to edge effects.

R2.14 As detailed in our response R2.02 above, the main reason for our sampling design is that we had to take into account variation in both island area and distance from forest edge. If we had only sampled the discrete 'core' of each island versus the 'edge', then the distance to the core would naturally get larger and larger as island area increased. This would create a spurious confounding effect between island area and distance to edge, and we would have been unable to disentangle relative edge and area effects. As pointed out in R2.02, 'edge to interior' transects do actually penetrate all the way into the center of small islands (<10ha) and most of the way into larger islands.

In terms of any effects that this sampling design might have on the results, we would point out sampling 'edge to interior' transects rather than strictly 'core' habitat will only make our findings more conservative, not less conservative. So, while the reviewer is correct that in some cases edge effects have been observed to extend more than 100m into the forest, our sampling at both near-to-edge distances and far-from-edge distances within our edge-to-interior transects will tend to dampen differences that would be expected at greater distances into the interior. Therefore, we are not likely to be over-extrapolating any effects, but rather underestimating them, if anything.

The pace and survey interval differed between edge and "interior" transects, with the "interior" transects being sampled at different rates. The reason given for this is that there were more flowering plants at edges, demanding longer survey intervals. Although this makes some sense, this may also have resulted in lower detectability of flowering plants and of pollinators in the "interior" transects.

R2.15 As detailed in our response R2.03 above, in dense forested systems, such as this, the abundance of flowering plants in the interior is much lower than at the edge, so the issue is actually the reverse to what the reviewer indicates. For the same level of survey effort, the likelihood of detecting all the flowers along the transect is higher in the interior (not lower), and the available observation time per flower would actually be greater in the interior. In either case, the comparison of time-standardised accumulation curves in response R2.03 above shows that the difference in survey effort per transect can not account for the observed findings.

Is it possible that phenology differed between edge and interior, for example with peak flowering in the interior happening after peak flowering at the edge? If so, could this have influenced the results?

R2.16 This is an interesting point, and it is certainly possible for there to be phenological differences depending on varying environmental conditions between edge and interior, but we did not observe this to be the case in our data. Moreover, we sampled all island and mainland sites within multiple (6 or 7) consecutive 2-week survey rounds covering the entire flowering season each year (from early spring to late autumn), and we randomized the sequence in which transects were sampled, so this would have overcome any differences in phenology within or between sites. For example, the sequence of sampling particular islands was randomised, as was the sampling time period during either the morning or afternoon, and whether starting at the edge or interior. In the revised paper, we have added additional explanation in the Methods section,

i.e., “*We randomized the sequence in which transects were sampled during the 2-week sampling periods, in order to overcome potential differences in phenology within or between sites.*” (Lines 356-358).

"We considered an insect to be a putative pollinator only if it was touching the anthers and/or stigmas of the flowers." - this is appropriate, but I'm wondering about detectability. Was the observation time sufficient to ensure that pollinators actually touched the anthers, especially considering the difference in pace between edge and interior?

R2.17 From our observations, when an insect was noted approaching a flower it almost invariably moved rapidly to the anthers and/or stigmas (please bear in mind that these are relatively warm subtropical forests). Insects wandering slowly across the leaves or flower petals, not touching the anthers and/or stigmas, were not included in counts of pollinators. Despite this fairly conservative approach, we believe the observation time was sufficient to generate a robust dataset, and exclude transient ‘tourists’. More importantly, the observation time at the site level was consistent, so data from each island or mainland site are directly comparable. We addressed earlier the issue of the 15-minute vs 10-minute survey time for edge vs interior transects, respectively, and note again that it is the 18-fold greater floral abundance at edges that drives difference in pollinator insect abundance, and not differences in detectability due to sampling methodology.

We modified the text in the manuscript slightly to read: “*We recorded an insect as a putative pollinator only if it was actually touching the anthers and/or stigmas of the flowers.*” Please see Lines 361-362 in the revised paper.

"we estimated the flower area of each plant along each transect at each sampling period" - How was this done? It is not clear to me what you mean by "flower area" and it is not clear how estimates were made along 100 m-long transects.

R2.18 Our apologies for the lack of a clear explanation. We now replace the term ‘flower area’ with ‘flower resource availability’ (at Line 366 in the Manuscripts), and we include a detailed explanation in the Supplementary information (at Lines 27-56 in the Appendix S1: Supplementary Methods 1).

We have no direct measure of pollen or nectar resources available per flower on different plant species, but we expect that greater numbers of flowers would equate to greater resources, and that larger flowers would have more resources than smaller flowers (all other things being equal). Therefore, we calculated species-specific estimates of the number and size (two-dimensional area) of flowers per individual plant per species, and summed these across the transect to obtain an estimate of total flower resource availability. Although this measure will not be accurate in an absolute sense, we expect that it will scale proportionate to increasing floral resources across sites.

Estimating flower abundance and size (estimation methods differ for different plant species): (i) For herbs with dense flowering (e.g., *Erigeron annuus*, *Cnidium monnieri*, *Sinosenecio oldhamianus*), the ground cover area of the whole plant was estimated as the flower area by observation; (ii) For plants with sparse flowers (e.g. *Potentilla kleiniana*, *Lysimachia congestiflora*, *Hypericum sampsonii*), we projected the two-dimensional area of the petals onto a circle and approximated the area of the circle as the flower area (i.e., floral display size). This method also applies to plants with large flowers (such as *Rosa laevigata* and *Rosa bracteata*), that were straight-forward to count and measure flower area. This method is reasonable, since flower display size is related to pollinator attraction. (iii) For plants with long inflorescences (e.g., *Buddleja lindleyana*, *Leonurus artemisia*) or vines (e.g., *Millettia dielsiana*, *Wisteria sinensis*), the total inflorescence size was estimated as the surface area of a cylinder based on the radius and height of the inflorescence. (iv) For densely-flowering shrubs with small flowers (e.g., *Vitex negundo*, *Symplocos paniculata*, *Photinia serrulata*), the crown surface area was estimated as the flower area, considering the crown of the shrub as a convex hull such as a sphere, cylinder or cone, as appropriate.

In a single survey, the observed floral display area of plants was estimated per species and then accumulated to obtain floral resource availability per transect, and then finally summed across multiple surveys. Therefore, the summed floral resource across multiple surveys represents the relative size of floral resource on the mainland and each island.

Adedoja O, Dormann CF, Kehinde T, Samways MJ. Refuges from fire maintain pollinator-plant interaction networks. *Ecology and Evolution* 9, 5777-5786 (2019).

Sutter L, Jeanneret P, Bartual AM, Bocci G, Albrecht M, Macivor S. Enhancing plant diversity in agricultural landscapes promotes both rare bees and dominant crop-pollinating bees through complementary increase in key floral resources. *Journal of Applied Ecology* 54, 1856-1864 (2017).

Vazquez, Diego, P., Dorado, Jimena. The diversity-stability relationship in floral production. *Oikos* 123, 1137-1143 (2014).

Weiner CN, Werner M, Linsenmair KE, Blüthgen N. Land-use impacts on plant-pollinator networks: interaction strength and specialization predict pollinator declines. *Ecology* 95, 466-474 (2016).

Finally, if larger islands had more transects sampled, wouldn't these differences in sampling effort be confounded with area effects? This should be discussed and justified.

R2.19 Potential bias due to differences in sampling effort is always an important issue, and for this reason we went to great lengths to check on the level of sampling completeness for each island, and presented sample coverage estimates in Fig. S1. The rationale here is that if sampling is essentially 'complete' then the observed community or network metrics are a reliable estimate of the true value (Lines 121-123, "*Sample coverage.....(Fig. S1).*"). Here, sampling completeness was high for plants, but did show lower values on small islands compared with large islands for pollinators and for interactions. For this reason, we were careful to carry out null model comparisons to distinguish true fragmentation effects from potential sampling artefacts. Throughout the results, we make every attempt to explain where effects are greater (or less than) expected from a null draw of the same-sized sample from the reference pool, and ours is the first study (to our knowledge) to compare the observed SEM effects to the expected effects from 1000 null-draw SEMs based solely on passive sampling effects the reference network.

Network metrics

The relative connectance measure is defined as the number of observed interactions divided by the number of interactions of these species in the whole region. It is not clear, however, how it would be calculated if a species is absent from a transect. For example, pollinator Po1 interacted with plant P11 in the network, but in the whole region it was observed to interact with plants P11, P12 and P13. However, plant P13 was not found in the transect. In the total number of observed interactions for Po1 2 or 3, in this case?

R2.20 Relative connectance is a proportion, so in the reviewer's example there would be two observed interactions in the local transect (Po1-P11 and Po1-P12), divided by the three known interactions in the whole region (Po1 to P11, P12, P13). To make this clearer, we now plot a figure below (Fig. R.4) to explain the calculation of relative connectance in networks on different islands.

70Fig. R.4. A hypothetical example of the process for calculating relative connectance of networks on different islands. (a) The combined networks of all observation sites (i.e., combined networks b, c, d). (b)-(d) The local networks on three hypothetical islands.

The phrase “in the whole region” in our definition is perhaps easy to misunderstand, so we now replace this with the phrase “*all observation sites*” in the revised paper. In the revised paper we now write “*Relative connectance is defined as $RC = I/S_{obs}$, the ratio of the number of observed interactions among species recorded in the network (I) to the total observed number of interactions of these species across all observation sites over the 3 years of the study (S_{obs}).*” Please see Lines 398-401 in the revised paper.

The NODF index should be explained. The modularity score and the algorithm (DIRTLPAwb+) should also be explained.

R2.21 We have added more descriptions for NODF, i.e., “*Specifically, overlap denotes complete overlap of the presences (observed interactions) from lower to upper rows and from right to left columns, and decreasing fills denotes decreasing marginal totals between all pairs of columns and all pairs of rows*”⁶⁶. (Lines 412-415 in the revised paper).

We have also added more descriptions for the algorithm (DIRTLPAwb+) and improved the sentence, i.e., “*We used the DIRTLPAwb+ algorithm to maximize a measure of modularity*”⁶⁸,

i.e., each node is randomly assigned a unique label, then at each iterating step, each node is forced to assume the label that is shared by the neighbours each node is connected to within the graph. The algorithm then partitions the nodes of the network into separate subsets or modules, finally modularity is maximized whenever each of these modules are relatively isolated from the other modules in the network.” (Lines 417-422 in the revised paper).

Structural Equation Modelling:

The hypotheses and the underlying logic make sense to me.

R2.22 Thank you for the positive comments.

Are structural equations really appropriate for "non-normal error distributions in models"? As far as I know, one of their assumptions is multivariate normality (although I think there are alternatives).

R2.23 Yes, the piecewiseSEM approach is different from a simple path analysis approach using whole-model estimation (e.g., in lavaan or AMOS), which assumes that all observations are independent, and all variables follow a (multivariate) normal distribution (Grace 2006). As Lefcheck (2016) states “piecewiseSEM extends SEM to all current (generalized) linear, (phylogenetic) least-square, and mixed effects models. Using the piecewiseSEM package, SEMs are built using a list of structured equations, which can be specified using most common linear modelling functions in R, and thus can accommodate non-normal distributions, hierarchical structures and different estimation procedures.” (Lefcheck 2016).

In the revised paper, we have improved the related sentence, i.e., “*We tested the causal structure of the hypothesized model (Fig. 2) using piecewiseSEM version 2.1.0, which extends SEM to non-normal distribution models*⁷⁵.” (Lines 445-446).

I think that figure S2 could be in the manuscript and not in the supplementary material.

R2.24 At the Editor’s discretion, we are happy to include Fig. S2 in the manuscript, although it does increase the size of the paper (now Fig. 2).

In addition, I have some concerns regarding the model complexity compared to sample size. A total of 41 islands were samples, in addition to 16 mainland transects. If I understood correctly, the edge transects at a given island were combined to form an "edge" network, as were the "interior" transects. The same was done for the 16 mainland edge transects and the 16 mainland

72interior transects. Thus, there would be a total of 82 networks, but only 42 independent sampling units. Yet there are 45 paths in the model in Figure S2.

R2.25 Please see our detailed response to this point at R2.04 above. The mainland networks were used for the null model testing not the SEM testing of fragmentation effects on network structure (since they lack attributes of area and isolation). The SEM models involve 82 island networks (edge and interior for 41 islands) and the issue of hierarchical nesting of edge and interior transects within each island is explicitly handled by the ability of piecewiseSEM to incorporate non-independence and non-normality in models. As mentioned earlier, we follow the recommended general rule of Grace et al. (2015) that the ratio of the total number of samples to the number of variables should not fall below 5.

Is the stepwise approach to calculate a "final" SEM appropriate? There is much criticism to stepwise model selection in which non-significant effects are sequentially removed (e.g. <https://besjournals.onlinelibrary.wiley.com/doi/full/10.1111/j.1365-2656.2006.01141.x?s=09>). Considering the large number of paths and the problems with multiple testing in sequential model selection, I think that this approach may have resulted in biased results. I would suggest using AIC to compare between a predefined set of ecological models or, alternatively, focusing on simpler, univariate analysis. Alternatively, it should be justified why the approach is appropriate and would not result in statistical artifacts.

R2.26 As detailed in R2.05 above, we now use AIC-based comparisons of candidate models rather than stepwise selection based on p-values.

In the revised paper, we have improved related descriptions by replacing “with non-significant effects ($p > 0.05$)” with “*We selected a 'final' SEM by sequentially removing model predictors (direct paths) with the lowest AIC value until all remaining paths were significant and the 'global' SEM p-value was non-significant (i.e., no remaining 'missing' paths).*” Lines 454-457 in the revised paper.

Note also, that (unlike many other piecewiseSEM studies) we also validate the reliability of the models with the lowest AIC values using null model testing, and bootstrapped confidence intervals for all paths coefficients (Table S5); i.e., “*Direct, indirect, and total effects for the SEM were calculated using the semEff package, version 0.6.0, with effect sizes adjusted for*

multicollinearity among predictors. The 95% confidence interval for effects was calculated using 1000 bootstrapped estimates for each response.” (Lines 457-460 in the revised paper).

Null models:

As I understand, the mainland transects were used as an "interactions pool" (which I think can be considered analogous to a species pool). However, sampling effort in the mainland was relatively small compared to the sampling effort in the islands. Wouldn't it be more appropriate to use all the data (from the mainland and the islands) as the pool of possible interactions? In addition to the sampling effort issue, it is possible that novel interactions occur at the islands which are not present in the mainland, due to novel ecological conditions.

R2.27 As detailed in R2.06 above, the level of sampling completeness of plants and pollinators on the mainland was very high, at 0.92 and 0.99, respectively (sample coverage values in Appendix Fig. S1). This indicates that samples are highly representative of the true mainland community, and that sampling effort in the mainland is sufficient to characterise the reference networks for plants and pollinators.

In R2.06 above, we also explained the conceptual rationale why the mainland sites represent the ideal reference network for null model comparisons, and not the entire dataset included both mainland and islands. We now add the explanation at Lines 469-477 in the methods section, “In selecting the reference pool, we follow the principle of Gotelli and Graves (1996)⁸¹ that “A null model is a pattern-generating model that is based on randomization of ecological data ..[...]. designed to produce a pattern that would be expected in the absence of a particular ecological mechanism”. In our case, we are interested in the null pattern of community diversity and network structure that would be expected in the absence of fragmentation and reduction in island area. Given that we do not have ‘pre-fragmentation’ data to directly test re-assembly trajectories through time on each island, the combined set of sampling plots from the adjacent ‘unfragmented’ mainland is the most appropriate reference pool available for the null draws.”

In terms of the types of interactions occurring between the mainland and island, note that the null model comparisons are testing the expected network structure in the reference network (i.e., nestedness, relative connectance, modularity etc) and not the specific interaction identities, per se. In the revised paper, we have added explanations at Lines 496-500, “Note that interaction identities are not involved in the entire process of constructing the null model estimates of network structure, therefore it is not relevant whether particular interactions occur on islands

which are not present in the mainland (or vice versa). The important issue is that sampling completeness is relatively high for the reference pool.”

Of course, if novel ecological conditions on islands resulted in a deviation in network architecture on islands compared with the null reference community, i.e., the difference between the observed value and the null model value, then this is exactly the sort of ‘fragmentation’ signal we are testing for (over and above passive sampling effects).

"If the observed values differed significantly from the null model values, this indicates that the observed values showed non-random assembly trajectories as a result of habitat fragmentation, over and above any stochastic effects due to confounding bias in sampling effort between islands." - I think that this would make sense if the mainland transects were truly representative, but I am not convinced that this is the case. Thus, I think that deviations from the null models could have arisen due to sampling error in the mainland transects, with many interactions not being detected there.

R2.28 Please see the detailed response in R2.06 above, the mainland is highly representative in terms of (1) containing a larger assemblage and network size than any of the islands (mainland: 54 plant species and 269 pollinator species; largest island B01: 47 plant species and 264 pollinators; Table S1), so we are not drawing a larger sample from a smaller pool, as might have been implied; and more importantly (2) the exceptionally high level of sampling completeness of the mainland reference pool (Appendix Fig. S1).

I think this manuscript may be of interest:

<https://www.annualreviews.org/doi/abs/10.1146/annurev-ecolsys-120213-091759>

R2.29 Thank you for this suggestion.

In Null Model I, were the transects drawn with replacement (as in bootstrap)?

R2.30 Yes, the transects were drawn with replacement, which is similar to the method of bootstrapping (i.e., in the case where a set of observations are assumed to be from an independent and identically distributed population). We randomly draw the same number of transects from the mainland as observed on each of the 41 sampled islands.

We have added the extra detail to the manuscript, i.e., “...which is similar to the method of bootstrapping (i.e., in the case where a set of observations are assumed to be from an independent and identically distributed population).” Please see Lines 484-485 in the revised paper.

Regarding Null Model II, in its stepwise version: "we constrained the reference pollinator pool to only the potential (i.e. non-forbidden) links" - did this exclude pollinators that had forbidden links with some, but not all, plants in the random network? If so, would this not somehow bias the results?

Apart from this, I don't see any problems with this null model - the algorithm seems valid and well developed, for which I congratulate the authors.

R2.31 Thank you for the positive comments. Regarding Null Model II - no, a pollinator would not be excluded (a priori) from the network if it had forbidden links with just some, but not all, plants in the random network. A pollinator would only be excluded if it had forbidden links to all plants in the random draw.

Null model SEM:

This analysis seems valid to me.

R2.32 Thank you.

Additional suggestion:

When dealing with effects of area loss and fragmentation, it is important to also look at changes in species composition. Thus, I suggest including at least a simple comparison (e.g. PERMANOVA or canonical correspondence analysis) to see how island area and edges affect species composition.

R2.33 As would be expected, given the large differences we showed in flower resources, plant and pollinator abundance and richness, and network properties between edge and interior sites, there is also a significant difference in species composition between edge and interior sites. In the earlier version of the manuscript we had been more focused on network properties, but it is indeed a good idea to present additional PERMANOVA comparisons.

Based on the sampling data for the mainland and 41 islands, the results of a non-metric multidimensional scaling (NMDS) ordination analyses show strong differentiation between edge and interior of the mainland and 41 islands, for both plants (PERMANOVA, $df = 1$, $F = 17.12$, $p < 0.001$) and pollinators (PERMANOVA, $df = 1$, $F = 13.64$, $p < 0.001$) (Fig. R.5).

Fig. R.5. Non-metric multidimensional scaling (NMDS) ordination diagrams based on three dimensions for (a) plants and (b) pollinators. Red circles with solid outlines represent island edge sites and the blue diamonds with solid outlines represent island interior sites, respectively. Red circles and blue diamonds with dashed outlines represent the mainland edge and mainland interior sites, respectively. Symbol size is proportional to the island area.

In the revised paper, we have put Fig. R.5 in Fig. S3 (Lines 562-567), added an additional sentence to the Results section to briefly explain this point, i.e., “*Species composition of both plants and pollinators differed significantly between edge and interior (PERMANOVA: plants, $df = 1$, $F = 17.12$, $p < 0.001$; pollinators, $df = 1$, $F = 13.64$, $p < 0.001$) (Fig. S3).*” (Lines 127-129). We also have added the Method section in Appendix S1: Supplementary Methods 3 (Lines 69-80, “*Species composition on the mainland and 41 islands was examined by ordination analysis using non-metric multidimensional scaling (NMDS). NMDS identifies the dominant gradients of variation in site dissimilarity in ordination space (using Bray-Curtis distance in our case) and maps the rank ordering of sites onto a predefined small number of axes ($k = 3$) in an iterative search for an optimal solution* ^{14, 15}. We used permutational multivariate analysis of

variance (PERMANOVA) to test differences in species composition between edge and interior on the mainland and 41 islands. As a non-parametric multivariate test based on distances (again, Bray-Curtis distance in our case), PERMANOVA uses permutation to test the null hypothesis that the centroids and dispersion are equivalent for all groups¹⁶. NMDS and PERMANOVA were conducted using the metaMDS and adonis2 functions in R package vegan, version 2.5-7¹⁷.”).

One more additional suggestion:

As the discussion is focused mostly on area and edge effects, are SEM truly necessary? Couldn't the study questions be answered with simple GLMMs testing for edge and area effects individually for the response variables?

R2.34 The piecewiseSEM is built from component GLMM, but the advantage of the piecewiseSEM is that it can test for cascading indirect effects from community composition to network architecture to robustness to extinction. A case in point is that the effects of edge and area on network robustness are entirely indirect, and simple GLMMs would not be able to tease out the indirect pathways of effect in the way that SEM can.

Results:

Overall the results make sense. Although there is little mention of effect sizes in the text, they can be seen in figures 2 and 3. In figures 2 and 3, I suggest changing the colors to colorblind-friendly ones. Also, I liked the shading in figure 3.

However, there is one issue. In Figure 3, there are clear differences between edge and interior for plant and pollinator richness and abundance (or floral resources), as well as for nestedness, but the differences between edge and interior for network robustness are quite small. So I am not convinced that edges actually increase network robustness, notwithstanding the indirect effects.

R2.35 Colours: We have changed the colours of figure 2, 3 to colourblind-friendly ones, i.e., using gray, dark blue, light-blue, red and light-red.

Effect sizes: As detailed in our response R2.08 above, while the effect sizes for network robustness are ‘smaller’, in a relative sense, than the other effects in Fig. 4, they are nevertheless substantive in their own right, and ecologically important. Our conclusion is not just ‘whether’ there is an effect, but that the ‘direction’ of effect is actually the opposite of what is found in non-forest systems. In other words, anthropogenic edge effects are considered to exacerbate the

effects of area loss on pollinators, but here we show that forest edges not only dampen the response but completeness reverse the sign of the effect. At Lines 195-201 in the Discussion section, we write “*Edges maintained 10-fold higher pollinator abundance and richness than forest interiors, ... More importantly, edges increased network robustness to extinction, rather than exacerbating the negative effect of forest area loss on network architecture, thereby reducing the risk of plant-pollinator system collapse.*”

This is why the conclusion for network robustness is so important.

We feel justified in calling this a ‘buffering’ effect of edges because of the statistically significant interaction effect between decreasing area and increasing proximity to edge. In Fig. 4, this is reflected in the diverging slopes of the fitted lines as area decreases. On large islands, there is ‘only’ a 10-20% higher robustness to extinction at edges compared with interiors, whereas on small islands there is a 50-100% higher robustness to extinction at edges compared with interiors (Fig. 4). True, this is not as great as the 10-fold greater values for richness and 2-fold higher values for measures of network architecture, but we feel these values are substantial nonetheless.

Discussion:

"we show that habitat edges can buffer the likelihood of pollinator network to collapse with declining forest area" - I don't think that the results permit to make this statement. Although network robustness was greater at edges, it is possible that the species at the edges differ from those in the interior. Thus, interior pollinator networks could collapse and only the edge networks would remain. If there are spillover effects from the edge into interior, I think it may even be possible that the increased abundance of generalist pollinators at the edge may outcompete specialist pollinators in the interior, resulting in species and interaction losses.

- Pavel Dodonov

R2.36 Thank you for the time and effort taken in reviewing the manuscript. We hope that the detailed responses given above (particularly, responses R2.08 and R2.35) have addressed the key points. Yes, species composition differs between edge and interior, but the observed pattern in this study system is not one in which there is predominantly specialist species in the interior and generalist species at the edge. More species in edge communities are actually highly specialized compared to interior sites (see new Fig. S4). So, in this situation, it does not seem to be the case that interior specialist species might be lost from small forest fragments leaving only ‘unimportant’ generalist species. Instead, the observed findings match our conclusions that the

higher availability of floral resources at the edge actually buffer small forest fragments from loss of biodiversity or network collapse (compared to the situation where there was no edge availability, or light-gap availability in secondary forests).

We have re-stated the sentence mentioned at the start of the discussion to try and be clearer; i.e., *“In our study system, forest edges had positive effects, buffering rather than exacerbating the decrease in network robustness to species extinction caused by forest area loss.”* (Lines 189-191).

As mentioned above, we feel justified in calling this a ‘buffering’ effect of edges because of the statistically significant interaction effect between decreasing area and increasing proximity to edge. In Fig. 4, this is reflected in the diverging slopes of the fitted lines as area decreases. On large islands, there is ‘only’ a 10-20% higher robustness to extinction at edges compared with interiors, whereas on small islands there is a 50-100% higher robustness to extinction at edges compared with interiors (Fig. 4).

It is important to reiterate that our conclusion is not just ‘whether’ there is an effect, but that the ‘direction’ of effect is actually the opposite of what is found in non-forest systems. In other words, anthropogenic edge effects are typically considered to exacerbate the effects of area loss on pollinators, but here we show that forest edges not only dampen the response but completeness reverse the sign of the effect.

Reviewer #3 (Remarks to the Author):

The manuscript entitled “Edges increase pollinator network robustness to extinction with declining forest area” by Ren and collaborators presents a study of pollination networks in a fragmented land-bridge island system. The study analyzes, through Structural Equation Models and null models, the direct and indirect effects of fragmentation variables on network robustness through community and network structural patterns. The authors found a positive effect of forest edges on biodiversity that somehow buffers the negative effect of decreasing fragment areas and preventing further extinctions. This result is interesting because it presents a holistic view of the habitat fragmentation process that contradicts our prejudices and provides a counterintuitive mechanism that allows communities to persist in modified habitats.

Despite its complexity, the methods applied are appropriate to test the hypothesis and the conclusions are coherent with the results. The null model approach to disentangle sampling effects vs fragmentation effects looks effective although a bit forced. Nevertheless, I think the approach is quite conservative and the conclusions made by the authors are appropriate. The codes applied are provided by the authors which make this study repeatable.

R3.01 Thank you very much for the positive assessment of our work.

Despite the manuscript being appealing, I have some concerns listed below:

1-I wonder if we are evaluating pollinator communities’ responses to fragmentation in the same way we evaluate other, forest-specialized organisms. Although evidence is always necessary, I think pollination-oriented studies should focus more on the vulnerability of interactions rather than focusing on species diversity and abundance. In the same logic and from a conservation perspective, the dilemma between interior vs edges should contemplate the identity of interactions occurring at both habitats. It could be possible that some interior interactions are more rare or specialized than those occurring at edges and its conservation value be higher. Accordingly, higher nestedness in edges vs higher modularity in interiors could reflect the presence of generalist, opportunistic species disrupting a syndrome-based pollination network and, consequently the co-evolutive dynamics. Although this sentence is merely speculative, the change in network patterns between interior and edges deserves more study, maybe involving trait-based analysis which is beyond the scope of this study. I would like to read what authors think about these topics and if possible, mention them in the discussion.

Also, the conclusions about edge-interior differences should consider what could be happening in the canopy of forest interiors to have a complete picture of what is going on in these islands. I

81understand that it would be logistically difficult but at least it should be mentioned in the discussion.

R3.02 *Change in network patterns between interior and edges:*

Thank you, these are all excellent points for discussion. As you pointed out, we are constrained in our ability to address all of the points analytically, because we do not have available trait data, or empirical samples from the forest canopy. However, we have included relevant discussion of the key points, and we have been able to quantitatively address one of the important points about the distribution of generalist versus specialist interaction partners at edges vs. interiors (Fig. S4). See also responses R2.07 above, in which Reviewer 2 makes a similar point.

Importantly, edges do not appear to predominantly harbour generalist ('opportunistic') species, and interior sites do not appear to be dominated by specialists. Edges also have many highly specialized species in our study system (Fig. S4). As mentioned in response to Reviewer 2, we have incorporated discussion of this point throughout the manuscript. More specifically, in one section of the discussion we have made sure to link this to potential caveats about interaction identity, behavioural plasticity (eg interaction re-wiring) and potential caveats about a 'static' measure of network robustness (Lines 238-269):

*“The positive effect of habitat edges is also reflected in the way that habitat area effects on network architecture were strongly buffered by proximity to edge. Plant-pollinator networks in the interior of smaller islands had low network size, higher relative connectance among interaction partners^{26, 54, 55, 56}, lower nestedness and higher network modularity. In fragmented habitats, it is typically thought that changes in network architecture are driven by a loss of specialist pollinators that visit only a limited number of plants and are more likely to become extinct than generalists^{12, 57}. The high extinction risk of specialists potentially breaks the degree of interaction asymmetry in smaller networks, which is a key attribute of nestedness⁵⁸, and finally reduces nestedness and increases the modularity of island interior networks. In our study, however, edges were not dominated solely by generalist species, but also had many highly specialist species. Instead, the greater floral resource availability and larger size of plant-pollinator networks at forest edges appear to buffer the impact of forest area loss on relative connectance, and significantly increase nestedness, decrease modularity and increase network robustness to plant extinction. Naturally, an important caveat on inferring changes in network robustness analytically (from observed patterns of network architecture) is that it reflects a **static** view of network fragility, which does not allow for plasticity of behavioural responses such as interaction rewiring in the face of resource extinction. Although this warrants further study, particularly from trait-based and phylogenetic perspectives on edge alteration of coevolved*

pollination syndromes, we would expect that any potential effects of interaction rewiring on network robustness should be greatest for generalist compared with specialist pollinators. But again, we did not find that edges predominantly harboured more generalist ‘opportunistic’ species in our study system. As in previous studies^{32, 34, 59}, increasing network nestedness in response to decreasing habitat area appears to be the key factor in mitigating secondary extinctions from species loss. It is increasingly appreciated that anthropogenic stressors can potentially offset one another, reducing secondary extinction risk⁵⁷. Although our effect size for edge influence on network robust is small, the direction of effect we find in our forest system is actually the opposite of what is found in non-forest systems. Anthropogenic edge effects are typically considered to exacerbate the effects of forest area loss on pollinators, but here we show that forest edges not only dampen the response but completely reverse the sign of the effect. In fragmented secondary forest, our research provides evidence that habitat edges have the potential to play a positive role.”

Floral resources in the forest canopy: Again, this is an important point, and we now incorporate this into our discussion.

Forest canopies can provide important floral resources, but are critically undersampled. In previous research, one of us (Didham & Ewers 2014; <https://doi.org/10.2984/68.4.4>) tested microclimatic gradients at a forest edge and pointed out that the vertical microclimate gradient down from the hot sunny upper-forest canopy to the ground is analogous to the horizontal microclimate gradient from the forest edge to the interior. We also compared capture rates of bumblebees in the canopy versus the ground of a temperate forest and found that floral resources in the canopy are likely very important to inferring edge responses of pollinator insects (Ewers et al. 2013; <https://doi.org/10.1111/icad.12014>). So, we completely agree with this point.

Unfortunately, in the current study we do not have comparable canopy samples from our sites. However, in our particular forest type it would be very unlikely for canopy sampling to have made a substantive difference to our findings, because the forest is dominated by gymnosperms (predominantly *Pinus massoniana*) and there is a low proportion of canopy angiosperm trees or flowering lianas.

For future studies, one of the great challenges to overcome will be the logistical difficulties of sampling in the canopy, particularly to obtain standardised floral contact observations and hand-netting of voucher specimens.

In light of the important comment about forest canopies, we have modified the discussion section: i.e., *“However, previous network studies have been conducted almost entirely in non-forest systems in which the relative contrast in light conditions and floral resources between edge and interior is weak (or favours interior habitats), whereas our findings for the forested system at TIL are more in line with the observation that pollinators prefer open sunny conditions rather than the dark interior within closed-canopy forests. For example, on the islands we studied, forest canopy density was high (~90% under *Pinus massoniana*)⁴⁰, few shrub species could persist (e.g., *Loropetalum chinense*, *Symplocos sumuntia* and *Symplocos paniculata*), herbs were rare, and only a few shade-tolerant species were seen flowering occasionally (e.g., *Liriope spicata* and *Lysimachia congestiflora*). In other forest systems, floral resources in the sunny, open environment of the upper canopy might be expected to have a positive effect similar to that described here for the forest edge^{41, 42}, but the TIL forests are dominated by gymnosperms (predominantly *Pinus massoniana*) with relatively few flowering lianas (*Rosa multiflora*, *Wisteria sinensis*, *Millettia dielsiana*). However, floral resource availability in the forest canopy warrants greater attention, notwithstanding the logistical challenges in ensuring standardized sampling^{43, 44}.”* Please see Lines 204-218 in the revised paper.

2-the robustness analysis often reflects a static view of network fragility by not allowing different behavioral responses of consumers in face of resource extinctions such as rewiring. How do the authors think the presence of opportunistic and generalist species in edges could affect network robustness?

R3.03 We agree this is an important limitation of the robustness measure. We now add an appropriate caveat that the robustness analysis reflects a static view of network fragility by not allowing differential behavioural (or life history) plasticity. This certainly could influence the conclusions if edges were dominated by opportunistic/generalist species and had few specialists, but our quantitative comparison of generalist vs specialists suggests that this is not the case in our system.

In light of this important point, we have amended the discussion as noted here (and in response R3.02 above): *“...and increase network robustness to plant extinction. Naturally, an important caveat on inferring changes in network robustness analytically (from observed patterns of network architecture) is that it reflects a static view of network fragility, which does not allow for plasticity of behavioural responses such as interaction rewiring in the face of resource extinction. Although this warrants further study, particularly from trait-based and phylogenetic*

perspectives on edge alteration of coevolved pollination syndromes, we would expect that any potential effects of interaction rewiring on network robustness should be greatest for generalist compared with specialist pollinators. But again, we did not find that edges predominantly harboured more generalist ‘opportunistic’ species in our study system.” (Lines 251-260).

3-I found the “relative connectance” idea interesting, nevertheless, its accuracy may depend on how complete is the sampling of the regional network and how you define the limits of the region. Some clarifications on this regard would be needed.

R3.04 Thank you for your suggestion, this is a good point. We have changed the word ‘region’, to avoid implying that we have sampled the whole regional species pool, and instead refer to ‘the total pool of network observations’. We agree that the accuracy of relative connectance depends on sampling completeness, which we feel is reasonable for our system with 20 surveys at each of 116 pairs of edge and interior transects over 3 years (Fig. S2). Nevertheless, sample coverage estimates in Fig. S1 suggest that many more plant-pollinator interactions remain to be observed. In the revised paper, we have added appropriate caveats on these points in the discussion, i.e., *“The accuracy of relative connectance depends on the level of sampling completeness and the degree to which pooling of the sampled networks adequately represents the true regional network as a whole. Therefore, long-term observation data are necessary when using this parameter.”* Please see Lines 401-404 in the revised paper.

4-Is the orientation of edges important in your area? In some cases the orientation of edges can have a huge impact on microclimatic conditions, affecting community structure (Bernaschini et al. 2020, 2021).

R3.05 Thank you for the literature suggestions. In our study system, the orientation of edges is also important, since the plant species composition on different aspects varies somewhat. When we sampled, we used at least 4 edge transects on large islands (the 7 islands ≥ 10 ha), which explicitly covered the four aspects (east, west, north and south); whereas there were 1, 2 or 3 transects on small islands (34 islands, <10 ha), which were randomly selected with respect to aspect, but could not completely cover edges on all four aspects. However, note that on the smallest islands, even a single edge transect wrapped around a reasonable portion of the island, allowing coverage of both the shady and sunny sides of each island as much as possible. In the revised paper, we have now added appropriate description of these points in the methods section, i.e., *“When we sampled, we used at least 4 paired edge versus interior transects on large islands*

(7 islands ≥ 10 ha), which allowed us to cover all four aspects (east, west, north, south), whereas there were 1, 2 or 3 paired transects randomly located on small islands (34 islands < 10 ha; Table S1), which could not completely cover edges on the four aspects. However, on the smallest islands, even a single edge transect wrapped around a reasonable portion of the island, allowing coverage of both the shady and sunny sides of each island to the greatest degree possible.” (Lines 336-342).

5-SI, line 73-74. I think it is not necessarily so. In a highly compartmentalized network with strong boundaries between modules adding new species would not necessarily increase nestedness.

R3.06 Yes, that is an important consideration that we had not mentioned. We have re-worded the SI Lines 127-132 to raise this as an important exception to the generally observed observations in the literature. We feel that the hypothesis still represents the strongest *a priori* logic for the relationships among variables, but we now note that there could indeed be cases where a highly compartmentalised network has strong boundaries between modules, such that the addition of a new species does not necessarily alter nestedness (Cai *et al.* 2020).

In light of this comment, we have made additional effort to improve the wording, i.e., “In keeping with these generally observed relationships, we hypothesized that an increase in plant and pollinator richness would potentially decrease connectance and nestedness, but increase modularity, while the opposite relationships would hold for floral resources and pollinator abundance. We note that these hypotheses would not be supported in cases where a highly compartmentalised network has strong boundaries between modules, since the addition of a new species would then not necessarily alter nestedness⁴⁵.” (Lines 127-132 of Appendix).

Cai, W., Snyder, J., Hastings, A., & D’Souza, R. M. (2020). Mutualistic networks emerging from adaptive niche-based interactions. *Nature communications*, 11(1), 1-10.

-line 213-214: replace “tends to suggest” with “suggest”

R3.07 Corrected, as suggested.

-lines 219-224: I don’t see so clear causality between high connectance and low nestedness. I think those variables are not necessarily related.

R3.08 Yes, in our study, those variables are not related. We are very sorry for the misunderstanding caused by our improper wording. We have carefully checked corresponding results and removed the suggestion of a causal relationship implied by words such as “consequently” and “leading to”, when what we intended to express was just a summary of attributes. So in the revised paper, we have improved the related sentence, i.e., “*Plant-pollinator networks in the interior of smaller islands had low network size, higher relative connectance among interaction partners*^{26, 54, 55, 56}, lower nestedness and higher network modularity.” (Lines 239-241).

-lines 230-231: “in the face of” phrase is repeated.

R3.09 In the revised paper, we deleted the second “in the face of” and improved the related sentence, i.e., “*As in previous studies*^{32, 34, 59}, increasing network nestedness in response to decreasing habitat area appears to be the key factor in mitigating secondary extinctions from species loss.” (Lines 260-262).

-Line 231: replace “link” with factor or similar word.

R3.10 We have replaced “link” with “factor” as suggested. Please see Line 261 in the revised paper.

References:

Bernaschini, M., M. Rossetti, G. Valladares, and A. Salvo. 2021. Microclimatic edge effects in a fragmented forest: disentangling the drivers of ecological processes in plant-leafminer-parasitoid food webs. *Ecological Entomology*.

Bernaschini, M., G. Valladares, and A. Salvo. 2020. Edge effects on insect–plant food webs: assessing the influence of geographical orientation and microclimatic conditions. *Ecological Entomology*.

R3.11 Thank you for the useful suggestions.

Decision Letter, first revision:

10th October 2022

87Dear Dr. Ding,

Thank you for submitting your revised manuscript "Forest edges increase pollinator network robustness to extinction with declining area" (NATECOLEVOL-220115647A). It has now been seen again by the original reviewers and their comments are below. The reviewers find that the paper has improved in revision, and therefore we'll be happy in principle to publish it in Nature Ecology & Evolution, pending minor revisions to satisfy the reviewers' final requests and to comply with our editorial and formatting guidelines.

[REDACTED]

Reviewer #2 (Remarks to the Author):

Dear Editor,

I have now reviewed the revised version of the manuscript "Forest edges increase pollinator network robustness to extinction with declining area".

Firstly, I would like to applaud and thank the authors for the detailed response. I am now convinced that the sampling design and analysis are valid and robust, and were probably the best choices to answer the study's questions. Thus, I suggest that this manuscript is accepted in its current form or following just some minor revisions (mostly adding some information to the manuscript, if permitted by the journal's style).

I would like to thank the authors for explaining the reason for the edge-interior transects. Given the explanation, I think this makes a lot of sense, especially considering the island topography. If possible, I would very much like to see the figure R.1 (photos of the islands) within the manuscript, perhaps in the same figure as the map. It really helps for readers to understand the kind of environment you are working in!

If possible, I suggest placing the information "floral resources were approximately 18.52 times higher at the edge than in the interior, and pollinator abundance 519 approximately 18.43 times higher at the edge. The 1.5 times difference in survey 522 interval cannot account for the very large difference in observed detections." in the manuscript, in addition to citing the supplementary material. To me

88this is convincing evidence that even if there were some bias in sampling, it did not meaningfully alter the results. If possible, also the information that "For the same level of survey effort, the likelihood of detecting all the flowers along the transect is higher in the interior (not lower), and the available observation time per flower is actually greater in the interior (because there are relatively few flowers)." My reason for this suggestions is that you want readers to see that the methods are generally valid without having to look deep in the supplementary material and this might be a key issue.

I also suggest adding some information on topography (e.g. "The average difference in elevation between edge transect and interior transect on each island varied from just 6.80-48.60 m across sites") to the methods.

Reviewer #3 (Remarks to the Author):

The revised version of the manuscript is a substantially clearer version of the study, although some aspects regarding sampling design and analyses could not be modified at this point the justification offered by the authors is solid. All my comments were answered carefully. In my opinion, this is an important piece of evidence for those interested in habitat fragmentation effects on natural communities.

Our ref: NATECOLEVOL-220115647A

25th October 2022

Dear Dr. Ding,

Thank you for your patience as we've prepared the guidelines for final submission of your Nature Ecology & Evolution manuscript, "Forest edges increase pollinator network robustness to extinction with declining area" (NATECOLEVOL-220115647A). Please carefully follow the step-by-step instructions provided in the attached file, and add a response in each row of the table to indicate the changes that you have made. Please also check and comment on any additional marked-up edits we

89have proposed within the text. Ensuring that each point is addressed will help to ensure that your revised manuscript can be swiftly handed over to our production team.

****We would like to start working on your revised paper, with all of the requested files and forms, as soon as possible (preferably within two weeks). Please get in contact with us immediately if you anticipate it taking more than two weeks to submit these revised files.****

In recognition of the time and expertise our reviewers provide to Nature Ecology & Evolution's editorial process, we would like to formally acknowledge their contribution to the external peer review of your manuscript entitled "Forest edges increase pollinator network robustness to extinction with declining area". For those reviewers who give their assent, we will be publishing their names alongside the published article.

Nature Ecology & Evolution offers a Transparent Peer Review option for new original research manuscripts submitted after December 1st, 2019. As part of this initiative, we encourage our authors to support increased transparency into the peer review process by agreeing to have the reviewer comments, author rebuttal letters, and editorial decision letters published as a Supplementary item. When you submit your final files please clearly state in your cover letter whether or not you would like to participate in this initiative. Please note that failure to state your preference will result in delays in accepting your manuscript for publication.

Cover suggestions

As you prepare your final files we encourage you to consider whether you have any images or illustrations that may be appropriate for use on the cover of Nature Ecology & Evolution.

Please submit your suggestions, clearly labeled, along with your final files. We'll be in touch if more

90information is needed.

Nature Ecology & Evolution has now transitioned to a unified Rights Collection system which will allow our Author Services team to quickly and easily collect the rights and permissions required to publish your work. Approximately 10 days after your paper is formally accepted, you will receive an email in providing you with a link to complete the grant of rights. If your paper is eligible for Open Access, our Author Services team will also be in touch regarding any additional information that may be required to arrange payment for your article.

Please note that *Nature Ecology & Evolution* is a Transformative Journal (TJ). Authors may publish their research with us through the traditional subscription access route or make their paper immediately open access through payment of an article-processing charge (APC). Authors will not be required to make a final decision about access to their article until it has been accepted. [Find out more about Transformative Journals](https://www.springernature.com/gp/open-research/transformative-journals)

Authors may need to take specific actions to achieve [compliance with funder and institutional open access mandates](https://www.springernature.com/gp/open-research/funding/policy-compliance-faqs). If your research is supported by a funder that requires immediate open access (e.g. according to [Plan S principles](https://www.springernature.com/gp/open-research/plan-s-compliance)) then you should select the gold OA route, and we will direct you to the compliant route where possible. For authors selecting the subscription publication route, the journal's standard licensing terms will need to be accepted, including [the journal's standard licensing terms](https://www.nature.com/nature-portfolio/editorial-policies/self-archiving-and-license-to-publish). Those licensing terms will supersede any other terms that the author or any third party may assert apply to any version of the manuscript.

[REDACTED]

[REDACTED]

Reviewer #2:
Remarks to the Author:
Dear Editor,

I have now reviewed the revised version of the manuscript "Forest edges increase pollinator network robustness to extinction with declining area".

Firstly, I would like to applaud and thank the authors for the detailed response. I am now convinced that the sampling design and analysis are valid and robust, and were probably the best choices to answer the study's questions. Thus, I suggest that this manuscript is accepted in its current form or following just some minor revisions (mostly adding some information to the manuscript, if permitted by the journal's style).

I would like to thank the authors for explaining the reason for the edge-interior transects. Given the explanation, I think this makes a lot of sense, especially considering the island topography. If possible, I would very much like to see the figure R.1 (photos of the islands) within the manuscript, perhaps in the same figure as the map. It really helps for readers to understand the kind of environment you are working in!

If possible, I suggest placing the information "floral resources were approximately 18.52 times higher at the edge than in the interior, and pollinator abundance 519 approximately 18.43 times higher at the edge. The 1.5 times difference in survey 522 interval cannot account for the very large difference in observed detections." in the manuscript, in addition to citing the supplementary material. To me this is convincing evidence that even if there were some bias in sampling, it did not meaningfully alter the results. If possible, also the information that "For the same level of survey effort, the likelihood of detecting all the flowers along the transect is higher in the interior (not lower), and the available observation time per flower is actually greater in the interior (because there are relatively few flowers)." My reason for this suggestions is that you want readers to see that the methods are generally valid without having to look deep in the supplementary material and this might be a key issue.

I also suggest adding some information on topography (e.g. "The average difference in elevation between edge transect and interior transect on each island varied from just 6.80-48.60 m across sites") to the methods.

Reviewer #3:
Remarks to the Author:

The revised version of the manuscript is a substantially clearer version of the study, although some aspects regarding sampling design and analyses could not be modified at this point the justification offered by the authors is solid. All my comments were answered carefully. In my opinion, this is an important piece of evidence for those interested in habitat fragmentation effects on natural communities.

92Final Decision Letter:

16th December 2022

Dear Professor Ding,

We are pleased to inform you that your Article entitled "Forest edges increase pollinator network robustness to extinction with declining area", has now been accepted for publication in Nature Ecology & Evolution.

Over the next few weeks, your paper will be copyedited to ensure that it conforms to Nature Ecology and Evolution style. Once your paper is typeset, you will receive an email with a link to choose the appropriate publishing options for your paper and our Author Services team will be in touch regarding any additional information that may be required

You will not receive your proofs until the publishing agreement has been received through our system

Due to the importance of these deadlines, we ask you please us know now whether you will be difficult to contact over the next month. If this is the case, we ask you provide us with the contact information (email, phone and fax) of someone who will be able to check the proofs on your behalf, and who will be available to address any last-minute problems . Once your paper has been scheduled for online publication, the Nature press office will be in touch to confirm the details.

Acceptance of your manuscript is conditional on all authors' agreement with our publication policies (see www.nature.com/authors/policies/index.html). In particular your manuscript must not be published elsewhere and there must be no announcement of the work to any media outlet until the publication date (the day on which it is uploaded onto our web site).

Please note that *Nature Ecology & Evolution* is a Transformative Journal (TJ). Authors may publish their research with us through the traditional subscription access route or make their paper immediately open access through payment of an article-processing charge (APC). Authors will not be required to make a final decision about access to their article until it has been accepted. [Find out more about Transformative Journals](https://www.springernature.com/gp/open-research/transformative-journals)

Authors may need to take specific actions to achieve [compliance](https://www.springernature.com/gp/open-research/funding/policy-compliance-faqs) with funder and institutional open access mandates. If your research

93is supported by a funder that requires immediate open access (e.g. according to [Plan S principles](https://www.springernature.com/gp/open-research/plan-s-compliance)) then you should select the gold OA route, and we will direct you to the compliant route where possible. For authors selecting the subscription publication route, the journal's standard licensing terms will need to be accepted, including <https://www.nature.com/nature-portfolio/editorial-policies/self-archiving-and-license-to-publish>. Those licensing terms will supersede any other terms that the author or any third party may assert apply to any version of the manuscript.

We welcome the submission of potential cover material (including a short caption of around 40 words) related to your manuscript; suggestions should be sent to Nature Ecology & Evolution as electronic files (the image should be 300 dpi at 210 x 297 mm in either TIFF or JPEG format). Please note that such pictures should be selected more for their aesthetic appeal than for their scientific content, and that colour images work better than black and white or grayscale images. Please do not try to design a cover with the Nature Ecology & Evolution logo etc., and please do not submit composites of images related to your work. I am sure you will understand that we cannot make any promise as to whether any of your suggestions might be selected for the cover of the journal.

You can generate the link yourself when you receive your article DOI by entering it here: <http://authors.springernature.com/share>.

[REDACTED]

94P.S. Click on the following link if you would like to recommend Nature Ecology & Evolution to your librarian <http://www.nature.com/subscriptions/recommend.html#forms>

** Visit the Springer Nature Editorial and Publishing website at http://editorial-jobs.springernature.com?utm_source=ejp_NEcoE_email&utm_medium=ejp_NEcoE_email&utm_campaign=ejp_NEcoE for more information about our career opportunities. If you have any questions please click [here](mailto:editorial.publishing.jobs@springernature.com).**